# GLOBAL CONVERGENCE OF THREE-LAYER NEURAL NETWORKS IN THE MEAN FIELD REGIME[*]

Huy Tuan Pham[†] and Phan-Minh Nguyen[‡ §]

## ABSTRACT

In the mean field regime, neural networks are appropriately scaled so that as the width tends to infinity, the learning dynamics tends to a nonlinear and nontrivial dynamical limit, known as the mean field limit. This lends a way to study large-width neural networks via analyzing the mean field limit. Recent works have successfully applied such analysis to two-layer networks and provided global convergence guarantees. The extension to multilayer ones however has been a highly challenging puzzle, and little is known about the optimization efficiency in the mean field regime when there are more than two layers.

In this work, we prove a global convergence result for unregularized feedforward three-layer networks in the mean field regime. We first develop a rigorous framework to establish the mean field limit of three-layer networks under stochastic gradient descent training. To that end, we propose the idea of a *neuronal embedding*, which comprises of a fixed probability space that encapsulates neural networks of arbitrary sizes. The identified mean field limit is then used to prove a global convergence guarantee under suitable regularity and convergence mode assumptions, which – unlike previous works on two-layer networks – does not rely critically on convexity. Underlying the result is a universal approximation property, natural of neural networks, which importantly is shown to hold at *any* finite training time (not necessarily at convergence) via an algebraic topology argument.

## 1 INTRODUCTION

Interests in the theoretical understanding of the training of neural networks have led to the recent discovery of a new operating regime: the neural network and its learning rates are scaled appropriately, such that as the width tends to infinity, the network admits a limiting learning dynamics in which all parameters evolve nonlinearly with time[1]. This is known as the mean field (MF) limit (Mei et al. (2018); Chizat & Bach (2018); Rotskoff & Vanden-Eijnden (2018); Sirignano & Spiliopoulos (2018); Nguyen (2019); Araújo et al. (2019); Sirignano & Spiliopoulos (2019)). The four works Mei et al. (2018); Chizat & Bach (2018); Rotskoff & Vanden-Eijnden (2018); Sirignano & Spiliopoulos (2018) led the first wave of efforts in 2018 and analyzed two-layer neural networks. They established a connection between the network under training and its MF limit. They then used the MF limit to prove that two-layer networks could be trained to find (near) global optima using variants of gradient descent, despite non-convexity (Mei et al. (2018); Chizat & Bach (2018)). The MF limit identified by these works assumes the form of gradient flows in the measure space, which factors out the invariance from the action of a symmetry group on the model. Interestingly, by lifting to the measure space, with a convex loss function (e.g. squared loss), one obtains a limiting optimization problem that is convex (Bengio et al. (2006); Bach (2017)). The analyses of Mei et al. (2018);

---

[*]This paper is a conference submission. We refer to the work Nguyen & Pham (2020) and its companion note Pham & Nguyen (2020) for generalizations as well as other conditions for global convergence in the case of multilayer neural networks.

[†]Department of Mathematics, Stanford University. This work was done in parts while H. T. Pham was at the University of Cambridge.

[‡]The Voleon Group. This work was done while P.-M. Nguyen was at Stanford University.

[§]The author ordering is randomized.

[1]This is to be contrasted with another major operating regime (the NTK regime) where parameters essentially do not evolve and the model behaves like a kernel method (Jacot et al. (2018); Chizat et al. (2019); Du et al. (2019); Allen-Zhu et al. (2019); Zou et al. (2018); Lee et al. (2019)).

Chizat & Bach (2018) utilize convexity, although the mechanisms to attain global convergence in these works are more sophisticated than the usual convex optimization setup in Euclidean spaces.

The extension to multilayer networks has enjoyed much less progresses. The works Nguyen (2019); Araújo et al. (2019); Sirignano & Spiliopoulos (2019) argued, heuristically or rigorously, for the existence of a MF limiting behavior under gradient descent training with different assumptions. In fact, it has been argued that the difficulty is not simply technical, but rather conceptual (Nguyen (2019)): for instance, the presence of intermediate layers exhibits multiple symmetry groups with intertwined actions on the model. Convergence to the global optimum of the model under gradient-based optimization has not been established when there are more than two layers.

In this work, we prove a global convergence guarantee for feedforward three-layer networks trained with unregularized stochastic gradient descent (SGD) in the MF regime. After an introduction of the three-layer setup and its MF limit in Section 2, our development proceeds in two main steps:

**Step 1 (Theorem 3 in Section 3):** We first develop a rigorous framework that describes the MF limit and establishes its connection with a large-width SGD-trained three-layer network. Here we propose the new idea of a *neuronal embedding*, which comprises of an appropriate non-evolving probability space that encapsulates neural networks of arbitrary sizes. This probability space is in general abstract and is constructed according to the (not necessarily i.i.d.) initialization scheme of the neural network. This idea addresses directly the intertwined action of multiple symmetry groups, which is the aforementioned conceptual obstacle (Nguyen (2019)), thereby covering setups that cannot be handled by formulations in Araújo et al. (2019); Sirignano & Spiliopoulos (2019) (see also Section 5 for a comparison). Our analysis follows the technique from Sznitman (1991); Mei et al. (2018) and gives a quantitative statement: in particular, the MF limit yields a good approximation of the neural network as long as $n_{\min}^{-1} \log n_{\max} \ll 1$ independent of the data dimension, where $n_{\min}$ and $n_{\max}$ are the minimum and maximum of the widths.

**Step 2 (Theorem 8 in Section 4):** We prove that the MF limit, given by our framework, converges to the global optimum under suitable regularity and convergence mode assumptions. Several elements of our proof are inspired by Chizat & Bach (2018); the technique in their work however does not generalize to our three-layer setup. Unlike previous two-layer analyses, we do not exploit convexity; instead we make use of a new element: a universal approximation property. The result turns out to be conceptually new: global convergence can be achieved even when the loss function is non-convex. An important crux of the proof is to show that the universal approximation property holds at *any* finite training time (but not necessarily at convergence, i.e. at infinite time, since the property may not realistically hold at convergence).

Together these two results imply a positive statement on the optimization efficiency of SGD-trained unregularized feedforward three-layer networks (Corollary 10). Our results can be extended to the general multilayer case – with new ideas on top and significantly more technical works – or used to obtain new global convergence guarantees in the two-layer case (Nguyen & Pham (2020); Pham & Nguyen (2020)). We choose to keep the current paper concise with the three-layer case being a prototypical setup that conveys several of the basic ideas. Complete proofs are presented in appendices.

**Notations.** $K$ denotes a generic constant that may change from line to line. $|\cdot|$ denotes the absolute value for a scalar and the Euclidean norm for a vector. For an integer $n$, we let $[n] = \{1, ..., n\}$.

## 2 THREE-LAYER NEURAL NETWORKS AND THE MEAN FIELD LIMIT

### 2.1 THREE-LAYER NEURAL NETWORK

We consider the following three-layer network at time $k \in \mathbb{N}_{\geq 0}$ that takes as input $x \in \mathbb{R}^d$:

$$\hat{\mathbf{y}}(x; \mathbf{W}(k)) = \varphi_3(\mathbf{H}_3(x; \mathbf{W}(k))), \tag{1}$$

$$\mathbf{H}_3(x; \mathbf{W}(k)) = \frac{1}{n_2} \sum_{j_2=1}^{n_2} \mathbf{w}_3(k, j_2) \varphi_2(\mathbf{H}_2(x, j_2; \mathbf{W}(k))),$$

$$\mathbf{H}_2\left(x, j_2; \mathbf{W}\left(k\right)\right) = \frac{1}{n_1} \sum_{j_1=1}^{n_1} \mathbf{w}_2\left(k, j_1, j_2\right) \varphi_1\left(\langle \mathbf{w}_1\left(k, j_1\right), x \rangle\right),$$

in which $\mathbf{W}\left(k\right) = \left(\mathbf{w}_1\left(k, \cdot\right), \mathbf{w}_2\left(k, \cdot, \cdot\right), \mathbf{w}_3\left(k, \cdot\right)\right)$ consists of the weights[2] $\mathbf{w}_1\left(k, j_1\right) \in \mathbb{R}^d$, $\mathbf{w}_2\left(k, j_1, j_2\right) \in \mathbb{R}$ and $\mathbf{w}_3\left(k, j_2\right) \in \mathbb{R}$. Here $\varphi_1 : \mathbb{R} \to \mathbb{R}$, $\varphi_2 : \mathbb{R} \to \mathbb{R}$ and $\varphi_3 : \mathbb{R} \to \mathbb{R}$ are the activation functions, and the network has widths $\{n_1, n_2\}$.

We train the network with SGD w.r.t. the loss $\mathcal{L} : \mathbb{R} \times \mathbb{R} \to \mathbb{R}_{\geq 0}$. We assume that at each time $k$, we draw independently a fresh sample $z\left(k\right) = \left(x\left(k\right), y\left(k\right)\right) \in \mathbb{R}^d \times \mathbb{R}$ from a training distribution $\mathcal{P}$. Given an initialization $\mathbf{W}\left(0\right)$, we update $\mathbf{W}\left(k\right)$ according to

$$\mathbf{w}_3\left(k+1, j_2\right) = \mathbf{w}_3\left(k, j_2\right) - \epsilon \xi_3\left(k\epsilon\right) \mathrm{Grad}_3\left(z\left(k\right), j_2; \mathbf{W}\left(k\right)\right),$$
$$\mathbf{w}_2\left(k+1, j_1, j_2\right) = \mathbf{w}_2\left(k, j_1, j_2\right) - \epsilon \xi_2\left(k\epsilon\right) \mathrm{Grad}_2\left(z\left(k\right), j_1, j_2; \mathbf{W}\left(k\right)\right),$$
$$\mathbf{w}_1\left(k+1, j_1\right) = \mathbf{w}_1\left(k, j_1\right) - \epsilon \xi_1\left(k\epsilon\right) \mathrm{Grad}_1\left(z\left(k\right), j_1; \mathbf{W}\left(k\right)\right),$$

in which $j_1 = 1, ..., n_1$, $j_2 = 1, ..., n_2$, $\epsilon \in \mathbb{R}_{>0}$ is the learning rate, $\xi_i : \mathbb{R}_{\geq 0} \mapsto \mathbb{R}_{\geq 0}$ is the learning rate schedule for $\mathbf{w}_i$, and for $z = \left(x, y\right)$, we define

$$\mathrm{Grad}_3\left(z, j_2; \mathbf{W}\left(k\right)\right) = \partial_2 \mathcal{L}\left(y, \hat{\mathbf{y}}\left(x; \mathbf{W}\left(k\right)\right)\right) \varphi_3'\left(\mathbf{H}_3\left(x; \mathbf{W}\left(k\right)\right)\right) \varphi_2\left(\mathbf{H}_2\left(x, j_2; \mathbf{W}\left(k\right)\right)\right),$$

$$\mathrm{Grad}_2\left(z, j_1, j_2; \mathbf{W}\left(k\right)\right) = \Delta_2^{\mathbf{H}}\left(z, j_2; \mathbf{W}\left(k\right)\right) \varphi_1\left(\langle \mathbf{w}_1\left(k, j_1\right), x \rangle\right),$$

$$\mathrm{Grad}_1\left(z, j_1; \mathbf{W}\left(k\right)\right) = \left(\frac{1}{n_2} \sum_{j_2=1}^{n_2} \Delta_2^{\mathbf{H}}\left(z, j_2; \mathbf{W}\left(k\right)\right) \mathbf{w}_2\left(k, j_1, j_2\right)\right) \varphi_1'\left(\langle \mathbf{w}_1\left(k, j_1\right), x \rangle\right) x,$$

$$\Delta_2^{\mathbf{H}}\left(z, j_2; \mathbf{W}\left(k\right)\right) = \partial_2 \mathcal{L}\left(y, \hat{\mathbf{y}}\left(x; \mathbf{W}\left(k\right)\right)\right) \varphi_3'\left(\mathbf{H}_3\left(x; \mathbf{W}\left(k\right)\right)\right) \mathbf{w}_3\left(k, j_2\right) \varphi_2'\left(\mathbf{H}_2\left(x, j_2; \mathbf{W}\left(k\right)\right)\right).$$

We note that this setup follows the same scaling w.r.t. $n_1$ and $n_2$, which is applied to both the forward pass and the learning rates in the backward pass, as Nguyen (2019).

## 2.2 MEAN FIELD LIMIT

The MF limit is a continuous-time infinite-width analog of the neural network under training. To describe it, we first introduce the concept of a *neuronal ensemble*. Given a product probability space $\left(\Omega, \mathcal{F}, P\right) = \left(\Omega_1 \times \Omega_2, \mathcal{F}_1 \times \mathcal{F}_1, P_1 \times P_2\right)$, we independently sample $C_i \sim P_i$, $i = 1, 2$. In the following, we use $\mathbb{E}_{C_i}$ to denote the expectation w.r.t. the random variable $C_i \sim P_i$ and $c_i$ to denote an arbitrary point $c_i \in \Omega_i$. The space $\left(\Omega, \mathcal{F}, P\right)$ is referred to as a *neuronal ensemble*.

Given a neuronal ensemble $\left(\Omega, \mathcal{F}, P\right)$, the MF limit is described by a time-evolving system with state/parameter $W\left(t\right)$, where the time $t \in \mathbb{R}_{\geq 0}$ and $W\left(t\right) = \left(w_1\left(t, \cdot\right), w_2\left(t, \cdot, \cdot\right), w_3\left(t, \cdot\right)\right)$ with $w_1 : \mathbb{R}_{\geq 0} \times \Omega_1 \to \mathbb{R}^d$, $w_2 : \mathbb{R}_{\geq 0} \times \Omega_1 \times \Omega_2 \to \mathbb{R}$ and $w_3 : \mathbb{R}_{\geq 0} \times \Omega_2 \to \mathbb{R}$. It entails the quantities:

$$\hat{y}\left(x; W\left(t\right)\right) = \varphi_3\left(H_3\left(x; W\left(t\right)\right)\right),$$
$$H_3\left(x; W\left(t\right)\right) = \mathbb{E}_{C_2}\left[w_3\left(t, C_2\right) \varphi_2\left(H_2\left(x, C_2; W\left(t\right)\right)\right)\right],$$
$$H_2\left(x, c_2; W\left(t\right)\right) = \mathbb{E}_{C_1}\left[w_2\left(t, C_1, c_2\right) \varphi_1\left(\langle w_1\left(t, C_1\right), x \rangle\right)\right].$$

Here for each $t \in \mathbb{R}_{\geq 0}$, $w_1\left(t, \cdot\right)$ is $\left(\Omega_1, \mathcal{F}_1\right)$-measurable, and similar for $w_2\left(t, \cdot, \cdot\right)$, $w_3\left(t, \cdot\right)$. The MF limit evolves according to a continuous-time dynamics, described by a system of ODEs, which we refer to as the *MF ODEs*. Specifically, given an initialization $W\left(0\right) = \left(w_1\left(0, \cdot\right), w_2\left(0, \cdot, \cdot\right), w_3\left(0, \cdot\right)\right)$, the dynamics solves:

$$\partial_t w_3\left(t, c_2\right) = -\xi_3\left(t\right) \Delta_3\left(c_2; W\left(t\right)\right),$$
$$\partial_t w_2\left(t, c_1, c_2\right) = -\xi_2\left(t\right) \Delta_2\left(c_1, c_2; W\left(t\right)\right),$$
$$\partial_t w_1\left(t, c_1\right) = -\xi_1\left(t\right) \Delta_1\left(c_1; W\left(t\right)\right).$$

Here $c_1 \in \Omega_1$, $c_2 \in \Omega_2$, $\mathbb{E}_Z$ denotes the expectation w.r.t. the data $Z = \left(X, Y\right) \sim \mathcal{P}$, and for $z = \left(x, y\right)$, we define

$$\Delta_3\left(c_2; W\left(t\right)\right) = \mathbb{E}_Z\left[\partial_2 \mathcal{L}\left(Y, \hat{y}\left(X; W\left(t\right)\right)\right) \varphi_3'\left(H_3\left(X; W\left(t\right)\right)\right) \varphi_2\left(H_2\left(X, c_2; W\left(t\right)\right)\right)\right],$$

---

[2]To absorb first layer's bias term to $\mathbf{w}_1$, we assume the input $x$ to have 1 appended to the last entry.

$$\Delta_2\left(c_1, c_2; W\left(t\right)\right) = \mathbb{E}_Z\left[\Delta_2^H\left(Z, c_2; W\left(t\right)\right)\varphi_1\left(\langle w_1\left(t, c_1\right), X\rangle\right)\right],$$

$$\Delta_1\left(c_1; W\left(t\right)\right) = \mathbb{E}_Z\left[\mathbb{E}_{C_2}\left[\Delta_2^H\left(Z, C_2; W\left(t\right)\right) w_2\left(t, c_1, C_2\right)\right]\varphi_1'\left(\langle w_1\left(t, c_1\right), X\rangle\right) X\right],$$

$$\Delta_2^H\left(z, c_2; W\left(t\right)\right) = \partial_2\mathcal{L}\left(y, \hat{y}\left(x; W\left(t\right)\right)\right)\varphi_3'\left(H_3\left(x; W\left(t\right)\right)\right) w_3\left(t, c_2\right)\varphi_2'\left(H_2\left(x, c_2; W\left(t\right)\right)\right).$$

In Appendix B, we show well-posedness of MF ODEs under the following regularity conditions.

**Assumption 1** (Regularity). *We assume that $\varphi_1$ and $\varphi_2$ are $K$-bounded, $\varphi_1'$, $\varphi_2'$ and $\varphi_3'$ are $K$-bounded and $K$-Lipschitz, $\varphi_2'$ and $\varphi_3'$ are non-zero everywhere, $\partial_2\mathcal{L}\left(\cdot, \cdot\right)$ is $K$-Lipschitz in the second variable and $K$-bounded, and $|X| \leq K$ with probability 1. Furthermore $\xi_1$, $\xi_2$ and $\xi_3$ are $K$-bounded and $K$-Lipschitz.*

**Theorem 1.** *Under Assumption 1, given any neuronal ensemble and an initialization $W\left(0\right)$ such that[3] ess-sup $|w_2\left(0, C_1, C_2\right)|$, ess-sup $|w_3\left(0, C_2\right)| \leq K$, there exists a unique solution $W$ to the MF ODEs on $t \in [0, \infty)$.*

An example of a suitable setup is $\varphi_1 = \varphi_2 = \tanh$, $\varphi_3$ is the identity, $\mathcal{L}$ is the Huber loss, although a non-convex sufficiently smooth loss function suffices. In fact, all of our developments can be easily modified to treat the squared loss with an additional assumption $|Y| \leq K$ with probability 1.

So far, given an arbitrary neuronal ensemble $\left(\Omega, \mathcal{F}, P\right)$, for each initialization $W\left(0\right)$, we have defined a MF limit $W\left(t\right)$. The connection with the neural network's dynamics $\mathbf{W}\left(k\right)$ is established in the next section.

## 3    CONNECTION BETWEEN NEURAL NETWORK AND ITS MEAN FIELD LIMIT

### 3.1    NEURONAL EMBEDDING AND THE COUPLING PROCEDURE

To formalize a connection between the neural network and its MF limit, we consider their initializations. In practical scenarios, to set the initial parameters $\mathbf{W}\left(0\right)$ of the neural network, one typically randomizes $\mathbf{W}\left(0\right)$ according to some distributional law $\rho$. We note that since the neural network is defined w.r.t. a set of finite integers $\{n_1, n_2\}$, so is $\rho$. We consider a family Init of initialization laws, each of which is indexed by the set $\{n_1, n_2\}$:

Init $= \{\rho : \rho$ is the initialization law of a neural network of size $\{n_1, n_2\}$, $n_1, n_2 \in \mathbb{N}_{>0}\}$.

This is helpful when one is to take a limit that sends $n_1, n_2 \to \infty$, in which case the size of this family $|$Init$|$ is infinite. More generally we allow $|$Init$| < \infty$ (for example, Init contains a single law $\rho$ of a network of size $\{n_1, n_2\}$ and hence $|$Init$| = 1$). We make the following crucial definition.

**Definition 2.** Given a family of initialization laws Init, we call $\left(\Omega, \mathcal{F}, P, \left\{w_i^0\right\}_{i=1,2,3}\right)$ a *neuronal embedding* of Init if the following holds:

1. $\left(\Omega, \mathcal{F}, P\right) = \left(\Omega_1 \times \Omega_2, \mathcal{F}_1 \times \mathcal{F}_2, P_1 \times P_2\right)$ a product measurable space. As a reminder, we call it a neuronal ensemble.

2. The deterministic functions $w_1^0 : \Omega_1 \to \mathbb{R}^d$, $w_2^0 : \Omega_1 \times \Omega_2 \to \mathbb{R}$ and $w_3^0 : \Omega_2 \to \mathbb{R}$ are such that, for each index $\{n_1, n_2\}$ of Init and the law $\rho$ of this index, if — with an abuse of notations — we independently sample $\left\{C_i\left(j_i\right)\right\}_{j_i \in [n_i]} \sim P_i$ i.i.d. for each $i = 1, 2$, then

   $$\text{Law}\left(w_1^0\left(C_1\left(j_1\right)\right), w_2^0\left(C_1(j_1), C_2\left(j_2\right)\right), w_3^0\left(C_2(j_2)\right), j_i \in [n_i], i = 1, 2\right) = \rho.$$

To proceed, given Init and $\{n_1, n_2\}$ in its index set, we perform the following *coupling procedure*:

1. Let $\left(\Omega, \mathcal{F}, P, \left\{w_i^0\right\}_{i=1,2,3}\right)$ be a neuronal embedding of Init.

2. We form the MF limit $W\left(t\right)$ (for $t \in \mathbb{R}_{\geq 0}$) associated with the neuronal ensemble $\left(\Omega, \mathcal{F}, P\right)$ by setting the initialization $W\left(0\right)$ to $w_1\left(0, \cdot\right) = w_1^0\left(\cdot\right)$, $w_2\left(0, \cdot, \cdot\right) = w_2^0\left(\cdot, \cdot\right)$ and $w_3\left(0, \cdot\right) = w_3^0\left(\cdot\right)$ and running the MF ODEs described in Section 2.2.

---

[3]We recall the definition of ess-sup in Appendix A.

3. We independently sample $C_i(j_i) \sim P_i$ for $i = 1, 2$ and $j_i = 1, ..., n_i$. We then form the neural network initialization $\mathbf{W}(0)$ with $\mathbf{w}_1(0, j_1) = w_1^0(C_1(j_1))$, $\mathbf{w}_2(0, j_1, j_2) = w_2^0(C_1(j_1), C_2(j_2))$ and $\mathbf{w}_3(0, j_2) = w_3^0(C_2(j_2))$ for $j_1 \in [n_1]$, $j_2 \in [n_2]$. We obtain the network's trajectory $\mathbf{W}(k)$ for $k \in \mathbb{N}_{\geq 0}$ as in Section 2.1, with the data $z(k)$ generated independently of $\{C_i(j_i)\}_{i=1,2}$ and hence $\mathbf{W}(0)$.

We can then define a measure of closeness between $\mathbf{W}(\lfloor t/\epsilon \rfloor)$ and $W(t)$ for $t \in [0, T]$:

$$\mathscr{D}_T(W, \mathbf{W}) = \sup \Big\{ |\mathbf{w}_1(\lfloor t/\epsilon \rfloor, j_1) - w_1(t, C_1(j_1))|, \ |\mathbf{w}_2(\lfloor t/\epsilon \rfloor, j_1, j_2) - w_2(t, C_1(j_1), C_2(j_2))|,$$
$$|\mathbf{w}_3(\lfloor t/\epsilon \rfloor, j_2) - w_3(t, C_2(j_2))| : \ t \leq T, \ j_1 \leq n_1, \ j_2 \leq n_2 \Big\}. \tag{2}$$

Note that $W(t)$ is a deterministic trajectory independent of $\{n_1, n_2\}$, whereas $\mathbf{W}(k)$ is random for all $k \in \mathbb{N}_{\geq 0}$ due to the randomness of $\{C_i(j_i)\}_{i=1,2}$ and the generation of the training data $z(k)$. Similarly $\mathscr{D}_T(W, \mathbf{W})$ is a random quantity.

The idea of the coupling procedure is closely related to the coupling argument in Sznitman (1991); Mei et al. (2018). Here, instead of playing the role of a proof technique, the coupling serves as a vehicle to establish the connection between $W$ and $\mathbf{W}$ on the basis of the neuronal embedding. This connection is shown in Theorem 3 below, which gives an upper bound on $\mathscr{D}_T(W, \mathbf{W})$.

We note that the coupling procedure can be carried out to provide a connection between $W$ and $\mathbf{W}$ *as long as there exists a neuronal embedding for* Init. Later in Section 4.1, we show that for a common initialization scheme (in particular, i.i.d. initialization) for Init, there exists a neuronal embedding. Theorem 3 applies to, but is not restricted to, this initialization scheme.

## 3.2 MAIN RESULT: APPROXIMATION BY THE MF LIMIT

**Assumption 2** (Initialization of second and third layers). *We assume that* ess-sup $|w_2^0(C_1, C_2)|$, ess-sup $|w_3^0(C_2)| \leq K$, *where $w_2^0$ and $w_3^0$ are as described in Definition 2*.

**Theorem 3.** *Given a family* Init *of initialization laws and a tuple $\{n_1, n_2\}$ that is in the index set of* Init, *perform the coupling procedure as described in Section 3.1. Fix a terminal time $T \in \epsilon\mathbb{N}_{\geq 0}$. Under Assumptions 1 and 2, for $\epsilon \leq 1$, we have with probability at least $1 - 2\delta$,*

$$\mathscr{D}_T(W, \mathbf{W}) \leq e^{K_T} \left( \frac{1}{\sqrt{n_{\min}}} + \sqrt{\epsilon} \right) \log^{1/2} \left( \frac{3(T+1) n_{\max}^2}{\delta} + e \right) \equiv \mathsf{err}_{\delta, T}(\epsilon, n_1, n_2),$$

*in which $n_{\min} = \min\{n_1, n_2\}$, $n_{\max} = \max\{n_1, n_2\}$, and $K_T = K(1 + T^K)$.*

The theorem gives a connection between $\mathbf{W}(\lfloor t/\epsilon \rfloor)$, which is defined upon finite widths $n_1$ and $n_2$, and the MF limit $W(t)$, whose description is independent of $n_1$ and $n_2$. It lends a way to extract properties of the neural network in the large-width regime.

**Corollary 4.** *Under the same setting as Theorem 3, consider any test function $\psi : \mathbb{R} \times \mathbb{R} \to \mathbb{R}$ which is $K$-Lipschitz in the second variable uniformly in the first variable (an example of $\psi$ is the loss $\mathcal{L}$). For any $\delta > 0$, with probability at least $1 - 3\delta$,*

$$\sup_{t \leq T} |\mathbb{E}_Z[\psi(Y, \hat{\mathbf{y}}(X; \mathbf{W}(\lfloor t/\epsilon \rfloor)))] - \mathbb{E}_Z[\psi(Y, \hat{y}(X; W(t)))]| \leq e^{K_T} \mathsf{err}_{\delta, T}(\epsilon, n_1, n_2).$$

These bounds hold for any $n_1$ and $n_2$, similar to Mei et al. (2018); Araújo et al. (2019), in contrast with non-quantitative results in Chizat & Bach (2018); Sirignano & Spiliopoulos (2019). These bounds suggest that $n_1$ and $n_2$ can be chosen independent of the data dimension $d$. This agrees with the experiments in Nguyen (2019), which found width $\approx 1000$ to be typically sufficient to observe MF behaviors in networks trained with real-life high-dimensional data.

We observe that the MF trajectory $W(t)$ is defined as per the choice of the neuronal embedding $(\Omega, \mathcal{F}, P, \{w_i^0\}_{i=1,2,3})$, which may not be unique. On the other hand, the neural network's trajectory $\mathbf{W}(k)$ depends on the randomization of the initial parameters $\mathbf{W}(0)$ according to an initialization law from the family Init (as well as the data $z(k)$) and hence is independent of this choice. Another corollary of Theorem 3 is that given the same family Init, the law of the MF trajectory is insensitive to the choice of the neuronal embedding of Init.

**Corollary 5.** *Consider a family* Init *of initialization laws, indexed by a set of tuples* $\{m_1, m_2\}$ *that contains a sequence of indices* $\{m_1(m), m_2(m): m \in \mathbb{N}\}$ *in which as* $m \to \infty$, $\min \{m_1(m), m_2(m)\}^{-1} \log (\max \{m_1(m), m_2(m)\}) \to 0$. *Let* $W(t)$ *and* $\hat{W}(t)$ *be two MF trajectories associated with two choices of neuronal embeddings of* Init, $(\Omega, \mathcal{F}, P, \{w_i^0\}_{i=1,2,3})$ *and* $(\hat{\Omega}, \hat{\mathcal{F}}, \hat{P}, \{\hat{w}_i^0\}_{i=1,2,3})$. *The following statement holds for any* $T \geq 0$ *and any two positive integers* $n_1$ *and* $n_2$*: if we independently sample* $C_i(j_i) \sim P_i$ *and* $\hat{C}_i(j_i) \sim \hat{P}_i$ *for* $j_i \in [n_i]$, $i = 1, 2$, *then* $\mathrm{Law}(\mathcal{W}(n_1, n_2, T)) = \mathrm{Law}(\hat{\mathcal{W}}(n_1, n_2, T))$, *where we define* $\mathcal{W}(n_1, n_2, T)$ *as the below collection w.r.t.* $W(t)$, *and similarly define* $\hat{\mathcal{W}}(n_1, n_2, T)$ *w.r.t.* $\hat{W}(t)$:

$$\mathcal{W}(n_1, n_2, T) = \{w_1(t, C_1(j_1)),\ w_2(t, C_1(j_1), C_2(j_2)),\ w_3(t, C_2(j_2)):$$
$$j_1 \in [n_1],\ j_2 \in [n_2],\ t \in [0, T]\}.$$

The proofs are deferred to Appendix C.

## 4 CONVERGENCE TO GLOBAL OPTIMA

In this section, we prove a global convergence guarantee for three-layer neural networks via the MF limit. We consider a common class of initialization: i.i.d. initialization.

### 4.1 I.I.D. INITIALIZATION

**Definition 6.** An initialization law $\rho$ for a neural network of size $\{n_1, n_2\}$ is called $(\rho^1, \rho^2, \rho^3)$-i.i.d. initialization (or i.i.d. initialization, for brevity), where $\rho^1$, $\rho^2$ and $\rho^3$ are probability measures over $\mathbb{R}^d$, $\mathbb{R}$ and $\mathbb{R}$ respectively, if $\{\mathbf{w}_1(0, j_1)\}_{j_1 \in [n_1]}$ are generated i.i.d. according to $\rho^1$, $\{\mathbf{w}_2(0, j_1, j_2)\}_{j_1 \in [n_1],\, j_2 \in [n_2]}$ are generated i.i.d. according to $\rho^2$ and $\{\mathbf{w}_3(0, j_2)\}_{j_2 \in [n_2]}$ are generated i.i.d. according to $\rho^3$, and $\mathbf{w}_1$, $\mathbf{w}_2$ and $\mathbf{w}_3$ are independent.

Observe that given $(\rho^1, \rho^2, \rho^3)$, one can build a family Init of i.i.d. initialization laws that contains *any* index set $\{n_1, n_2\}$. Furthermore i.i.d. initializations are supported by our framework, as stated in the following proposition and proven in Appendix D.

**Proposition 7.** *There exists a neuronal embedding* $\left(\Omega, \mathcal{F}, P, \{w_i^0\}_{i=1,2,3}\right)$ *for any family* Init *of initialization laws, which are* $(\rho^1, \rho^2, \rho^3)$*-i.i.d.*

### 4.2 MAIN RESULT: GLOBAL CONVERGENCE

To measure the learning quality, we consider the loss averaged over the data $Z \sim \mathcal{P}$:

$$\mathscr{L}(V) = \mathbb{E}_Z[\mathcal{L}(Y, \hat{y}(X; V))],$$

where $V = (v_1, v_2, v_3)$ is a set of three measurable functions $v_1 : \Omega_1 \to \mathbb{R}^d$, $v_2 : \Omega_1 \times \Omega_2 \to \mathbb{R}$, $v_3 : \Omega_2 \to \mathbb{R}$.

**Assumption 3.** *Consider a neuronal embedding* $\left(\Omega, \mathcal{F}, P, \{w_i^0\}_{i=1,2,3}\right)$ *of the* $(\rho^1, \rho^2, \rho^3)$*-i.i.d. initialization, and the associated MF limit with initialization* $W(0)$ *such that* $w_1(0, \cdot) = w_1^0(\cdot)$, $w_2(0, \cdot, \cdot) = w_2^0(\cdot, \cdot)$ *and* $w_3(0, \cdot) = w_3^0(\cdot)$. *Assume:*

1. *Support: The support of* $\rho^1$ *is* $\mathbb{R}^d$.

2. *Convergence mode: There exist limits* $\bar{w}_1$, $\bar{w}_2$ *and* $\bar{w}_3$ *such that as* $t \to \infty$,

$$\mathbb{E}\left[(1 + |\bar{w}_3(C_2)|)\,|\bar{w}_3(C_2)|\,|\bar{w}_2(C_1, C_2)|\,|w_1(t, C_1) - \bar{w}_1(C_1)|\right] \to 0, \quad (3)$$
$$\mathbb{E}\left[(1 + |\bar{w}_3(C_2)|)\,|\bar{w}_3(C_2)|\,|w_2(t, C_1, C_2) - \bar{w}_2(C_1, C_2)|\right] \to 0, \quad (4)$$
$$\mathbb{E}\left[(1 + |\bar{w}_3(C_2)|)\,|w_3(t, C_2) - \bar{w}_3(C_2)|\right] \to 0, \quad (5)$$
$$\text{ess-sup}\,\mathbb{E}_{C_2}\left[|\partial_t w_2(t, C_1, C_2)|\right] \to 0. \quad (6)$$

3. *Universal approximation:* $\left\{\varphi_1\left(\langle u, \cdot\rangle\right) : u \in \mathbb{R}^d\right\}$ *has dense span in* $L^2\left(\mathcal{P}_X\right)$ *(the space of square integrable functions w.r.t.* $\mathcal{P}_X$ *the distribution of the input* $X$*).*

Assumption 3 is inspired by the work Chizat & Bach (2018) on two-layer networks, with certain differences. Assumptions 3.1 and 3.3 are natural in neural network learning (Cybenko (1989); Chen & Chen (1995)), while we note Chizat & Bach (2018) does not utilize universal approximation. Similar to Chizat & Bach (2018), Assumption 3.2 is technical and does not seem removable. Note that this assumption specifies the mode of convergence and is not an assumption on the limits $\bar{w}_1, \bar{w}_2$ and $\bar{w}_3$. Specifically conditions (3)-(5) are similar to the convergence assumption in Chizat & Bach (2018). We differ from Chizat & Bach (2018) fundamentally in the essential supremum condition (6). On one hand, this condition helps avoid the Morse-Sard type condition in Chizat & Bach (2018), which is difficult to verify in general and not simple to generalize to the three-layer case. On the other hand, it turns out to be a natural assumption to make, in light of Remark 9 below.

We now state the main result of the section. The proof is in Appendix D.

**Theorem 8.** *Consider a neuronal embedding* $\left(\Omega, \mathcal{F}, P, \left\{w_i^0\right\}_{i=1,2,3}\right)$ *of* $\left(\rho^1, \rho^2, \rho^3\right)$*-i.i.d. initialization. Consider the MF limit corresponding to the network* (1)*, such that they are coupled together by the coupling procedure in Section 3.1, under Assumptions 1, 2 and 3. For simplicity, assume* $\xi_1\left(\cdot\right) = \xi_2\left(\cdot\right) = 1$*. Further assume either:*

- *(untrained third layer)* $\xi_3\left(\cdot\right) = 0$ *and* $w_3^0\left(C_2\right) \neq 0$ *with a positive probability, or*

- *(trained third layer)* $\xi_3\left(\cdot\right) = 1$ *and* $\mathscr{L}\left(w_1^0, w_2^0, w_3^0\right) < \mathbb{E}_Z\left[\mathcal{L}\left(Y, \varphi_3\left(0\right)\right)\right]$*.*

*Then the following hold:*

- *Case 1 (convex loss): If* $\mathcal{L}$ *is convex in the second variable, then*

$$\lim_{t\to\infty} \mathscr{L}\left(W\left(t\right)\right) = \inf_V \mathscr{L}\left(V\right) = \inf_{\tilde{y}:\ \mathbb{R}^d\to\mathbb{R}} \mathbb{E}_Z\left[\mathcal{L}\left(Y, \tilde{y}\left(X\right)\right)\right].$$

- *Case 2 (generic non-negative loss): Suppose that* $\partial_2\mathcal{L}\left(y, \hat{y}\right) = 0$ *implies* $\mathcal{L}\left(y, \hat{y}\right) = 0$*. If* $y = y(x)$ *is a function of* $x$*, then* $\mathscr{L}\left(W\left(t\right)\right) \to 0$ *as* $t \to \infty$*.*

Remarkably here the theorem allows for non-convex losses. A further inspection of the proof shows that no convexity-based property is used in Case 2 (see, for instance, the high-level proof sketch in Section 4.3); in Case 1, the key steps in the proof are the same, and the convexity of the loss function serves as a convenient technical assumption to handle the arbitrary extra randomness of $Y$ conditional on $X$. We also remark that the same proof of global convergence should extend beyond the specific fully-connected architecture considered here. Similar to previous results on SGD-trained two-layer networks Mei et al. (2018); Chizat & Bach (2018), our current result in the three-layer case is non-quantitative.

*Remark* 9. Interestingly there is a converse relation between global convergence and the essential supremum condition (6): under the same setting, global convergence is unattainable if condition (6) does not hold. A similar observation was made in Wojtowytsch (2020) for two-layer ReLU networks. A precise statement and its proof can be found in Appendix E.

The following result is straightforward from Theorem 8 and Corollary 4, establishing the optimization efficiency of the neural network with SGD.

**Corollary 10.** *Consider the neural network* (1)*. Under the same setting as Theorem 8, in Case 1,*

$$\lim_{t\to\infty} \lim_{n_1,n_2} \lim_{\epsilon\to 0} \mathbb{E}_Z\left[\mathcal{L}\left(Y, \hat{\mathbf{y}}\left(X; \mathbf{W}\left(\lfloor t/\epsilon\rfloor\right)\right)\right)\right] = \inf_{f_1,f_2,f_3} \mathscr{L}\left(f_1, f_2, f_3\right) = \inf_{\tilde{y}} \mathbb{E}_Z\left[\mathcal{L}\left(Y, \tilde{y}\left(X\right)\right)\right]$$

*in probability, where the limit of the widths is such that* $\min\left\{n_1, n_2\right\}^{-1} \log\left(\max\left\{n_1, n_2\right\}\right) \to 0$*. In Case 2, the same holds with the right-hand side being* 0*.*

## 4.3 HIGH-LEVEL IDEA OF THE PROOF

We give a high-level discussion of the proof. This is meant to provide intuitions and explain the technical crux, so our discussion may simplify and deviate from the actual proof.

Our first insight is to look at the second layer's weight $w_2$. At convergence time $t = \infty$, we expect to have zero movement and hence, denoting $W(\infty) = (\bar{w}_1, \bar{w}_2, \bar{w}_3)$:

$$\Delta_2(c_1, c_2; W(\infty)) = \mathbb{E}_Z\left[\Delta_2^H(Z, c_2; W(\infty))\, \varphi_1(\langle \bar{w}_1(c_1), X\rangle)\right] = 0,$$

for $P$-almost every $c_1$, $c_2$. Suppose for the moment that we are allowed to make an additional (strong) assumption on the limit $\bar{w}_1$: $\mathrm{supp}(\bar{w}_1(C_1)) = \mathbb{R}^d$. It implies that the universal approximation property, described in Assumption 3, holds at $t = \infty$; more specifically, it implies $\{\varphi_1(\langle \bar{w}_1(c_1), \cdot\rangle) : c_1 \in \Omega_1\}$ has dense span in $L^2(\mathcal{P}_X)$. This thus yields

$$\mathbb{E}_Z\left[\Delta_2^H(Z, c_2; W(\infty))\big| X = x\right] = 0,$$

for $\mathcal{P}$-almost every $x$. Recalling the definition of $\Delta_2^H$, one can then easily show that

$$\mathbb{E}_Z\left[\partial_2 \mathcal{L}(Y, \hat{y}(X; W(\infty)))| X = x\right] = 0.$$

Global convergence follows immediately; for example, in Case 2 of Theorem 8, this is equivalent to that $\partial_2 \mathcal{L}(y(x), \hat{y}(x; W(\infty))) = 0$ and hence $\mathcal{L}(y(x), \hat{y}(x; W(\infty))) = 0$ for $\mathcal{P}$-almost every $x$. In short, the gradient flow structure of the dynamics of $w_2$ provides a seamless way to obtain global convergence. Furthermore there is no critical reliance on convexity.

However this plan of attack has a potential flaw in the strong assumption that $\mathrm{supp}(\bar{w}_1(C_1)) = \mathbb{R}^d$, i.e. the universal approximation property holds at convergence time. Indeed there are setups where it is desirable that $\mathrm{supp}(\bar{w}_1(C_1)) \neq \mathbb{R}^d$ (Mei et al. (2018); Chizat (2019)); for instance, it is the case where the neural network is to learn some "sparse and spiky" solution, and hence the weight distribution at convergence time, if successfully trained, cannot have full support. On the other hand, one can entirely expect that if $\mathrm{supp}(w_1(0, C_1)) = \mathbb{R}^d$ initially at $t = 0$, then $\mathrm{supp}(w_1(t, C_1)) = \mathbb{R}^d$ at *any* finite $t \geq 0$. The crux of our proof is to show the latter without assuming $\mathrm{supp}(\bar{w}_1(C_1)) = \mathbb{R}^d$.

This task is the more major technical step of the proof. To that end, we first show that there exists a mapping $(t, u) \mapsto M(t, u)$ that maps from $(t, w_1(0, c_1)) = (t, u)$ to $w_1(t, c_1)$ via a careful measurability argument. This argument rests on a scheme that exploits the symmetry in the network evolution. Furthermore the map $M$ is shown to be continuous. The desired conclusion then follows from an algebraic topology argument that the map $M$ preserves a homotopic structure through time.

## 5 DISCUSSION

The MF literature is fairly recent. A long line of works (Nitanda & Suzuki (2017); Mei et al. (2018); Chizat & Bach (2018); Rotskoff & Vanden-Eijnden (2018); Sirignano & Spiliopoulos (2018); Wei et al. (2019); Javanmard et al. (2019); Mei et al. (2019); Shevchenko & Mondelli (2019); Wojtowytsch (2020)) have focused mainly on two-layer neural networks, taking an interacting particle system approach to describe the MF limiting dynamics as Wasserstein gradient flows. The three works Nguyen (2019); Araújo et al. (2019); Sirignano & Spiliopoulos (2019) independently develop different formulations for the MF limit in multilayer neural networks, under different assumptions. These works take perspectives that are different from ours. In particular, while the central object in Nguyen (2019) is a new abstract representation of each individual neuron, our neuronal embedding idea instead takes a keen view on a whole ensemble of neurons. Likewise our idea is also distant from Araújo et al. (2019); Sirignano & Spiliopoulos (2019): the central objects in Araújo et al. (2019) are paths over the weights across layers; those in Sirignano & Spiliopoulos (2019) are time-dependent functions of the initialization, which are simplified upon i.i.d. initializations.

The result of our perspective is a neuronal embedding framework that allows one to describe the MF limit in a clean and rigorous manner. In particular, it avoids extra assumptions made in Araújo et al. (2019); Sirignano & Spiliopoulos (2019): unlike our work, Araújo et al. (2019) assumes untrained first and last layers and requires non-trivial technical tools; Sirignano & Spiliopoulos (2019) takes an unnatural sequential limit $n_1 \to \infty$ before $n_2 \to \infty$ and proves a non-quantitative result, unlike Theorem 3 which only requires sufficiently large $\min\{n_1, n_2\}$. We note that Theorem 3 can be extended to general multilayer networks using the neuronal embedding idea. The advantages of our framework come from the fact that while MF formulations in Araújo et al. (2019); Sirignano & Spiliopoulos (2019) are specific to and exploit i.i.d. initializations, our formulation does not. Remarkably as shown in Araújo et al. (2019), when there are more than three layers and no biases,

i.i.d. initializations lead to a certain simplifying effect on the MF limit. On the other hand, our framework supports non-i.i.d. initializations which avoid the simplifying effect, as long as there exist suitable neuronal embeddings (Nguyen & Pham (2020)). Although our global convergence result in Theorem 8 is proven in the context of i.i.d. initializations for three-layer networks, in the general multilayer case, it turns out that the use of a special type of non-i.i.d. initialization allows one to prove a global convergence guarantee (Pham & Nguyen (2020)).

In this aspect, our framework follows closely the spirit of the work Nguyen (2019), whose MF formulation is also not specific to i.i.d. initializations. Yet though similar in the spirit, Nguyen (2019) develops a heuristic formalism and does not prove global convergence.

Global convergence in the two-layer case with convex losses has enjoyed multiple efforts with a lot of new and interesting results (Mei et al. (2018); Chizat & Bach (2018); Javanmard et al. (2019); Rotskoff et al. (2019); Wei et al. (2019)). Our work is the first to establish a global convergence guarantee for SGD-trained three-layer networks in the MF regime. Our proof sends a new message that the crucial factor is not necessarily convexity, but rather that the whole learning trajectory maintains the universal approximation property of the function class represented by the first layer's neurons, together with the gradient flow structure of the second layer's weights. As a remark, our approach can also be applied to prove a similar global convergence guarantee for two-layer networks, removing the convex loss assumption in previous works (Nguyen & Pham (2020)). The recent work Lu et al. (2020) on a MF resnet model (a composition of many two-layer MF networks) and a recent update of Sirignano & Spiliopoulos (2019) essentially establish conditions of stationary points to be global optima. They however require strong assumptions on the support of the limit point. As explained in Section 4.3, we analyze the training dynamics without such assumption and in fact allow it to be violated.

Our global convergence result is non-quantitative. An important, highly challenging future direction is to develop a quantitative version of global convergence; previous works on two-layer networks Javanmard et al. (2019); Wei et al. (2019); Rotskoff et al. (2019); Chizat (2019) have done so under sophisticated modifications of the architecture and training algorithms.

Finally we remark that our insights here can be applied to prove similar global convergence guarantees and derive other sufficient conditions for global convergence of two-layer or multilayer networks (Nguyen & Pham (2020); Pham & Nguyen (2020)).

## ACKNOWLEDGEMENT

H. T. Pham would like to thank Jan Vondrak for many helpful discussions and in particular for the shorter proof of Lemma 19. We would like to thank Andrea Montanari for the succinct description of the difficulty in extending the mean field formulation to the multilayer case, in that there are multiple symmetry group actions in a multilayer network.

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

## A  NOTATIONAL PRELIMINARIES

For a real-valued random variable $Z$ defined on a probability space $(\Omega, \mathcal{F}, P)$, we recall

$$\text{ess-sup} Z = \inf \{z \in \mathbb{R} : \ P(Z > z) = 0\}.$$

We also introduce some convenient definitions which we use throughout the appendices. For a set of neural network's parameter $\mathbf{W}$, we define

$$\|\|\mathbf{W}\|\|_T = \max \left\{ \max_{j_1 \leq n_1, \, j_2 \leq n_2} \sup_{t \leq T} |\mathbf{w}_2 \left(\lfloor t/\epsilon \rfloor, j_1, j_2\right)|, \ \max_{j_2 \leq n_2} \sup_{t \leq T} |\mathbf{w}_3 \left(\lfloor t/\epsilon \rfloor, j_2\right)| \right\}.$$

Similarly for a set of MF parameters $W$, we define:

$$\|\|W\|\|_T = \max \left\{ \text{ess-sup} \sup_{t \leq T} |w_2 \left(t, C_1, C_2\right)|, \ \text{ess-sup} \sup_{t \leq T} |w_3 \left(t, C_2\right)| \right\}.$$

For two sets of neural network's parameters $\mathbf{W}', \mathbf{W}''$, we define their distance:

$$\|\mathbf{W}' - \mathbf{W}''\|_T = \sup \left\{ |\mathbf{w}_1' \left(\lfloor t/\epsilon \rfloor, j_1\right) - \mathbf{w}_1'' \left(\lfloor t/\epsilon \rfloor, j_1\right)|, \ |\mathbf{w}_2' \left(\lfloor t/\epsilon \rfloor, j_1, j_2\right) - \mathbf{w}_2'' \left(\lfloor t/\epsilon \rfloor, j_1, j_2\right)|, \right.$$
$$\left. |\mathbf{w}_3' \left(\lfloor t/\epsilon \rfloor, j_2\right) - \mathbf{w}_3'' \left(\lfloor t/\epsilon \rfloor, j_2\right)| : \ t \in [0, T], \ j_1 \in [n_1], \ j_2 \in [n_2] \right\}.$$

Similarly for two sets of MF parameters $W', W''$, we define their distance:

$$\|W' - W''\|_T = \text{ess-sup} \sup_{t \in [0,T]} \left\{ |w_1' \left(t, C_1\right) - w_1'' \left(t, C_1\right)|, \ |w_2' \left(t, C_1, C_2\right) - w_2'' \left(t, C_1, C_2\right)|, \right.$$
$$\left. |w_3' \left(t, C_2\right) - w_3'' \left(t, C_2\right)| \right\}.$$

## B  EXISTENCE AND UNIQUENESS OF THE SOLUTION TO MF ODES

We first collect some a priori estimates.

**Lemma 11.** *Under Assumption 1, consider a solution $W$ to the MF ODEs with initialization $W(0)$ such that $\|\|W\|\|_0 < \infty$. If this solution exists, it satisfies the following a priori bounds, for any $T \geq 0$:*

$$\text{ess-sup} \sup_{t \leq T} |w_3 \left(t, C_2\right)| \leq \|\|W\|\|_0 + KT \equiv \|\|W\|\|_0 + K_{0,3} \left(T\right),$$
$$\text{ess-sup} \sup_{t \leq T} |w_2 \left(t, C_1, C_2\right)| \leq \|\|W\|\|_0 + KTK_{0,3} \left(T\right) \equiv \|\|W\|\|_0 + K_{0,2} \left(T\right),$$

*and consequently, $\|\|W\|\|_T \leq 1 + \max \{K_{0,2} \left(T\right), \ K_{0,3} \left(T\right)\}.$*

*Proof.* The bounds can be obtained easily by bounding the respective initializations and update quantities separately. In particular,

$$\text{ess-sup} \sup_{t \leq T} |w_3 \left(t, C_2\right)| \leq \text{ess-sup} |w_3 \left(0, C_2\right)| + T \text{ess-sup} \sup_{t \leq T} \left| \frac{\partial}{\partial t} w_3 \left(t, C_2\right) \right| \leq \|\|W\|\|_0 + KT,$$

$$\text{ess-sup} \sup_{t \leq T} |w_2 \left(t, C_1, C_2\right)| \leq \text{ess-sup} |w_2 \left(0, C_1, C_2\right)| + T \text{ess-sup} \sup_{t \leq T} \left| \frac{\partial}{\partial t} w_2 \left(t, C_1, C_2\right) \right|$$
$$\leq \text{ess-sup} |w_2 \left(0, C_1, C_2\right)| + KT \text{ess-sup} \sup_{t \leq T} |w_3 \left(t, C_2\right)|$$
$$\leq \|\|W\|\|_0 + KTK_{0,3} \left(T\right).$$

$\square$

Inspired by the a priori bounds in Lemma 11, given an arbitrary terminal time $T$ and the initialization $W(0)$, let us consider:

- for a tuple $(a, b) \in \mathbb{R}^2_{\geq 0}$, a space $\mathcal{W}_T(a, b)$ of $W' = (W'(t))_{t \leq T} = (w'_1(t, \cdot), w'_2(t, \cdot, \cdot), w'_3(t, \cdot))_{t \leq T}$ such that

$$\text{ess-sup} \sup_{t \leq T} |w'_3(t, C_2)| \leq b,$$

$$\text{ess-sup} \sup_{t \leq T} |w'_2(t, C_1, C_2)| \leq a,$$

where $w'_1 : \mathbb{R}_{\geq 0} \times \Omega_1 \to \mathbb{R}^d$, $w'_2 : \mathbb{R}_{\geq 0} \times \Omega_1 \times \Omega_2 \mapsto \mathbb{R}$, $w'_3 : \mathbb{R}_{\geq 0} \times \Omega_3 \mapsto \mathbb{R}$,

- for a tuple $(a, b) \in \mathbb{R}^2_{\geq 0}$ and $W(0)$, a space $\mathcal{W}^+_T(a, b, W(0))$ of $W' \in \mathcal{W}_T(a, b)$ such that $W'(0) = W(0)$ additionally (and hence every $W'$ in this space shares the same initialization $W(0)$).

We equip the spaces with the metric $\|W' - W''\|_T$. It is easy to see that both spaces are complete. Note that Lemma 11 implies, under Assumption 1 and $\|\|W\|\|_0 < \infty$, we have any MF solution $W$, if exists, is in $\mathcal{W}_T(\|\|W\|\|_0 + K_{0,2}(T), \|\|W\|\|_0 + K_{0,3}(T))$. For the proof of Theorem 1, we work mainly with $\mathcal{W}^+_T(\|\|W\|\|_0 + K_{0,2}(T), \|\|W\|\|_0 + K_{0,3}(T), W(0))$, although several intermediate lemmas are proven in more generality for other uses.

**Lemma 12.** *Under Assumption 1, for $T \geq 0$, any $W', W'' \in \mathcal{W}_T(a, b)$ and almost every $z \sim \mathcal{P}$:*

$$\text{ess-sup} \sup_{t \leq T} \left| \Delta_2^H(z, C_2; W'(t)) \right| \leq K_{a,b},$$

$$\text{ess-sup} \sup_{t \leq T} |H_2(x, C_2; W'(t)) - H_2(x, C_2; W''(t))| \leq K_{a,b} \|W' - W''\|_T,$$

$$\sup_{t \leq T} |H_3(x; W'(t)) - H_3(x; W''(t))| \leq K_{a,b} \|W' - W''\|_T,$$

$$\sup_{t \leq T} |\partial_2 \mathcal{L}(y, \hat{y}(x; W'(t))) - \partial_2 \mathcal{L}(y, \hat{y}(x; W''(t)))| \leq K_{a,b} \|W' - W''\|_T,$$

$$\text{ess-sup} \sup_{t \leq T} \left| \Delta_2^H(z, C_2; W'(t)) - \Delta_2^H(z, C_2; W''(t)) \right| \leq K_{a,b} \|W' - W''\|_T,$$

*where $K_{a,b} \geq 1$ is a generic constant that grows polynomially with $a$ and $b$.*

*Proof.* The first bound is easy to see:

$$\text{ess-sup} \sup_{t \leq T} \left| \Delta_2^H(z, C_2; W'(t)) \right| \leq \text{ess-sup} \sup_{t \leq T} |w'_3(t, C_2)| \leq b.$$

We prove the second bound, invoking Assumption 1:

$$|H_2(x, C_2; W'(t)) - H_2(x, C_2; W''(t))|$$
$$\leq K |w'_2(t, C_1, C_2)| |\varphi_1(\langle w'_1(t, C_1), x \rangle) - \varphi_1(\langle w''_1(t, C_1), x \rangle)|$$
$$+ K |w'_2(t, C_1, C_2) - w''_2(t, C_1, C_2)|$$
$$\leq K (|w'_2(t, C_1, C_2)| + 1) \|W' - W''\|_T,$$

which yields by the fact $W' \in \mathcal{W}_T(a, b)$:

$$\text{ess-sup} \sup_{t \leq T} |H_2(x, C_2; W'(t)) - H_2(x, C_2; W''(t))| \leq K(a+1) \|W' - W''\|_T.$$

Consequently, we have:

$$|H_3(x; W'(t)) - H_3(x; W''(t))| \leq K |w'_3(t, C_2)| |\varphi_2(H_2(x, C_2; W'(t))) - \varphi_2(H_2(x, C_2; W''(t)))|$$
$$+ K |w'_3(t, C_2) - w''_3(t, C_2)|$$
$$\leq K |w'_3(t, C_2)| |H_2(x, C_2; W'(t)) - H_2(x, C_2; W''(t))|$$
$$+ K \|W' - W''\|_T,$$
$$|\partial_2 \mathcal{L}(y, \hat{y}(x; W'(t))) - \partial_2 \mathcal{L}(y, \hat{y}(x; W''(t)))| \leq K |\hat{y}(x; W'(t)) - \hat{y}(x; W''(t))|$$
$$\leq K |H_3(x; W'(t)) - H_3(x; W''(t))|,$$

which then yield the third and fourth bounds by the fact $W', W'' \in \mathcal{W}_T(a, b)$. Using these bounds, we obtain the last bound:

$$
\begin{aligned}
&\left| \Delta_2^H \left( z, C_2; W'(t) \right) - \Delta_2^H \left( z, C_2; W''(t) \right) \right| \\
&\leq K \left| w_3'(t, C_2) \right| \Big( \left| \partial_2 \mathcal{L} \left( y, \hat{y} \left( x; W'(t) \right) \right) - \partial_2 \mathcal{L} \left( y, \hat{y} \left( x; W''(t) \right) \right) \right| \\
&\qquad + \left| H_3 \left( x; W'(t) \right) - H_3 \left( x; W''(t) \right) \right| + \left| H_2 \left( x, C_2; W'(t) \right) - H_2 \left( x, C_2; W''(t) \right) \right| \Big) \\
&\quad + K \left| w_3'(t, C_2) - w_3''(t, C_2) \right|,
\end{aligned}
$$

from which the last bound follows. $\qquad\square$

To prove Theorem 1, for a given $W(0)$, we define a mapping $F_{W(0)}$ that maps from $W' = (w_1', w_2', w_3') \in \mathcal{W}_T(a, b)$ to $F_{W(0)}(W') = \bar{W}' = (\bar{w}_1', \bar{w}_2', \bar{w}_3')$, defined by $\bar{W}'(0) = W(0)$ and

$$
\begin{aligned}
\frac{\partial}{\partial t} \bar{w}_3'(t, c_2) &= -\xi_3(t) \Delta_3(c_2; W'(t)), \\
\frac{\partial}{\partial t} \bar{w}_2'(t, c_1, c_2) &= -\xi_2(t) \Delta_2(c_1, c_2; W'(t)), \\
\frac{\partial}{\partial t} \bar{w}_1'(t, c_1) &= -\xi_1(t) \Delta_1(c_1; W'(t)).
\end{aligned}
$$

Notice that the right-hand sides do not involve $\bar{W}'$. Note that the MF ODEs' solution, initialized at $W(0)$, is a fixed point of this mapping.

We establish the following estimates for this mapping.

**Lemma 13.** *Under Assumption 1, for $T \geq 0$, any initialization $W(0)$ and any $W', W'' \in \mathcal{W}_T(a, b)$,*

$$
\begin{aligned}
\operatorname*{ess-sup}_{} \sup_{s \leq t} \left| \Delta_3(C_2; W'(s)) - \Delta_3(C_2; W''(s)) \right| &\leq K_{a,b} \left\| W' - W'' \right\|_t, \\
\operatorname*{ess-sup}_{} \sup_{s \leq t} \left| \Delta_2(C_1, C_2; W'(s)) - \Delta_2(C_1, C_2; W''(s)) \right| &\leq K_{a,b} \left\| W' - W'' \right\|_t, \\
\operatorname*{ess-sup}_{} \sup_{s \leq t} \left| \Delta_1(C_1; W'(s)) - \Delta_1(C_1; W''(s)) \right| &\leq K_{a,b} \left\| W' - W'' \right\|_t,
\end{aligned}
$$

*and consequently, if in addition $W'(0) = W''(0)$ (not necessarily equal $W(0)$), then*

$$
\begin{aligned}
\operatorname*{ess-sup}_{} \sup_{t \leq T} \left| \bar{w}_3'(t, C_2) - \bar{w}_3''(t, C_2) \right| &\leq K_{a,b} \int_0^T \left\| W' - W'' \right\|_s ds, \\
\operatorname*{ess-sup}_{} \sup_{t \leq T} \left| \bar{w}_2'(t, C_1, C_2) - \bar{w}_2''(t, C_1, C_2) \right| &\leq K_{a,b} \int_0^T \left\| W' - W'' \right\|_s ds, \\
\operatorname*{ess-sup}_{} \sup_{t \leq T} \left| \bar{w}_1'(t, C_1) - \bar{w}_1''(t, C_1) \right| &\leq K_{a,b} \int_0^T \left\| W' - W'' \right\|_s ds,
\end{aligned}
$$

*in which $\bar{W}' = (\bar{w}_1', \bar{w}_2', \bar{w}_3') = F_{W(0)}(W')$, $\bar{W}'' = (\bar{w}_1'', \bar{w}_2'', \bar{w}_3'') = F_{W(0)}(W'')$ and $K_{a,b} \geq 1$ is a generic constant that grows polynomially with $a$ and $b$.*

*Proof.* From Assumption 1 and the fact $W', W'' \in \mathcal{W}_T(a, b)$, we get:

$$
\begin{aligned}
\left| \Delta_3(C_2; W'(s)) - \Delta_3(C_2; W''(s)) \right| &\leq K \mathbb{E}_Z \left[ \left| \partial_2 \mathcal{L}(Y, \hat{y}(X; W'(s))) - \partial_2 \mathcal{L}(Y, \hat{y}(X; W''(s))) \right| \right] \\
&\quad + K \mathbb{E}_Z \left[ \left| H_3(X; W'(s)) - H_3(X; W''(s)) \right| \right] \\
&\quad + K \mathbb{E}_Z \left[ \left| H_2(X, C_2; W'(s)) - H_2(X, C_2; W''(s)) \right| \right], \\
\left| \Delta_2(C_1, C_2; W'(s)) - \Delta_2(C_1, C_2; W''(s)) \right| &\leq K_{a,b} \left| w_1'(s, C_1) - w_1''(s, C_1) \right| \\
&\quad + K \left| \mathbb{E}_Z \left[ \Delta_2^H(Z, C_2; W'(s)) - \Delta_2^H(Z, C_2; W''(s)) \right] \right|, \\
\left| \Delta_1(C_1; W'(s)) - \Delta_1(C_1; W''(s)) \right| &\leq K_{a,b} \mathbb{E}_Z \left[ \left| \Delta_2^H(Z, C_2; W'(s)) - \Delta_2^H(Z, C_2; W''(s)) \right| \right]
\end{aligned}
$$

$$+ K_{a,b} \left| w'_2 \left( s, C_1, C_2 \right) - w''_2 \left( s, C_1, C_2 \right) \right|$$
$$+ K_{a,b} \left| w'_1 \left( s, C_1 \right) - w''_1 \left( s, C_1 \right) \right|,$$

from which the first three estimates then follow, in light of Lemma 12. The last three estimates then follow from the fact that $\bar{W}'(0) = \bar{W}''(0)$ and Assumption 1; for instance,

$$\underset{t \leq T}{\text{ess-sup} \sup} \left| \bar{w}'_3 \left( t, C_2 \right) - \bar{w}''_3 \left( t, C_2 \right) \right| \leq \int_0^T \text{ess-sup} \left| \frac{\partial}{\partial t} \bar{w}'_3 \left( s, C_2 \right) - \frac{\partial}{\partial t} \bar{w}''_3 \left( s, C_2 \right) \right| ds$$

$$\leq K \int_0^T \text{ess-sup} \left| \Delta_3 \left( C_2; W'(s) \right) - \Delta_3 \left( C_2; W''(s) \right) \right| ds.$$

$\square$

We are now ready to prove Theorem 1.

*Proof of Theorem 1.* We will use a Picard-type iteration. To lighten notations:

$$\mathcal{W}_T^+ \equiv \mathcal{W}_T^+ \left( \|\|W\|\|_0 + K_{0,2}(T), \|\|W\|\|_0 + K_{0,3}(T), W(0) \right), \qquad F \equiv F_{W(0)}.$$

Since $\|\|W\|\|_0 \leq K$ by assumption, we have $\|\|W\|\|_0 + K_{0,2}(T) + K_{0,3}(T) \leq K_T$. Recall that $\mathcal{W}_T^+$ is complete. For an arbitrary $T > 0$, consider $W', W'' \in \mathcal{W}_T^+$. Lemma 13 yields:

$$\left\| F(W') - F(W'') \right\|_T \leq K_T \int_0^T \left\| W' - W'' \right\|_s ds.$$

Note that $F$ maps to $\mathcal{W}_T^+$ under Assumption 1 by the same argument as Lemma 11. Hence we are allowed to iterating this inequality and get, for an arbitrary $T > 0$,

$$\left\| F^{(k)}(W') - F^{(k)}(W'') \right\|_T \leq K_T \int_0^T \left\| F^{(k-1)}(W') - F^{(k-1)}(W'') \right\|_{T_2} dT_2$$

$$\leq K_T^2 \int_0^T \int_0^{T_2} \left\| F^{(k-2)}(W') - F^{(k-2)}(W'') \right\|_{T_3} \mathbb{I}(T_2 \leq T) \, dT_3 dT_2$$

$$\dots$$

$$\leq K_T^k \int_0^T \int_0^{T_2} \dots \int_0^{T_k} \left\| W' - W'' \right\|_{T_{k+1}} \mathbb{I}(T_k \leq \dots \leq T_2 \leq T) \, dT_{k+1}...dT_2$$

$$\leq \frac{1}{k!} K_T^k \left\| W' - W'' \right\|_T.$$

By substituting $W'' = F(W')$, we have:

$$\sum_{k=1}^\infty \left\| F^{(k+1)}(W') - F^{(k)}(W') \right\|_T = \sum_{k=1}^\infty \left\| F^{(k)}(W'') - F^{(k)}(W') \right\|_T$$

$$\leq \sum_{k=1}^\infty \frac{1}{k!} K_T^k \left\| W' - W'' \right\|_T$$

$$< \infty.$$

Hence as $k \to \infty$, $F^{(k)}(W')$ converges to a limit in $\mathcal{W}_T^+$, which is a fixed point of $F$. The uniqueness of a fixed point follows from the above estimate, since if $W'$ and $W''$ are fixed points then

$$\left\| W' - W'' \right\|_T = \left\| F^{(k)}(W') - F^{(k)}(W'') \right\|_T \leq \frac{1}{k!} K_T^k \left\| W' - W'' \right\|_T,$$

while one can take $k$ arbitrarily large. This proves that the solution exists and is unique on $t \in [0, T]$. Since $T$ is arbitrary, we have existence and uniqueness of the solution on the time interval $[0, \infty)$. $\square$

## C  CONNECTION BETWEEN THE NEURAL NET AND ITS MF LIMIT: PROOFS FOR SECTION 3

### C.1  PROOF OF THEOREM 3

We construct an auxiliary trajectory, which we call the *particle ODEs*:

$$\frac{\partial}{\partial t}\tilde{w}_3\left(t, j_2\right) = -\xi_3\left(t\right)\mathbb{E}_Z\left[\partial_2\mathcal{L}\left(Y, \hat{\mathbf{y}}\left(X; \tilde{W}\left(t\right)\right)\right)\varphi_3'\left(\mathbf{H}_3\left(X; \tilde{W}\left(t\right)\right)\right)\varphi_2\left(\mathbf{H}_2\left(X, j_2; \tilde{W}\left(t\right)\right)\right)\right],$$

$$\frac{\partial}{\partial t}\tilde{w}_2\left(t, j_1, j_2\right) = -\xi_2\left(t\right)\mathbb{E}_Z\left[\Delta_2^{\mathbf{H}}\left(Z, j_2; \tilde{W}\left(t\right)\right)\varphi_1\left(\langle\tilde{w}_1\left(t, j_1\right), X\rangle\right)\right],$$

$$\frac{\partial}{\partial t}\tilde{w}_1\left(t, j_1\right) = -\xi_1\left(t\right)\mathbb{E}_Z\left[\frac{1}{n_2}\sum_{j_2=1}^{n_2}\Delta_2^{\mathbf{H}}\left(Z, j_2; \tilde{W}\left(t\right)\right)\tilde{w}_2\left(t, j_1, j_2\right)\varphi_1'\left(\langle\tilde{w}_1\left(t, j_1\right), X\rangle\right)X\right],$$

in which $j_1 = 1, ..., n_1$, $j_2 = 1, ..., n_2$, $\tilde{W}\left(t\right) = \left(\tilde{w}_1\left(t, \cdot\right), \tilde{w}_2\left(t, \cdot, \cdot\right), \tilde{w}_3\left(t, \cdot\right)\right)$, and $t \in \mathbb{R}_{\geq 0}$. We specify the initialization $\tilde{W}\left(0\right)$: $\tilde{w}_1\left(0, j_1\right) = w_1^0\left(C_1\left(j_1\right)\right)$, $\tilde{w}_2\left(0, j_1, j_2\right) = w_2^0\left(C_1\left(j_1\right), C_2\left(j_2\right)\right)$ and $\tilde{w}_3\left(0, j_3\right) = w_3^0\left(C_2\left(j_2\right)\right)$. That is, it shares the same initialization with the neural network one $\mathbf{W}\left(0\right)$, and hence is coupled with the neural network and the MF ODEs. Roughly speaking, the particle ODEs are continuous-time trajectories of finitely many neurons, averaged over the data distribution. We note that $\tilde{W}\left(t\right)$ is random for all $t \in \mathbb{R}_{\geq 0}$ due to the randomness of $\{C_i\left(j_i\right)\}_{i=1,2}$.

The existence and uniqueness of the solution to the particle ODEs follows from the same proof as in Theorem 1, which we shall not repeat here. We equip $\tilde{W}\left(t\right)$ with the norm

$$\|\|\tilde{W}\|\|_T = \max\left\{\max_{j_1\leq n_1,\, j_2\leq n_2}\sup_{t\leq T}\left|\tilde{w}_2\left(t, j_1, j_2\right)\right|,\ \max_{j_2\leq n_2}\sup_{t\leq T}\left|\tilde{w}_3\left(t, j_2\right)\right|\right\}.$$

One can also define the measures $\mathscr{D}_T\left(W, \tilde{W}\right)$ and $\mathscr{D}_T\left(\tilde{W}, \mathbf{W}\right)$ similar to Eq. (2):

$$\mathscr{D}_T\left(W, \tilde{W}\right) = \sup\left\{\left|w_1\left(t, C_1\left(j_1\right)\right) - \tilde{w}_1\left(t, C_1\left(j_1\right)\right)\right|,\ \left|w_2\left(t, C_1\left(j_1\right), C_2\left(j_2\right)\right) - \tilde{w}_2\left(t, C_1\left(j_1\right), C_2\left(j_2\right)\right)\right|,\right.$$
$$\left.\left|w_3\left(t, C_2\left(j_2\right)\right) - \tilde{w}_3\left(t, C_2\left(j_2\right)\right)\right| :\ t \leq T,\ j_1 \leq n_1,\ j_2 \leq n_2\right\},$$

$$\mathscr{D}_T\left(\tilde{W}, \mathbf{W}\right) = \sup\left\{\left|\mathbf{w}_1\left(\lfloor t/\epsilon\rfloor, j_1\right) - \tilde{w}_1\left(t, C_1\left(j_1\right)\right)\right|,\ \left|\mathbf{w}_2\left(\lfloor t/\epsilon\rfloor, j_1, j_2\right) - \tilde{w}_2\left(t, C_1\left(j_1\right), C_2\left(j_2\right)\right)\right|,\right.$$
$$\left.\left|\mathbf{w}_3\left(\lfloor t/\epsilon\rfloor, j_2\right) - \tilde{w}_3\left(t, C_2\left(j_2\right)\right)\right| :\ t \leq T,\ j_1 \leq n_1,\ j_2 \leq n_2\right\}.$$

We have the following results:

**Theorem 14.** *Under the same setting as Theorem 3, for any $\delta > 0$, with probability at least $1 - \delta$,*

$$\mathscr{D}_T\left(W, \tilde{W}\right) \leq \frac{1}{\sqrt{n_{\min}}}\log^{1/2}\left(\frac{3\left(T + 1\right)n_{\max}^2}{\delta} + e\right)e^{K_T},$$

*in which $n_{\min} = \min\{n_1, n_2\}$, $n_{\max} = \max\{n_1, n_2\}$, and $K_T = K\left(1 + T^K\right)$.*

**Theorem 15.** *Under the same setting as Theorem 3, for any $\delta > 0$ and $\epsilon \leq 1$, with probability at least $1 - \delta$,*

$$\mathscr{D}_T\left(\tilde{W}, \mathbf{W}\right) \leq \sqrt{\epsilon\log\left(\frac{2n_1 n_2}{\delta} + e\right)}e^{K_T},$$

*in which $K_T = K\left(1 + T^K\right)$.*

*Proof of Theorem 3.* Using the fact

$$\mathscr{D}_T\left(W, \mathbf{W}\right) \leq \mathscr{D}_T\left(W, \tilde{W}\right) + \mathscr{D}_T\left(\tilde{W}, \mathbf{W}\right),$$

the thesis is immediate from Theorems 14 and 15. □

### C.2 PROOF OF THEOREMS 14 AND 15

*Proof of Theorem 14.* In the following, let $K_t$ denote an generic positive constant that may change from line to line and takes the form

$$K_t = K \left( 1 + t^K \right),$$

such that $K_t \geq 1$ and $K_t \leq K_T$ for all $t \leq T$. We first note that at initialization, $\mathscr{D}_0 \left( W, \tilde{W} \right) = 0$. Since $\|\|W\|\|_0 \leq K$, $\|\|W\|\|_T \leq K_T$ by Lemma 11. Furthermore it is easy to see that $\|\|\tilde{W}\|\|_0 \leq \|\|W\|\|_0 \leq K$ almost surely. By the same argument as in Lemma 11, $\|\|\tilde{W}\|\|_T \leq K_T$ almost surely.

We shall use all above bounds repeatedly in the proof. We decompose the proof into several steps.

**Step 1 - Main proof.** Let us define, for brevity

$$q_3 \left( t, x \right) = \mathbf{H}_3 \left( x; \tilde{W} \left( t \right) \right) - H_3 \left( x; W \left( t \right) \right),$$

$$q_2 \left( t, x, j_2, c_2 \right) = \mathbf{H}_2 \left( x, j_2; \tilde{W} \left( t \right) \right) - H_2 \left( x, c_2; W \left( t \right) \right),$$

$$q_\Delta \left( t, z, j_1, j_2, c_1, c_2 \right) = \Delta_2^{\mathbf{H}} \left( Z, j_2; \tilde{W} \left( t \right) \right) \tilde{w}_2 \left( t, j_1, j_2 \right) - \Delta_2^H \left( z, c_2; W \left( t \right) \right) w_2 \left( t, c_1, c_2 \right).$$

Consider $t \geq 0$. We first bound the difference in the updates between $W$ and $\tilde{W}$. Let us start with $w_3$ and $\tilde{w}_3$. By Assumption 1, we have:

$$\left| \frac{\partial}{\partial t} \tilde{w}_3 \left( t, j_2 \right) - \frac{\partial}{\partial t} w_3 \left( t, C_2 \left( j_2 \right) \right) \right| \leq K \mathbb{E}_Z \left[ \left| q_3 \left( t, X \right) \right| + \left| q_2 \left( t, X, j_2, C_2 \left( j_2 \right) \right) \right| \right].$$

Similarly, for $w_2$ and $\tilde{w}_2$,

$$\left| \frac{\partial}{\partial t} \tilde{w}_2 \left( t, j_1, j_2 \right) - \frac{\partial}{\partial t} w_2 \left( t, C_1 \left( j_1 \right), C_2 \left( j_2 \right) \right) \right|$$
$$\leq K \mathbb{E}_Z \left[ \left| \Delta_2^{\mathbf{H}} \left( Z, j_2; \tilde{W} \left( t \right) \right) - \Delta_2^H \left( Z, C_2 \left( j_2 \right); W \left( t \right) \right) \right| \right]$$
$$+ K \left| w_3 \left( t, C_2 \left( j_2 \right) \right) \right| \left| \tilde{w}_1 \left( t, j_1 \right) - w_1 \left( t, C_1 \left( j_1 \right) \right) \right|$$
$$\leq K_t \mathbb{E}_Z \left[ \left| q_3 \left( t, X \right) \right| + \left| q_2 \left( t, X, j_2, C_2 \left( j_2 \right) \right) \right| \right]$$
$$+ K_t \left( \left| \tilde{w}_1 \left( t, j_1 \right) - w_1 \left( t, C_1 \left( j_1 \right) \right) \right| + \left| \tilde{w}_3 \left( t, j_2 \right) - w_3 \left( t, C_2 \left( j_2 \right) \right) \right| \right)$$
$$\leq K_t \mathbb{E}_Z \left[ \left| q_3 \left( t, X \right) \right| + \left| q_2 \left( t, X, j_2, C_2 \left( j_2 \right) \right) \right| \right] + K_t \mathscr{D}_t \left( W, \tilde{W} \right),$$

and for $w_1$ and $\tilde{w}_1$, by Lemma 12,

$$\left| \frac{\partial}{\partial t} \tilde{w}_1 \left( t, j_1 \right) - \frac{\partial}{\partial t} w_1 \left( t, C_1 \left( j_1 \right) \right) \right|$$
$$\leq K \mathbb{E}_Z \left[ \left| \frac{1}{n_2} \sum_{j_2=1}^{n_2} \mathbb{E}_{C_2} \left[ q_\Delta \left( t, Z, j_1, j_2, C_1 \left( j_1 \right), C_2 \right) \right] \right| \right]$$
$$+ \mathbb{E}_{C_2} \left[ \left| \Delta_2^H \left( Z, C_2; W \left( t \right) \right) \right| \left| w_2 \left( t, C_1 \left( j_1 \right), C_2 \right) \right| \right] \left| \tilde{w}_1 \left( t, j_1 \right) - w_1 \left( t, C_1 \left( j_1 \right) \right) \right|$$
$$\leq K \mathbb{E}_Z \left[ \left| \frac{1}{n_2} \sum_{j_2=1}^{n_2} \mathbb{E}_{C_2} \left[ q_\Delta \left( t, Z, j_1, j_2, C_1 \left( j_1 \right), C_2 \right) \right] \right| \right]$$
$$+ K_t \mathscr{D}_t \left( W, \tilde{W} \right).$$

To further the bounding, we now make the following two claims:

- **Claim 1:** For any $\xi > 0$,

$$\max_{j_2 \leq n_2} \left| \frac{\partial}{\partial t} w_3 \left( t + \xi, C_2 \left( j_2 \right) \right) - \frac{\partial}{\partial t} w_3 \left( t, C_2 \left( j_2 \right) \right) \right| \leq K_{t+\xi} \xi,$$

$$\max_{j_1 \leq n_1, \, j_2 \leq n_2} \left| \frac{\partial}{\partial t} w_2 \left(t + \xi, C_1 \left(j_1\right), C_2 \left(j_2\right)\right) - \frac{\partial}{\partial t} w_2 \left(t, C_1 \left(j_1\right), C_2 \left(j_2\right)\right) \right| \leq K_{t+\xi} \xi,$$

$$\max_{j_1 \leq n_1} \left| \frac{\partial}{\partial t} w_1 \left(t + \xi, C_1 \left(j_1\right)\right) - \frac{\partial}{\partial t} w_1 \left(t, C_1 \left(j_1\right)\right) \right| \leq K_{t+\xi} \xi,$$

and similarly,

$$\max_{j_2 \leq n_2} \left| \frac{\partial}{\partial t} \tilde{w}_3 \left(t + \xi, j_2\right) - \frac{\partial}{\partial t} \tilde{w}_3 \left(t, j_2\right) \right| \leq K_{t+\xi} \xi,$$

$$\max_{j_1 \leq n_1, \, j_2 \leq n_2} \left| \frac{\partial}{\partial t} \tilde{w}_2 \left(t + \xi, j_1, j_2\right) - \frac{\partial}{\partial t} \tilde{w}_2 \left(t, j_1, j_2\right) \right| \leq K_{t+\xi} \xi,$$

$$\max_{j_1 \leq n_1} \left| \frac{\partial}{\partial t} \tilde{w}_1 \left(t + \xi, j_1\right) - \frac{\partial}{\partial t} \tilde{w}_1 \left(t, j_1\right) \right| \leq K_{t+\xi} \xi.$$

- **Claim 2:** For any $\gamma_1, \gamma_2, \gamma_3 > 0$ and $t \geq 0$,

$$\max \left\{ \max_{j_2 \leq n_2} \mathbb{E}_Z \left[ \left| q_2 \left(t, X, j_2, C_2 \left(j_2\right)\right) \right| \right], \quad \mathbb{E}_Z \left[ \left| q_3 \left(t, X\right) \right| \right], \right.$$

$$\left. \max_{j_1 \leq n_1} \mathbb{E}_Z \left[ \left| \frac{1}{n_2} \sum_{j_2=1}^{n_2} \mathbb{E}_{C_2} \left[ q_\Delta \left(t, Z, j_1, j_2, C_1 \left(j_1\right), C_2\right) \right] \right| \right] \right\}$$

$$\geq K_t \left( \mathscr{D}_t \left(W, \tilde{W}\right) + \gamma_1 + \gamma_2 + \gamma_3 \right),$$

with probability at most

$$\frac{n_1}{\gamma_1} \exp \left( -\frac{n_2 \gamma_1^2}{K_t} \right) + \frac{n_2}{\gamma_2} \exp \left( -\frac{n_1 \gamma_2^2}{K_t} \right) + \frac{1}{\gamma_3} \exp \left( -\frac{n_2 \gamma_3^2}{K_t} \right).$$

Combining these claims with the previous bounds, taking a union bound over $t \in \{0, \xi, 2\xi, ..., \lfloor T/\xi \rfloor \xi\}$ for some $\xi \in (0, 1)$, we obtain that

$$\max \left\{ \max_{j_2 \leq n_2} \left| \frac{\partial}{\partial t} \tilde{w}_3 \left(t, j_2\right) - \frac{\partial}{\partial t} w_3 \left(t, C_2 \left(j_2\right)\right) \right|, \right.$$

$$\max_{j_1 \leq n_1, \, j_2 \leq n_2} \left| \frac{\partial}{\partial t} \tilde{w}_2 \left(t, j_1, j_2\right) - \frac{\partial}{\partial t} w_2 \left(t, C_1 \left(j_1\right), C_2 \left(j_2\right)\right) \right|,$$

$$\left. \max_{j_1 \leq n_1} \left| \frac{\partial}{\partial t} \tilde{w}_1 \left(t, j_1\right) - \frac{\partial}{\partial t} w_1 \left(t, C_1 \left(j_1\right)\right) \right| \right\}$$

$$\leq K_T \left( \mathscr{D}_t \left(W, \tilde{W}\right) + \gamma_1 + \gamma_2 + \gamma_3 + \xi \right), \qquad \forall t \in [0, T],$$

with probability at least

$$1 - \frac{T+1}{\xi} \left[ \frac{n_1}{\gamma_1} \exp \left( -\frac{n_2 \gamma_1^2}{K_T} \right) + \frac{n_2}{\gamma_2} \exp \left( -\frac{n_1 \gamma_2^2}{K_T} \right) + \frac{1}{\gamma_3} \exp \left( -\frac{n_2 \gamma_3^2}{K_T} \right) \right].$$

The above event in turn implies

$$\mathscr{D}_t \left(W, \tilde{W}\right) \leq K_T \int_0^t \left( \mathscr{D}_s \left(W, \tilde{W}\right) + \gamma_1 + \gamma_2 + \gamma_3 + \xi \right) ds,$$

and hence by Gronwall's lemma and the fact $\mathscr{D}_0 \left(W, \tilde{W}\right) = 0$, we get

$$\mathscr{D}_T \left(W, \tilde{W}\right) \leq \left( \gamma_1 + \gamma_2 + \gamma_3 + \xi \right) e^{K_T}.$$

The theorem then follows from the choice

$$\xi = \frac{1}{\sqrt{n_{\max}}}, \quad \gamma_2 = \frac{K_T}{\sqrt{n_1}} \log^{1/2} \left( \frac{3 \left(T+1\right) n_{\max}^2}{\delta} + e \right), \quad \gamma_1 = \gamma_3 = \frac{K_T}{\sqrt{n_2}} \log^{1/2} \left( \frac{3 \left(T+1\right) n_{\max}^2}{\delta} + e \right).$$

We are left with proving the claims.

**Step 2 - Proof of Claim 1.** We have from Assumption 1,

$$\text{ess-sup}\,|w_3\,(t+\xi,C_2) - w_3\,(t,C_2)| \leq K \int_t^{t+\xi} \text{ess-sup}\left|\frac{\partial}{\partial t}w_3\,(s,C_2)\right| ds$$
$$\leq K\xi,$$

$$\text{ess-sup}\,|w_2\,(t+\xi,C_1,C_2) - w_2\,(t,C_1,C_2)| \leq K \int_t^{t+\xi} \text{ess-sup}\left|\frac{\partial}{\partial t}w_2\,(s,C_1,C_2)\right| ds$$
$$\leq K \int_t^{t+\xi} \text{ess-sup}\,|w_3\,(s,C_2)|\, ds$$
$$\leq K_{t+\xi}\xi,$$

$$\text{ess-sup}\,|w_1\,(t+\xi,C_1) - w_1\,(t,C_1)| \leq K \int_t^{t+\xi} \text{ess-sup}\left|\frac{\partial}{\partial t}w_1\,(s,C_1)\right| ds$$
$$\leq K \int_t^{t+\xi} \text{ess-sup}\,|w_3\,(s,C_2)\,w_2\,(s,C_1,C_2)|\, ds$$
$$\leq K_{t+\xi}\xi.$$

By Lemma 12, we then obtain that

$$\text{ess-sup}\mathbb{E}_Z\,[|H_2\,(X,C_2;W\,(t+\xi)) - H_2\,(X,C_2;W\,(t))|] \leq K_{t+\xi}\xi,$$
$$\mathbb{E}_Z\,[|H_3\,(X;W\,(t+\xi)) - H_3\,(X;W\,(t))|] \leq K_{t+\xi}\xi,$$
$$\text{ess-sup}\mathbb{E}_Z\,\left[\left|\Delta_2^H\,(Z,C_2;W\,(t+\xi)) - \Delta_2^H\,(Z,C_2;W\,(t))\right|\right] \leq K_{t+\xi}\xi.$$

Using these estimates, we thus have, by Assumption 1,

$$\max_{j_2 \leq n_2}\left|\frac{\partial}{\partial t}w_3\,(t+\xi,C_2\,(j_2)) - \frac{\partial}{\partial t}w_3\,(t,C_2\,(j_2))\right|$$
$$\leq K_{t+\xi}\xi + K\mathbb{E}_Z\,[|H_3\,(X;W\,(t+\xi)) - H_3\,(X;W\,(t))|]$$
$$\quad + K\text{ess-sup}\mathbb{E}_Z\,[|H_2\,(X,C_2;W\,(t+\xi)) - H_2\,(X,C_2;W\,(t))|]$$
$$\leq K_{t+\xi}\xi,$$
$$\max_{j_1 \leq n_1,\, j_2 \leq n_2}\left|\frac{\partial}{\partial t}w_2\,(t+\xi,C_1\,(j_1),C_2\,(j_2)) - \frac{\partial}{\partial t}w_2\,(t,C_1\,(j_1),C_2\,(j_2))\right|$$
$$\leq K_{t+\xi}\xi + K\text{ess-sup}\mathbb{E}_Z\,\left[\left|\Delta_2^H\,(Z,C_2;W\,(t+\xi)) - \Delta_2^H\,(Z,C_2;W\,(t))\right|\right]$$
$$\quad + K\text{ess-sup}\,|w_3\,(t,C_2)|\,|w_1\,(t+\xi,C_1) - w_1\,(t,C_1)|$$
$$\leq K_{t+\xi}\xi,$$
$$\max_{j_1 \leq n_1}\left|\frac{\partial}{\partial t}w_1\,(t+\xi,C_1\,(j_1)) - \frac{\partial}{\partial t}w_1\,(t,C_1\,(j_1))\right|$$
$$\leq K_{t+\xi}\xi + K\text{ess-sup}\mathbb{E}_Z\,\left[\mathbb{E}_{C_2}\,\left[\left|\Delta_2^H\,(Z,C_2;W\,(t+\xi)) - \Delta_2^H\,(Z,C_2;W\,(t))\right|\,|w_2\,(t,C_1,C_2)|\right]\right]$$
$$\quad + K\text{ess-sup}\mathbb{E}_{C_2}\,[|w_3\,(t,C_2)|\,|w_2\,(t+\xi,C_1,C_2) - w_2\,(t,C_1,C_2)|]$$
$$\quad + K\text{ess-sup}\mathbb{E}_{C_2}\,[|w_3\,(t,C_2)\,w_2\,(t,C_1,C_2)|]\,|w_1\,(t+\xi,C_1) - w_1\,(t,C_1)|$$
$$\leq K_{t+\xi}\xi.$$

The proof of the rest of the claim is similar.

**Step 3 - Proof of Claim 2.** We recall the definitions of $q_\Delta$, $q_2$ and $q_3$. Let us decompose them as follows. We start with $q_2$:

$$|q_2\,(t,x,j_2,C_2\,(j_2))|$$
$$= \left|\frac{1}{n_1}\sum_{j_1=1}^{n_1}\tilde{w}_2\,(t,j_1,j_2)\,\varphi_1\,(\langle\tilde{w}_1\,(t,j_1),x\rangle) - \mathbb{E}_{C_1}\,[w_2\,(t,C_1,C_2\,(j_2))\,\varphi_1\,(\langle w_1\,(t,C_1),x\rangle)]\right|$$
$$\leq \max_{j_1 \leq n_1}|\tilde{w}_2\,(t,j_1,j_2)\,\varphi_1\,(\langle\tilde{w}_1\,(t,j_1),x\rangle) - w_2\,(t,C_1\,(j_1),C_2\,(j_2))\,\varphi_1\,(\langle w_1\,(t,C_1\,(j_1)),x\rangle)|$$

$$+ \left| \frac{1}{n_1} \sum_{j_1=1}^{n_1} w_2\left(t, C_1\left(j_1\right), C_2\left(j_2\right)\right) \varphi_1\left(\left\langle w_1\left(t, C_1\left(j_1\right)\right), x\right\rangle\right) - \mathbb{E}_{C_1}\left[w_2\left(t, C_1, C_2\left(j_2\right)\right) \varphi_1\left(\left\langle w_1\left(t, C_1\right), x\right\rangle\right)\right] \right|$$

$$\equiv Q_{2,1}\left(x, j_2\right) + Q_{2,2}\left(x, j_2\right).$$

Similarly, we have for $q_3$:

$$\left| q_3\left(t, x\right) \right|$$

$$= \left| \frac{1}{n_2} \sum_{j_2=1}^{n_2} \tilde{w}_3\left(t, j_2\right) \varphi_2\left(\mathbf{H}_2\left(x, j_2; \tilde{W}\left(t\right)\right)\right) - \mathbb{E}_{C_2}\left[w_3\left(t, C_2\right) \varphi_2\left(H_2\left(x, C_2; W\left(t\right)\right)\right)\right] \right|$$

$$\leq \max_{j_2 \leq n_2} \left| \tilde{w}_3\left(t, j_2\right) \varphi_2\left(\mathbf{H}_2\left(x, j_2; \tilde{W}\left(t\right)\right)\right) - w_3\left(t, C_2\left(j_2\right)\right) \varphi_2\left(H_2\left(x, C_2\left(j_2\right); W\left(t\right)\right)\right) \right|$$

$$+ \left| \frac{1}{n_2} \sum_{j_2=1}^{n_2} w_3\left(t, C_2\left(j_2\right)\right) \varphi_2\left(H_2\left(x, C_2\left(j_2\right); W\left(t\right)\right)\right) - \mathbb{E}_{C_2}\left[w_3\left(t, C_2\right) \varphi_2\left(H_2\left(x, C_2; W\left(t\right)\right)\right)\right] \right|$$

$$\equiv Q_{3,1}\left(x\right) + Q_{3,2}\left(x\right).$$

Finally we have for $q_\Delta$:

$$\left| \frac{1}{n_2} \sum_{j_2=1}^{n_2} \mathbb{E}_{C_2}\left[q_\Delta\left(t, z, j_1, j_2, C_1\left(j_1\right), C_2\right)\right] \right|$$

$$\leq \max_{j_2 \leq n_2} \left| \Delta_2^{\mathbf{H}}\left(z, j_2; \tilde{W}\left(t\right)\right) \tilde{w}_2\left(t, j_1, j_2\right) - \Delta_2^H\left(z, C_2\left(j_2\right); W\left(t\right)\right) w_2\left(t, C_1\left(j_1\right), C_2\left(j_2\right)\right) \right|$$

$$+ \left| \frac{1}{n_2} \sum_{j_2=1}^{n_2} \Delta_2^H\left(z, C_2\left(j_2\right); W\left(t\right)\right) w_2\left(t, C_1\left(j_1\right), C_2\left(j_2\right)\right) - \mathbb{E}_{C_2}\left[\Delta_2^H\left(z, C_2; W\left(t\right)\right) w_2\left(t, C_1\left(j_1\right), C_2\right)\right] \right|$$

$$\equiv Q_{1,1}\left(z, j_1\right) + Q_{1,2}\left(z, j_1\right).$$

Now let us analyze each of the terms.

- We start with $Q_{2,1}$. We have from Assumption 1,

$$\max_{j_2 \leq n_2} \mathbb{E}_Z\left[Q_{2,1}\left(X, j_2\right)\right]$$
$$\leq K \max_{j_1 \leq n_1,\, j_2 \leq n_2} \left| \tilde{w}_2\left(t, j_1, j_2\right) - w_2\left(t, C_1\left(j_1\right), C_2\left(j_2\right)\right) \right|$$
$$+ K \max_{j_1 \leq n_1,\, j_2 \leq n_2} \left| w_2\left(t, C_1\left(j_1\right), C_2\left(j_2\right)\right) \right| \left| \tilde{w}_1\left(t, j_1\right) - w_1\left(t, C_1\left(j_1\right)\right) \right|$$
$$\leq K_t \mathscr{D}_t\left(W, \tilde{W}\right).$$

- To bound $Q_{2,2}$, let us write:

$$Z_2\left(x, c_1, c_2\right) = w_2\left(t, c_1, c_2\right) \varphi_1\left(\left\langle w_1\left(t, c_1\right), x\right\rangle\right).$$

Recall that $C_1\left(j_1\right)$ and $C_2\left(j_2\right)$ are independent. We thus have:

$$\mathbb{E}\left[Z_2\left(X, C_1\left(j_1\right), C_2\left(j_2\right)\right) \mid X, C_2\left(j_2\right)\right] = \mathbb{E}_{C_1}\left[Z_2\left(X, C_1, C_2\left(j_2\right)\right)\right].$$

Furthermore $\left\{Z_2\left(C_1\left(j_1\right), C_2\left(j_2\right)\right)\right\}_{j_1 \in [n_1]}$ are independent, conditional on $C_2\left(j_2\right)$. We also have, almost surely

$$\left| Z_2\left(X, C_1\left(j_1\right), C_2\left(j_2\right)\right) \right| \leq K_t,$$

by Assumption 1. Then by Lemma 19,

$$\mathbb{P}\left(\mathbb{E}_Z\left[Q_{2,2}\left(X, j_2\right)\right] \geq K_t \gamma_2\right) \leq \left(1/\gamma_2\right) \exp\left(-n_1 \gamma_2^2 / K_t\right).$$

- To bound $Q_{3,1}$, we have from Assumption 1,

$$\mathbb{E}_Z\left[Q_{3,1}\left(X\right)\right] \leq \max_{j_2 \leq n_2}\left(K\left|\tilde{w}_3\left(t, j_2\right) - w_3\left(t, C_2\left(j_2\right)\right)\right| + K_t\mathbb{E}_Z\left[\left|q_2\left(t, X, j_2, C_2\left(j_2\right)\right)\right|\right]\right)$$

$$\leq K\mathscr{D}_t\left(W, \tilde{W}\right) + K_t \max_{j_2 \leq n_2}\mathbb{E}_Z\left[\left|q_2\left(t, X, j_2, C_2\left(j_2\right)\right)\right|\right].$$

- To bound $Q_{3,2}$, noticing that almost surely

$$\left|w_3\left(t, C_2\left(j_2\right)\right)\varphi_2\left(H_2\left(x, C_2\left(j_2\right); W\left(t\right)\right)\right)\right| \leq K_t$$

by Assumption 1, we obtain

$$\mathbb{P}\left(\mathbb{E}_Z\left[Q_{3,2}\left(X\right)\right] \geq K_t\gamma_3\right) \leq \left(1/\gamma_3\right)\exp\left(-n_2\gamma_3^2/K_t\right),$$

similar to the treatment of $Q_{2,2}$.

- To bound $Q_{1,1}$, using Assumption 1,

$$\mathbb{E}_Z\left[Q_{1,1}\left(Z, j_1\right)\right] \leq K \max_{j_2 \leq n_2}\left|w_2\left(t, C_1\left(j_1\right), C_2\left(j_2\right)\right)\right|\mathbb{E}_Z\left[\left|\Delta_2^{\mathbf{H}}\left(Z, j_2; \tilde{W}\left(t\right)\right) - \Delta_2^H\left(Z, C_2\left(j_2\right); W\left(t\right)\right)\right|\right]$$

$$+ K \max_{j_2 \leq n_2}\left|\tilde{w}_2\left(t, j_1, j_2\right) - w_2\left(t, C_1\left(j_1\right), C_2\left(j_2\right)\right)\right|\mathbb{E}_Z\left[\left|\Delta_2^{\mathbf{H}}\left(Z, j_2; \tilde{W}\left(t\right)\right)\right|\right]$$

$$\leq K \max_{j_2 \leq n_2}\left|w_2\left(t, C_1\left(j_1\right), C_2\left(j_2\right)\right)\right|\left(\left|\tilde{w}_3\left(t, j_2\right) - w_3\left(t, C_2\left(j_2\right)\right)\right|\right.$$

$$\left. + \left|w_3\left(t, C_2\left(j_2\right)\right)\right|\mathbb{E}_Z\left[\left|q_3\left(t, X\right)\right| + \left|q_2\left(t, X, j_2, C_2\left(j_2\right)\right)\right|\right]\right)$$

$$+ K \max_{j_2 \leq n_2}\left|\tilde{w}_2\left(t, j_1, j_2\right) - w_2\left(t, C_1\left(j_1\right), C_2\left(j_2\right)\right)\right|\left|\tilde{w}_3\left(t, j_2\right)\right|$$

$$\leq K_t\left(\mathscr{D}_t\left(W, \tilde{W}\right) + \mathbb{E}_Z\left[\left|q_3\left(t, X\right)\right| + \max_{j_2 \leq n_2}\left|q_2\left(t, X, j_2, C_2\left(j_2\right)\right)\right|\right]\right).$$

- To bound $Q_{1,2}$, we note that almost surely

$$\left|\Delta_2^H\left(Z, C_2\left(j_2\right); W\left(t\right)\right)w_2\left(t, C_1\left(j_1\right), C_2\left(j_2\right)\right)\right| \leq K\left|w_3\left(t, C_2\left(j_2\right)\right)\right|\left|w_2\left(t, C_1\left(j_1\right), C_2\left(j_2\right)\right)\right|$$

$$\leq K_t.$$

Then similar to the bounding of $Q_{2,2}$, we get:

$$\mathbb{P}\left(\mathbb{E}_Z\left[Q_{1,2}\left(Z, j_1\right)\right] \geq K_t\gamma_1\right) \leq \left(1/\gamma_1\right)\exp\left(-n_2\gamma_1^2/K_t\right).$$

Finally, combining all of these bounds together, applying suitably the union bound over $j_1 \in [n_1]$ and $j_2 \in [n_2]$, we obtain the claim. □

*Proof of Theorem 15.* We consider $t \leq T$, for a given terminal time $T \in \epsilon\mathbb{N}_{\geq 0}$. We again reuse the notation $K_t$ from the proof of Theorem 3. Note that $K_t \leq K_T$ for all $t \leq \bar{T}$. We also note that at initialization, $\mathscr{D}_0\left(\mathbf{W}, \tilde{W}\right) = 0$. We also recall from the proof of Theorem 3 that $\|\|\tilde{W}\|\|_T \leq K_T$ almost surely.

For brevity, let us define several quantities that relate to the difference in the gradient updates between $\mathbf{W}$ and $\tilde{W}$:

$$q_3\left(k, z, \tilde{z}, j_2\right) = \partial_2\mathcal{L}\left(y, \hat{\mathbf{y}}\left(x; \mathbf{W}\left(k\right)\right)\right)\varphi_3'\left(\mathbf{H}_3\left(x; \mathbf{W}\left(k\right)\right)\right)\varphi_2\left(\mathbf{H}_2\left(x, j_2; \mathbf{W}\left(k\right)\right)\right)$$

$$- \partial_2\mathcal{L}\left(\tilde{y}, \hat{\mathbf{y}}\left(\tilde{x}; \tilde{W}\left(k\epsilon\right)\right)\right)\varphi_3'\left(\mathbf{H}_3\left(\tilde{x}; \tilde{W}\left(k\epsilon\right)\right)\right)\varphi_2\left(\mathbf{H}_2\left(\tilde{x}, j_2; \tilde{W}\left(k\epsilon\right)\right)\right),$$

$$r_3\left(k, z, j_2\right) = \xi_3\left(k\epsilon\right)\partial_2\mathcal{L}\left(y, \hat{\mathbf{y}}\left(x; \tilde{W}\left(k\epsilon\right)\right)\right)\varphi_3'\left(\mathbf{H}_3\left(x; \tilde{W}\left(k\epsilon\right)\right)\right)\varphi_2\left(\mathbf{H}_2\left(x, j_2; \tilde{W}\left(k\epsilon\right)\right)\right)$$

$$- \xi_3\left(k\epsilon\right)\mathbb{E}_Z\left[\partial_2\mathcal{L}\left(Y, \hat{\mathbf{y}}\left(X; \tilde{W}\left(k\epsilon\right)\right)\right)\varphi_3'\left(\mathbf{H}_3\left(X; \tilde{W}\left(k\epsilon\right)\right)\right)\varphi_2\left(\mathbf{H}_2\left(X, j_2; \tilde{W}\left(k\epsilon\right)\right)\right)\right],$$

$$q_2\left(k, z, \tilde{z}, j_1, j_2\right) = \Delta_2^{\mathbf{H}}\left(z, j_2; \mathbf{W}\left(k\right)\right)\varphi_1\left(\langle\mathbf{w}_1\left(k, j_1\right), x\rangle\right)$$

$$- \Delta_2^{\mathbf{H}} \left( \tilde{z}, j_2; \tilde{W} \left( k\epsilon \right) \right) \varphi_1 \left( \langle \tilde{w}_1 \left( k\epsilon, j_1 \right), \tilde{x} \rangle \right),$$

$$r_2 \left( k, z, j_1, j_2 \right) = \xi_2 \left( k\epsilon \right) \Delta_2^{\mathbf{H}} \left( z, j_2; \tilde{W} \left( k\epsilon \right) \right) \varphi_1 \left( \langle \tilde{w}_1 \left( k\epsilon, j_1 \right), x \rangle \right)$$

$$- \xi_2 \left( k\epsilon \right) \mathbb{E}_Z \left[ \Delta_2^{\mathbf{H}} \left( Z, j_2; \tilde{W} \left( k\epsilon \right) \right) \varphi_1 \left( \langle \tilde{w}_1 \left( k\epsilon, j_1 \right), X \rangle \right) \right],$$

$$q_1 \left( k, z, \tilde{z}, j_1 \right) = \frac{1}{n_2} \sum_{j_2=1}^{n_2} \Delta_2^{\mathbf{H}} \left( z, j_2; \mathbf{W} \left( k \right) \right) \mathbf{w}_2 \left( k, j_1, j_2 \right) \varphi_1' \left( \langle \mathbf{w}_1 \left( k, j_1 \right), x \rangle \right) x$$

$$- \frac{1}{n_2} \sum_{j_2=1}^{n_2} \Delta_2^{\mathbf{H}} \left( \tilde{z}, j_2; \tilde{W} \left( k\epsilon \right) \right) \tilde{w}_2 \left( k\epsilon, j_1, j_2 \right) \varphi_1' \left( \langle \tilde{w}_1 \left( k\epsilon, j_1 \right), \tilde{x} \rangle \right) \tilde{x},$$

$$r_1 \left( k, z, j_1 \right) = \xi_1 \left( k\epsilon \right) \frac{1}{n_2} \sum_{j_2=1}^{n_2} \Delta_2^{\mathbf{H}} \left( z, j_2; \tilde{W} \left( k\epsilon \right) \right) \tilde{w}_2 \left( k\epsilon, j_1, j_2 \right) \varphi_1' \left( \langle \tilde{w}_1 \left( k\epsilon, j_1 \right), x \rangle \right) x$$

$$- \xi_1 \left( k\epsilon \right) \mathbb{E}_Z \left[ \frac{1}{n_2} \sum_{j_2=1}^{n_2} \Delta_2^{\mathbf{H}} \left( Z, j_2; \tilde{W} \left( k\epsilon \right) \right) \tilde{w}_2 \left( k\epsilon, j_1, j_2 \right) \varphi_1' \left( \langle \tilde{w}_1 \left( k\epsilon, j_1 \right), X \rangle \right) X \right].$$

Let us also define:

$$q_3^H \left( k, x \right) = \mathbf{H}_3 \left( x; \mathbf{W} \left( k \right) \right) - \mathbf{H}_3 \left( x; \tilde{W} \left( k\epsilon \right) \right),$$

$$q_2^H \left( k, x, j_2 \right) = \mathbf{H}_2 \left( x, j_2; \mathbf{W} \left( k \right) \right) - \mathbf{H}_2 \left( x, j_2; \tilde{W} \left( k\epsilon \right) \right).$$

We proceed in several steps.

**Step 1: Decomposition.** As shown in the proof of Theorem 3:

$$\max_{j_2 \leq n_2} \left| \frac{\partial}{\partial t} \tilde{w}_3 \left( t + \xi, j_2 \right) - \frac{\partial}{\partial t} \tilde{w}_3 \left( t, j_2 \right) \right| \leq K_{t+\xi} \xi,$$

$$\max_{j_1 \leq n_1, \, j_2 \leq n_2} \left| \frac{\partial}{\partial t} \tilde{w}_2 \left( t + \xi, j_1, j_2 \right) - \frac{\partial}{\partial t} \tilde{w}_2 \left( t, j_1, j_2 \right) \right| \leq K_{t+\xi} \xi,$$

$$\max_{j_1 \leq n_1} \left| \frac{\partial}{\partial t} \tilde{w}_1 \left( t + \xi, j_1 \right) - \frac{\partial}{\partial t} \tilde{w}_1 \left( t, j_1 \right) \right| \leq K_{t+\xi} \xi.$$

for any $t \geq 0$ and $\xi \geq 0$. These time-interpolation estimates, along with Assumption 1, allow to derive the following. We first have:

$$\max_{j_2 \leq n_2} \left| \mathbf{w}_3 \left( \lfloor t/\epsilon \rfloor, j_2 \right) - \tilde{w}_3 \left( t, j_2 \right) \right|$$

$$\leq K \max_{j_2 \leq n_2} \left| \epsilon \sum_{k=0}^{\lfloor t/\epsilon \rfloor - 1} \xi_3 \left( k\epsilon \right) \mathbb{E}_Z \left[ q_3 \left( k, z \left( k \right), Z, j_2 \right) \right] \right| + t K_t \epsilon$$

$$\leq K \max_{j_2 \leq n_2} \left[ Q_{3,1} \left( \lfloor t/\epsilon \rfloor, j_2 \right) + Q_{3,2} \left( \lfloor t/\epsilon \rfloor, j_2 \right) \right] + t K_t \epsilon,$$

where we define

$$Q_{3,1} \left( \lfloor t/\epsilon \rfloor, j_2 \right) = \epsilon \sum_{k=0}^{\lfloor t/\epsilon \rfloor - 1} \left| q_3 \left( k, z \left( k \right), z \left( k \right), j_2 \right) \right|,$$

$$Q_{3,2} \left( \lfloor t/\epsilon \rfloor, j_2 \right) = \left| \epsilon \sum_{k=0}^{\lfloor t/\epsilon \rfloor - 1} r_3 \left( k, z \left( k \right), j_2 \right) \right|.$$

(Here $\sum_{k=0}^{\lfloor t/\epsilon \rfloor - 1} = 0$ if $\lfloor t/\epsilon \rfloor = 0$.) We have similarly:

$$\max_{j_1 \leq n_1} \left| \mathbf{w}_1 \left( \lfloor t/\epsilon \rfloor, j_1 \right) - \tilde{w}_1 \left( t, j_1 \right) \right| \leq K \max_{j_1 \leq n_1} \left[ Q_{1,1} \left( \lfloor t/\epsilon \rfloor, j_1 \right) + Q_{1,2} \left( \lfloor t/\epsilon \rfloor, j_1 \right) \right] + t K_t \epsilon,$$

$$\max_{j_1 \leq n_1,\, j_2 \leq n_2} |\mathbf{w}_2\left(\lfloor t/\epsilon \rfloor, j_1, j_2\right) - \tilde{w}_2\left(t, j_1, j_2\right)| \leq K \max_{j_1 \leq n_1,\, j_2 \leq n_2} \left[Q_{2,1}\left(\lfloor t/\epsilon \rfloor, j_1, j_2\right) + Q_{2,2}\left(\lfloor t/\epsilon \rfloor, j_1, j_2\right)\right] + tK_t\epsilon,$$

in which

$$Q_{1,1}\left(\lfloor t/\epsilon \rfloor, j_1\right) = \epsilon \sum_{k=0}^{\lfloor t/\epsilon \rfloor - 1} |q_1\left(k, z\left(k\right), z\left(k\right), j_1\right)|,$$

$$Q_{1,2}\left(\lfloor t/\epsilon \rfloor, j_1\right) = \left| \epsilon \sum_{k=0}^{\lfloor t/\epsilon \rfloor - 1} r_1\left(k, z\left(k\right), j_1\right) \right|,$$

$$Q_{2,1}\left(\lfloor t/\epsilon \rfloor, j_1, j_2\right) = \epsilon \sum_{k=0}^{\lfloor t/\epsilon \rfloor - 1} |q_2\left(k, z\left(k\right), z\left(k\right), j_1, j_2\right)|,$$

$$Q_{2,2}\left(\lfloor t/\epsilon \rfloor, j_1, j_2\right) = \left| \epsilon \sum_{k=0}^{\lfloor t/\epsilon \rfloor - 1} r_2\left(k, z\left(k\right), j_1, j_2\right) \right|.$$

The task is now to bound $Q_{1,1}$, $Q_{1,2}$, $Q_{2,1}$, $Q_{2,2}$, $Q_{3,1}$ and $Q_{3,2}$.

**Step 2: Bounding the terms.** Before we proceed, let us give some bounds for $q_3^H$ and $q_2^H$, which hold for any $x \in \mathbb{R}^d$:

$$\left|q_2^H\left(k, x, j_2\right)\right| \leq K \max_{j_1 \leq n_1} \left(|\tilde{w}_2\left(k\epsilon, j_1, j_2\right)| \, |\mathbf{w}_1\left(k, j_1\right) - \tilde{w}_1\left(k\epsilon, j_1\right)| + |\mathbf{w}_2\left(k, j_1, j_2\right) - \tilde{w}_2\left(k\epsilon, j_1, j_2\right)|\right)$$

$$\leq K_{k\epsilon} \mathscr{D}_{k\epsilon}\left(\tilde{W}, \mathbf{W}\right),$$

$$\left|q_3^H\left(k, x\right)\right| \leq K \max_{j_2 \leq n_2} \left(|\mathbf{w}_3\left(k, j_2\right) - \tilde{w}_3\left(k\epsilon, j_2\right)| + |\tilde{w}_3\left(k\epsilon, j_2\right)| \left|q_2^H\left(k, x, j_2\right)\right|\right)$$

$$\leq K_{k\epsilon} \mathscr{D}_{k\epsilon}\left(\tilde{W}, \mathbf{W}\right).$$

With these, we have the following:

- Let us bound $Q_{3,1}$. By Assumption 1,

$$|q_3\left(k, z\left(k\right), z\left(k\right), j_2\right)| \leq K \left(q_2^H\left(k, x\left(k\right), j_2\right) + \left|q_3^H\left(k, x\left(k\right)\right)\right|\right).$$

We then get:

$$\max_{j_2 \leq n_2} Q_{3,1}\left(\lfloor t/\epsilon \rfloor, j_2\right) \leq K_t \epsilon \sum_{k=0}^{\lfloor t/\epsilon \rfloor - 1} \mathscr{D}_{k\epsilon}\left(\tilde{W}, \mathbf{W}\right).$$

- Similarly to $Q_{3,1}$, we consider $Q_{2,1}$:

$$|q_2\left(k, z\left(k\right), z\left(k\right), j_1, j_2\right)| \leq K \left|\Delta_2^{\mathbf{H}}\left(z\left(k\right), j_2; \mathbf{W}\left(k\right)\right) - \Delta_2^{\mathbf{H}}\left(z\left(k\right), j_2; \tilde{W}\left(k\epsilon\right)\right)\right|$$

$$+ K \left|\Delta_2^{\mathbf{H}}\left(z\left(k\right), j_2; \tilde{W}\left(k\epsilon\right)\right)\right| |\mathbf{w}_1\left(k, j_1\right) - \tilde{w}_1\left(k\epsilon, j_1\right)|$$

$$\leq K_{k\epsilon}\left(\left|q_2^H\left(k, x\left(k\right), j_2\right)\right| + \left|q_3^H\left(k, x\left(k\right)\right)\right|\right)$$

$$+ K |\mathbf{w}_3\left(k, j_2\right) - \tilde{w}_3\left(k\epsilon, j_2\right)| + K_{k\epsilon} |\mathbf{w}_1\left(k, j_1\right) - \tilde{w}_1\left(k\epsilon, j_1\right)|,$$

which yields

$$\max_{j_1 \leq n_1,\, j_2 \leq n_2} Q_{2,1}\left(\lfloor t/\epsilon \rfloor, j_1, j_2\right) \leq K_t \epsilon \sum_{k=0}^{\lfloor t/\epsilon \rfloor - 1} \mathscr{D}_{k\epsilon}\left(\tilde{W}, \mathbf{W}\right).$$

- Again we get a similar bound for $Q_{1,1}$:

$$|q_1\left(k, z\left(k\right), z\left(k\right), j_1\right)| \leq \frac{K}{n_2} \sum_{j_2=1}^{n_2} |\tilde{w}_2\left(k\epsilon, j_1, j_2\right)| \left|\Delta_2^{\mathbf{H}}\left(z\left(k\right), j_2; \mathbf{W}\left(k\right)\right) - \Delta_2^{\mathbf{H}}\left(z\left(k\right), j_2; \tilde{W}\left(k\epsilon\right)\right)\right|$$

$$+ \frac{K}{n_2} \sum_{j_2=1}^{n_2} \left| \Delta_2^{\mathbf{H}} \left( z\left(k\right), j_2; \tilde{W}\left(k\epsilon\right) \right) \right| \left| \mathbf{w}_2\left(k, j_1, j_2\right) - \tilde{w}_2\left(k\epsilon, j_1, j_2\right) \right|$$

$$+ \frac{K}{n_2} \sum_{j_2=1}^{n_2} \left| \tilde{w}_2\left(k\epsilon, j_1, j_2\right) \right| \left| \Delta_2^{\mathbf{H}} \left( z\left(k\right), j_2; \tilde{W}\left(k\epsilon\right) \right) \right| \left| \mathbf{w}_1\left(k, j_1\right) - \tilde{w}_1\left(k\epsilon, j_1\right) \right|$$

$$\leq K_{k\epsilon} \left( \max_{j_2 \leq n_2} \left| q_2^H\left(k, x\left(k\right), j_2\right) \right| + \left| q_3^H\left(k, x\left(k\right)\right) \right| \right) + K_{k\epsilon} \mathscr{D}_{k\epsilon}\left( \tilde{W}, \mathbf{W} \right),$$

which yields

$$\max_{j_1 \leq n_1} Q_{1,1}\left( \lfloor t/\epsilon \rfloor, j_1 \right) \leq K_t \epsilon \sum_{k=0}^{\lfloor t/\epsilon \rfloor - 1} \mathscr{D}_{k\epsilon}\left( \tilde{W}, \mathbf{W} \right).$$

- Let us bound $Q_{3,2}$. Let us define:

$$\underline{r}_3\left(k, j_2\right) = \sum_{\ell=0}^{k-1} r_3\left(k, z\left(k\right), j_2\right), \qquad \underline{r}_3\left(0, j_2\right) = 0.$$

Let $\mathcal{F}_k$ be the sigma-algebra generated by $\{z\left(\ell\right) : \ell \in \{0, ..., k-1\}\}$. Note that $\{\underline{r}_3\left(k, j_2\right)\}_{k \in \mathbb{N}}$ is a martingale adapted to $\{\mathcal{F}_k\}_{k \in \mathbb{N}}$. Furthermore, for $k \leq T/\epsilon$, the martingale difference is bounded: $|r_3\left(k, z\left(k\right), j_2\right)| \leq K$ by Assumption 1. Therefore, by Theorem 20 and the union bound, we have:

$$\mathbb{P}\left( \max_{j_2 \leq n_2} \max_{\ell \in \{0,1,...,T/\epsilon\}} Q_{3,2}\left(\ell, j_2\right) \geq \xi \right) \leq 2n_2 \exp\left( -\frac{\xi^2}{K\left(T+1\right)\epsilon} \right).$$

- The bounding of $Q_{2,2}$ is similar: $|r_2\left(k, z\left(k\right), j_1, j_2\right)| \leq K_{k\epsilon}$ almost surely by Assumption 1, and thus

$$\mathbb{P}\left( \max_{j_1 \leq n_1, \, j_2 \leq n_2} \max_{\ell \in \{0,1,...,T/\epsilon\}} Q_{2,2}\left(\ell, j_1, j_2\right) \geq \xi \right) \leq 2n_1 n_2 \exp\left( -\frac{\xi^2}{K_T\left(T+1\right)\epsilon} \right).$$

- Again the bounding of $Q_{1,2}$ is also similar: $|r_1\left(k, z\left(k\right), j_1\right)| \leq K_{k\epsilon}$ almost surely by Assumption 1, and thus

$$\mathbb{P}\left( \max_{j_1 \leq n_1} \max_{\ell \in \{0,1,...,T/\epsilon\}} Q_{1,2}\left(\ell, j_1\right) \geq \xi \right) \leq 2n_1 \exp\left( -\frac{\xi^2}{K_T\left(T+1\right)\epsilon} \right).$$

**Step 3: Putting everything together.** All the above results give us

$$\mathscr{D}_{\lfloor t/\epsilon \rfloor \epsilon}\left( \tilde{W}, \mathbf{W} \right) \leq K_T \epsilon \sum_{k=0}^{\lfloor t/\epsilon \rfloor - 1} \mathscr{D}_{k\epsilon}\left( \tilde{W}, \mathbf{W} \right) + \xi + T K_T \epsilon \qquad \forall t \leq T,$$

which hold with probability at least

$$1 - 2n_1 n_2 \exp\left( -\frac{\xi^2}{K_T\left(T+1\right)\epsilon} \right).$$

The above event implies, by Gronwall's lemma,

$$\mathscr{D}_T\left( \tilde{W}, \mathbf{W} \right) \leq \left( \xi + \epsilon \right) e^{K_T}.$$

Choosing $\xi = K_T \sqrt{\left(T+1\right)\epsilon \log\left(2n_1 n_2/\delta\right)}$ completes the proof. $\qquad\square$

## C.3 PROOFS OF COROLLARIES 4 AND 5

*Proof of Corollary 4.* By the assumption on $\psi$ and Assumption 1, we have:

$$\left| \mathbb{E}_Z \left[ \psi\left( Y, \hat{\mathbf{y}}\left( X; \mathbf{W}\left( \lfloor t/\epsilon \rfloor \right) \right) \right) - \psi\left( Y, \hat{y}\left( X; W\left(t\right) \right) \right) \right] \right|$$

$$\leq K \mathbb{E}_Z \left[ \left| \mathbf{H}_3 \left( X; \mathbf{W} \left( \lfloor t/\epsilon \rfloor \right) \right) - H_3 \left( X; W \left( t \right) \right) \right| \right]$$

$$\leq K \mathbb{E}_Z \left[ \left| H_3 \left( X; W \left( t \right) \right) - H_3 \left( X; W \left( \lfloor t/\epsilon \rfloor \epsilon \right) \right) \right| \right] + K \mathbb{E}_Z \left[ \left| \mathbf{H}_3 \left( X; \mathbf{W} \left( \lfloor t/\epsilon \rfloor \right) \right) - H_3 \left( X; W \left( \lfloor t/\epsilon \rfloor \epsilon \right) \right) \right| \right].$$

An inspection of the proof of Theorem 3 (in particular, the proofs of Theorems 14 and 15) reveals that firstly,

$$\sup_{t \leq T} \mathbb{E}_Z \left[ \left| H_3 \left( X; W \left( t \right) \right) - H_3 \left( X; W \left( \lfloor t/\epsilon \rfloor \epsilon \right) \right) \right| \right] \leq K_T \epsilon,$$

and secondly,

$$\sup_{t \leq T} \mathbb{E}_Z \left[ \left| H_3 \left( X; W \left( \lfloor t/\epsilon \rfloor \epsilon \right) \right) - \mathbf{H}_3 \left( X; \mathbf{W} \left( \lfloor t/\epsilon \rfloor \right) \right) \right| \right]$$

$$\leq K_T \mathscr{D}_T \left( W, \mathbf{W} \right) + \frac{1}{\sqrt{n_{\min}}} \log^{1/2} \left( \frac{3 T n_{\max}^2}{\delta} + e \right) e^{K_T}$$

with probability at least $1 - \delta$. Together with Theorem 3, we obtain the claim. $\quad\square$

*Proof of Corollary 5.* Observe that for each index $\{N_1, N_2\}$ of Init, one obtains a neural network initialization $\mathbf{W}(0)$ with law $\rho$ by setting

$$\mathbf{w}_1(0, j_1) = w_1(0, C_1(j_1)), \quad \mathbf{w}_2(0, j_1, j_2) = w_2 \left( 0, C_1 \left( j_1 \right), C_2 \left( j_2 \right) \right),$$
$$\mathbf{w}_3(0, j_2) = w_3 \left( 0, C_2 \left( j_2 \right) \right), \quad j_1 \in [N_1], \ j_2 \in [N_2].$$

We consider the evolution $\mathbf{W}(k)$ starting from $\mathbf{W}(0)$, which is independent of $W$. Note that $\mathbf{W}(k)$ is a deterministic function of its initialization $\mathbf{W}(0)$ and the data $\{z(j)\}_{j \leq k}$. Similarly, we consider the counterpart for $\hat{W}$: the evolution $\hat{\mathbf{W}}(k)$ as a function of the initialization $\hat{\mathbf{W}}(0)$ and the data $\{\hat{z}(j)\}_{j \leq k}$. Due to sharing the same distribution for both the initialization and the data, these evolutions have the same law; to be specific, $\underline{\mathbf{W}}(n_1, n_2, T)$ and $\underline{\hat{\mathbf{W}}}(n_1, n_2, T)$ has the same distribution for any $n_1$, $n_2$ and $T$, where we define

$$\underline{\mathbf{W}}(n_1, n_2, T) = \left\{ \mathbf{w}_1 \left( k, j_1 \right), \ \mathbf{w}_2 \left( k, j_1, j_2 \right), \ \mathbf{w}_3 \left( k, j_2 \right) : \right.$$
$$\left. j_1 \in [n_1], \ j_2 \in [n_2], \ k \leq \lfloor T/\epsilon \rfloor \right\},$$

and a similar definition for $\underline{\hat{\mathbf{W}}}(n_1, n_2, T)$. In other words,

$$\mathscr{W}\left( \underline{\mathbf{W}}, \underline{\hat{\mathbf{W}}} \right) \equiv \inf_{\text{coupling of } \left( \underline{\mathbf{W}}, \underline{\hat{\mathbf{W}}} \right)} \mathbb{E} \left[ \max_{k \leq \lfloor T/\epsilon \rfloor, \ j_1 \leq n_1, \ j_2 \leq n_2} \left\{ \left| \mathbf{w}_1 \left( k, j_1 \right) - \hat{\mathbf{w}}_1 \left( k, j_1 \right) \right|, \right. \right.$$

$$\left. \left. \left| \mathbf{w}_2 \left( k, j_1, j_2 \right) - \hat{\mathbf{w}}_2 \left( k, j_1, j_2 \right) \right|, \ \left| \mathbf{w}_3 \left( k, j_2 \right) - \hat{\mathbf{w}}_3 \left( k, j_2 \right) \right| \right\} \right] = 0.$$

Theorem 3 implies that for any tuple $\{n_1, n_2\}$ such that $n_1 \leq N_1$ and $n_2 \leq N_2$, with probability at least $1 - 2\delta$,

$$\mathscr{D}_T^{(n_1, n_2)} \left( W, \mathbf{W} \right) \equiv \max \left( \sup_{t \leq T, \ j_1 \leq n_1, \ j_2 \leq n_2} \left| \mathbf{w}_2 \left( t, j_1, j_2 \right) - w_2 \left( t, C_1 \left( j_1 \right), C_2 \left( j_2 \right) \right) \right|, \right.$$

$$\sup_{t \leq T, \ j_2 \leq n_2} \left| \mathbf{w}_3 \left( t, j_2 \right) - w_3 \left( t, C_2 \left( j_2 \right) \right) \right|,$$

$$\left. \sup_{t \leq T, \ j_1 \leq n_1} \left| \mathbf{w}_1 \left( t, j_1 \right) - w_1 \left( t, C_1 \left( j_1 \right) \right) \right| \right) \leq \tilde{O}_{\delta, T} \left( \epsilon, N_1, N_2 \right),$$

where $\tilde{O}_{\delta, T} \left( \epsilon, N_1, N_2 \right) \to 0$ as $\epsilon \to 0$ and $N_{\min}^{-1} \log N_{\max} \to 0$ with $N_{\min} = \min \{N_1, N_2\}$ and $N_{\max} = \max \{N_1, N_2\}$. We also have a similar result for $\hat{\mathbf{W}}$ and $\hat{W}$. As such, with probability at least $1 - 4\delta$,

$$\mathscr{W}\left( \mathcal{W}, \hat{\mathcal{W}} \right) \equiv \inf_{\text{coupling of } \left( \mathcal{W}, \hat{\mathcal{W}} \right)} \mathbb{E} \left[ \sup_{t \leq T, \ j_1 \leq n_1, \ j_2 \leq n_2} \left\{ \left| w_1 \left( t, C_1 \left( j_1 \right) \right) - \hat{w}_1 \left( t, \hat{C}_1 \left( j_1 \right) \right) \right|, \right. \right.$$

$$\left. \left. \left| w_2 \left( t, C_1 \left( j_1 \right), C_2 \left( j_2 \right) \right) - \hat{w}_2 \left( t, \hat{C}_1 \left( j_1 \right), \hat{C}_2 \left( j_2 \right) \right) \right|, \ \left| w_3 \left( t, C_2 \left( j_2 \right) \right) - \hat{w}_3 \left( t, \hat{C}_2 \left( j_2 \right) \right) \right| \right\} \right]$$

$$\leq \mathscr{D}_T^{(n_1, n_2)} \left( W, \mathbf{W} \right) + \mathscr{D}_T^{(n_1, n_2)} \left( \hat{W}, \hat{\mathbf{W}} \right) + \mathscr{W}\left( \underline{\mathbf{W}}, \underline{\hat{\mathbf{W}}} \right)$$

$$\leq 2 \tilde{O}_{\delta, T} \left( \epsilon, N_1, N_2 \right).$$

By fixing the tuple $\{n_1, n_2\}$ while letting $\epsilon \to 0$, $N_{\min}^{-1} \log N_{\max} \to 0$ and $\delta \to 0$, we obtain the claim. $\quad\square$

## D  GLOBAL CONVERGENCE: PROOFS FOR SECTION 4

### D.1  PROOF OF PROPOSITION 7

*Proof of Proposition 7.* Consider a probability space $(\Lambda, \mathcal{G}, P_0)$ with random processes $\mathbb{R}^d$-valued $p_1(\theta_1)$, $\mathbb{R}$-valued $p_2(\theta_1, \theta_2)$ and $\mathbb{R}$-valued $p_3(\theta_2)$, which are indexed by $(\theta_1, \theta_2) \in [0, 1] \times [0, 1]$, such that the following holds. Let $m_1$ and $m_2$ be two arbitrary finite positive integers and, with these integers, let $\left\{ \theta_i^{(k_i)} \in [0, 1] : k_i \in [m_i], i = 1, 2 \right\}$ be an arbitrary collection. For each $i = 1, 2$, let $S_i$ be the set of unique elements in $\left\{ \theta_i^{(k_i)} : k_i \in [m_i] \right\}$. Similarly, let $R_2$ be the set of unique pairs in $\left\{ \left( \theta_1^{(k_1)}, \theta_2^{(k_2)} \right) : k_1 \in [m_1], k_2 \in [m_2] \right\}$. We have that $\{ p_1(\theta_1) : \theta_1 \in S_1 \}$, $\{ p_3(\theta_2) : \theta_2 \in S_2 \}$ and $\{ p_2(\theta_1, \theta_2) : (\theta_1, \theta_2) \in R_2 \}$ are all mutually independent. In addition, we also have Law $(p_1(\theta_1)) = \rho^1$, Law $(p_3(\theta_2)) = \rho^3$ and Law $(p_2(\theta_1', \theta_2')) = \rho^2$ for any $\theta_1 \in S_1$, $\theta_2 \in S_2$ and $(\theta_1', \theta_2') \in R_2$. Such a space exists by Kolmogorov's extension theorem.

We now construct the desired neuronal embedding. For $i = 1, 2$, consider $\Omega_i = \Lambda \times [0, 1]$ and $\mathcal{F}_i = \mathcal{G} \times \mathcal{B}([0, 1])$, equipped with the product measure $P_0 \times \text{Unif}([0, 1])$ in which $\text{Unif}([0, 1])$ is the uniform measure over $[0, 1]$ equipped with the Borel sigma-algebra $\mathcal{B}([0, 1])$. We construct $\Omega = \Omega_1 \times \Omega_2$ and $\mathcal{F} = \mathcal{F}_1 \times \mathcal{F}_2$, equipped with the product measure $P = (P_0 \times \text{Unif}([0, 1]))^2$. Define the deterministic functions $w_1^0 : \Omega_1 \to \mathbb{R}^d$, $w_2^0 : \Omega_1 \times \Omega_2 \to \mathbb{R}$ and $w_3^0 : \Omega_2 \to \mathbb{R}$:

$$w_1^0((\lambda_1, \theta_1)) = p_1(\theta_1)(\lambda_1),$$
$$w_2^0((\lambda_1, \theta_1), (\lambda_2, \theta_2)) = p_2(\theta_1, \theta_2)(\lambda_2),$$
$$w_3^0((\lambda_2, \theta_2)) = p_3(\theta_2)(\lambda_2).$$

It is easy to check that this construction yields the desired neuronal embedding. $\square$

### D.2  PROOF OF THEOREM 8

We first present a measurability argument, which is crucial to showing that a certain universal approximation property holds throughout the course of training.

**Lemma 16** (Measurability argument). *Consider a family* Init *of initialization laws, which are* $(\rho^1, \rho^2, \rho^3)$*-i.i.d., such that* $\rho^2$*-almost surely* $|\mathbf{w}_2| \leq K$ *and* $\rho^3$*-almost surely* $|\mathbf{w}_3| \leq K$. *There exists a neuronal embedding* $\left( \Omega, \mathcal{F}, P, \{ w_i^0 \}_{i=1,2,3} \right)$ *of* Init *such that there exist Borel functions* $w_1^*$ *and* $\Delta_2^{H*}$ *for which* $P$*-almost surely, for all* $t \geq 0$,

$$w_1(t, C_1) = w_1^*(t, w_1^0(C_1)),$$
$$\Delta_2^H(z, C_2; W(t)) = \Delta_2^{H*}(t, z, w_3^0(C_2)),$$

*where* $W(t)$ *is the MF dynamics formed under the coupling procedure with this neuronal embedding as described in Section 3.1. Furthermore,*

$$\frac{\partial}{\partial t} w_1^*(t, u_1) = -\xi_1(t) \int \mathbb{E}_Z \left[ \Delta_2^{H*}(t, Z, u_3) u_2 \varphi_1'(\langle w_1^*(t, u_1), X \rangle) X \right] \rho^2(du_2) \rho^3(du_3)$$

$$+ \xi_1(t) \int_0^t \xi_2(s) \mathbb{E}_{Z, Z'} \left[ \int \Delta_2^{H*}(t, Z, u_3) \Delta_2^{H*}(s, Z', u_3) \rho^3(du_3) \right.$$

$$\left. \times \varphi_1(\langle w_1^*(s, u_1), X' \rangle) \varphi_1'(\langle w_1^*(t, u_1), X \rangle) X \right] ds,$$

*with initialization* $w_1^*(0, u_1) = u_1$ *for all* $u_1 \in \text{supp}(\rho^1)$ *and* $t \geq 0$, *where* $Z'$ *is an independent copy of* $Z$.

*Proof.* We denote by $K_t$ a constant that may depend on $t$ and is finite with finite $t$. By Proposition 7, there exists a neuronal embedding that accommodates Init. We recall its construction and reuse the notations from the proof of Proposition 7; in particular:

$$w_1^0((\lambda_1, \theta_1)) = p_1(\theta_1)(\lambda_1),$$

$$w_2^0 \left( (\lambda_1, \theta_1), (\lambda_2, \theta_2) \right) = p_2 (\theta_1, \theta_2) (\lambda_2),$$
$$w_3^0 \left( (\lambda_2, \theta_2) \right) = p_3 (\theta_2) (\lambda_2).$$

Let $\mathcal{S}_1$, $\mathcal{S}_3$ and $\mathcal{S}_2$ denote the sigma-algebras generated by $w_1^0(C_1)$, $w_3^0(C_2)$ and $\left( w_1^0(C_1), w_2^0(C_1, C_2), w_3^0(C_2) \right)$ respectively. Let $\mathcal{S}_{13}$ denote the sigma-algebra generated by $\mathcal{S}_1$ and $\mathcal{S}_3$. We also let $\mathcal{S}_1^Z$ to denote the sigma-algebra generated by $\mathcal{S}_1$ and the sigma-algebra of the data $Z$. We define similarly for $\mathcal{S}_2^Z$ and $\mathcal{S}_3^Z$.

**Step 1: Reduced dynamics.** Given the MF dynamics $W(t)$, let us define
$$\bar{\Delta}_3 (t, c_2) = \mathbb{E} \left[ \Delta_3 (C_2; W(t)) | \mathcal{S}_3 \right] (c_2),$$
$$\bar{\Delta}_2 (t, c_1, c_2) = \mathbb{E} \left[ \Delta_2 (C_1, C_2; W(t)) | \mathcal{S}_2 \right] (c_1, c_2),$$
$$\bar{\Delta}_1 (t, c_1) = \mathbb{E} \left[ \Delta_1 (C_1; W(t)) | \mathcal{S}_1 \right] (c_1).$$
We recall from the proof of Theorem 3 that for any $t, s \geq 0$,
$$\text{ess-sup} \left| w_3 (t, C_2) - w_3 (s, C_2) \right| \leq K |t - s|,$$
$$\text{ess-sup} \left| w_2 (t, C_1, C_2) - w_2 (s, C_1, C_2) \right| \leq K_{t \vee s} |t - s|,$$
$$\text{ess-sup} \left| w_1 (t, C_1) - w_1 (s, C_1) \right| \leq K_{t \vee s} |t - s|.$$
Then by Lemma 13,
$$\mathbb{E} \left[ \left| \bar{\Delta}_3 (t, C_2) - \bar{\Delta}_3 (s, C_2) \right|^2 \right] \leq K_{t \vee s} |t - s|^2,$$
$$\mathbb{E} \left[ \left| \bar{\Delta}_2 (t, C_1, C_2) - \bar{\Delta}_2 (s, C_1, C_2) \right|^2 \right] \leq K_{t \vee s} |t - s|^2,$$
$$\mathbb{E} \left[ \left| \bar{\Delta}_1 (t, C_1) - \bar{\Delta}_1 (s, C_1) \right|^2 \right] \leq K_{t \vee s} |t - s|^2.$$
Therefore, by Kolmogorov continuity theorem, there exist continuous modifications of the (time-indexed) processes $\bar{\Delta}_1$, $\bar{\Delta}_2$ and $\bar{\Delta}_3$. We thus replace them with their continuous modifications, written by the same notations.

Given these continuous modifications, we consider the following *reduced dynamics*:
$$\frac{\partial}{\partial t} \bar{w}_3 (t, c_2) = -\xi_3 (t) \bar{\Delta}_3 (t, c_2),$$
$$\frac{\partial}{\partial t} \bar{w}_2 (t, c_1, c_2) = -\xi_2 (t) \bar{\Delta}_2 (t, c_1, c_2),$$
$$\frac{\partial}{\partial t} \bar{w}_1 (t, c_1) = -\xi_1 (t) \bar{\Delta}_1 (t, c_1),$$
in which:

- $\bar{w}_1 : \mathbb{R}_{\geq 0} \times \Omega_1 \to \mathbb{R}^d$, $\bar{w}_2 : \mathbb{R}_{\geq 0} \times \Omega_1 \times \Omega_2 \mapsto \mathbb{R}$, $\bar{w}_3 : \mathbb{R}_{\geq 0} \times \Omega_3 \mapsto \mathbb{R}$.

- $\bar{W}(t) = \{ \bar{w}_1 (t, \cdot), \bar{w}_2 (t, \cdot, \cdot), \bar{w}_3 (t, \cdot) \}$ is the collection of reduced parameters at time $t$,

- the initialization is $\bar{w}_1 (0, \cdot) = w_1^0 (\cdot)$, $\bar{w}_2 (0, \cdot, \cdot) = w_2^0 (\cdot, \cdot)$ and $\bar{w}_3 (0, \cdot) = w_3^0 (\cdot)$, i.e. $\bar{W}(0) = W(0)$.

**Step 2: Measurability of the reduced dynamics.** It is easy to see that $\bar{w}_3 (t, C_2)$ is $\mathcal{S}_3$-measurable by its construction and the fact $\bar{w}_3 (0, C_2) = w_3^0 (C_2)$ is $\mathcal{S}_3$-measurable. Similarly, $\bar{w}_2 (t, C_1, C_2)$ is $\mathcal{S}_2$-measurable and $\bar{w}_1 (t, C_1)$ is $\mathcal{S}_1$-measurable.

Notice that there exist Borel functions $\bar{w}_1^*$, $\bar{w}_2^*$ and $\bar{w}_3^*$ for which $P$-almost surely,
$$\bar{w}_1 (t, C_1) = \bar{w}_1^* \left( t, w_1^0 (C_1) \right),$$
$$\bar{w}_2 (t, C_1, C_2) = \bar{w}_2^* \left( t, w_1^0 (C_1), w_2^0 (C_1, C_2), w_3^0 (C_2) \right),$$
$$\bar{w}_3 (t, C_2) = \bar{w}_3^* \left( t, w_3^0 (C_2) \right).$$
Indeed, since $\bar{w}_2 (t, C_1, C_2)$ is $\mathcal{S}_2$-measurable, there exists a function $\bar{w}_2^* (t, \cdot)$ for each rational $t$ such that the desired identity holds for $P$-almost every $(C_1, C_2)$ and for all rational $t \geq 0$. Since $\bar{w}_2$ is continuous in time, there is a unique continuous (in time) function $\bar{w}_2^* (t, \cdot)$ such that the identity holds for all $t \geq 0$ and for $P$-almost every $(C_1, C_2)$. The same argument yields the construction of $\bar{w}_1^*$ and $\bar{w}_3^*$.

**Step 3: Measurability of constituent quantities.** We show that $H_2\left(X, C_2; \bar{W}(t)\right)$ is $\mathcal{S}_3^{\mathcal{Z}}$-measurable. Recall that

$$H_2\left(X, C_2; \bar{W}(t)\right) = \mathbb{E}_{C_1}\left[\bar{w}_2\left(t, C_1, C_2\right) \varphi_1\left(\langle\bar{w}_1\left(t, C_1\right), X\rangle\right)\right].$$

By the existence of $\bar{w}_1^*$ and $\bar{w}_2^*$, for each $t \geq 0$, there exists a Borel function $f_t$ such that almost surely

$$\bar{w}_2\left(t, C_1, C_2\right) \varphi_1\left(\langle\bar{w}_1\left(t, C_1\right), X\rangle\right) = f_t\left(X, w_1^0\left(C_1\right), w_2^0\left(C_1, C_2\right), w_3^0\left(C_2\right)\right).$$

We recall that $\rho^1$ and $\rho^2$ are the laws of $w_1^0\left(C_1\right)$ and $w_2^0\left(C_1, C_2\right)$. We analyze the following:

$$\mathbb{E}\left[\left|\left|H_2\left(X, C_2; \bar{W}(t)\right) - \int f_t\left(X, u_1, u_2, w_3^0\left(C_2\right)\right) \rho^1\left(du_1\right) \rho^2\left(du_2\right)\right|\right|^2\right]$$

$$= \mathbb{E}\left[\left|H_2\left(X, C_2; \bar{W}(t)\right)\right|^2\right] + \mathbb{E}\left[\left|\int f_t\left(X, u_1, u_2, w_3^0\left(C_2\right)\right) \rho^1\left(du_1\right) \rho^2\left(du_2\right)\right|^2\right]$$

$$- 2\mathbb{E}\left[H_2\left(X, C_2; \bar{W}(t)\right) \int f_t\left(X, u_1, u_2, w_3^0\left(C_2\right)\right) \rho^1\left(du_1\right) \rho^2\left(du_2\right)\right].$$

Let us evaluate the first term:

$$\mathbb{E}\left[\left|H_2\left(X, C_2; \bar{W}(t)\right)\right|^2\right]$$

$$= \mathbb{E}\left[\left|\mathbb{E}_{C_1}\left[f_t\left(X, w_1^0\left(C_1\right), w_2^0\left(C_1, C_2\right), w_3^0\left(C_2\right)\right)\right]\right|^2\right]$$

$$\stackrel{(a)}{=} \mathbb{E}\left[f_t\left(X, w_1^0\left(C_1\right), w_2^0\left(C_1, C_2\right), w_3^0\left(C_2\right)\right) f_t\left(X, w_1^0\left(C_1'\right), w_2^0\left(C_1', C_2\right), w_3^0\left(C_2\right)\right)\right]$$

$$\stackrel{(b)}{=} \mathbb{E}\Big[f_t\left(X, p_1\left(\theta_1\right)\left(\lambda_1\right), p_2\left(\theta_1, \theta_2\right)\left(\lambda_2\right), p_3\left(\theta_2\right)\left(\lambda_2\right)\right)$$

$$\times f_t\left(X, p_1\left(\theta_1'\right)\left(\lambda_1'\right), p_2\left(\theta_1', \theta_2\right)\left(\lambda_2\right), p_3\left(\theta_2\right)\left(\lambda_2\right)\right)\Big]$$

$$\stackrel{(c)}{=} \mathbb{E}\Big[f_t\left(X, p_1\left(\theta_1\right)\left(\lambda_1\right), p_2\left(\theta_1, \theta_2\right)\left(\lambda_2\right), p_3\left(\theta_2\right)\left(\lambda_2\right)\right)$$

$$\times f_t\left(X, p_1\left(\theta_1'\right)\left(\lambda_1'\right), p_2\left(\theta_1', \theta_2\right)\left(\lambda_2\right), p_3\left(\theta_2\right)\left(\lambda_2\right)\right) \mathbb{I}\left(\theta_1 \neq \theta_1'\right)\Big]$$

$$\stackrel{(d)}{=} \mathbb{E}_Z\left[\int f_t\left(X, u_1, u_2, u_3\right) f_t\left(X, u_1', u_2', u_3\right) \rho^1\left(du_1\right) \rho^1\left(du_1'\right) \rho^2\left(du_2\right) \rho^2\left(du_2'\right) \rho^3\left(du_3\right)\right],$$

where in step $(a)$, we define $C_1'$ to be an independent copy of $C_1$; in step $(b)$, we recall $C_1 = \left(\lambda_1, \theta_1\right)$; in step $(c)$, we recall $\theta_1, \theta_1' \sim \text{Unif}\left([0, 1]\right)$ and since $C_1$ is independent of $C_1'$, we have $\theta_1 \neq \theta_1'$ almost surely; step $(d)$ is owing to the independence property of the construction of the functions $p_1$, $p_2$ and $p_3$. We calculate the second term:

$$\mathbb{E}\left[\left|\int f_t\left(X, u_1, u_2, w_3^0\left(C_2\right)\right) \rho^1\left(du_1\right) \rho^2\left(du_2\right)\right|^2\right]$$

$$= \int \mathbb{E}\left[f_t\left(X, u_1, u_2, w_3^0\left(C_2\right)\right) f_t\left(X, u_1', u_2', w_3^0\left(C_2\right)\right)\right] \rho^1\left(du_1\right) \rho^2\left(du_2\right) \rho^1\left(du_1'\right) \rho^2\left(du_2'\right)$$

$$= \int \mathbb{E}_Z\left[f_t\left(X, u_1, u_2, u_3\right) f_t\left(X, u_1', u_2', u_3\right)\right] \rho^1\left(du_1\right) \rho^2\left(du_2\right) \rho^1\left(du_1'\right) \rho^2\left(du_2'\right) \rho^3\left(du_3\right),$$

as well as the last term:

$$\mathbb{E}\left[H_2\left(X, C_2; \bar{W}(t)\right) \int f_t\left(X, u_1, u_2, w_3^0\left(C_2\right)\right) \rho^1\left(du_1\right) \rho^2\left(du_2\right)\right]$$

$$= \mathbb{E}\left[f_t\left(X, w_1^0\left(C_1\right), w_2^0\left(C_1, C_2\right), w_3^0\left(C_2\right)\right) \int f_t\left(X, u_1, u_2, w_3^0\left(C_2\right)\right) \rho^1\left(du_1\right) \rho^2\left(du_2\right)\right]$$

$$= \int \mathbb{E}_Z\left[f_t\left(X, u_1, u_2, u_3\right) f_t\left(X, u_1', u_2', u_3\right)\right] \rho^1\left(du_1\right) \rho^2\left(du_2\right) \rho^1\left(du_1'\right) \rho^2\left(du_2'\right) \rho^3\left(du_3\right).$$

It is then easy to see that

$$\mathbb{E}\left[\left|H_2\left(X, C_2; \bar{W}(t)\right) - \int f_t\left(X, u_1, u_2, w_3^0\left(C_2\right)\right) \rho^1\left(du_1\right) \rho^2\left(du_2\right)\right|^2\right] = 0.$$

That is, we have almost surely

$$H_2\left(X, C_2; \bar{W}(t)\right) = \int f_t\left(X, u_1, u_2, w_3^0\left(C_2\right)\right) \rho^1\left(du_1\right) \rho^2\left(du_2\right).$$

Note that the right-hand side is $\mathcal{S}_3^Z$-measurable, and hence so is $H_2\left(X, C_2; \bar{W}(t)\right)$.

Next we consider $\Delta_2^H\left(Z, C_2; \bar{W}(t)\right)$. Recall that

$$\Delta_2^H\left(z, c_2; \bar{W}(t)\right) = \partial_2 \mathcal{L}\left(y, \hat{y}\left(x; \bar{W}(t)\right)\right) \varphi_3'\left(H_3\left(x; \bar{W}(t)\right)\right) \bar{w}_3\left(t, c_2\right) \varphi_2'\left(H_2\left(x, c_2; \bar{W}(t)\right)\right).$$

Then together with the existence of $\bar{w}_3^*$, we have $\Delta_2^H\left(Z, C_2; \bar{W}(t)\right)$ is $\mathcal{S}_3^Z$-measurable.

Now we consider $\mathbb{E}_{C_2}\left[\Delta_2^H\left(Z, C_2; \bar{W}(t)\right) \bar{w}_2\left(t, C_1, C_2\right)\right]$. With the existence of $\bar{w}_2^*$, there exists a Borel function $g_t$ such that

$$\Delta_2^H\left(Z, C_2; \bar{W}(t)\right) \bar{w}_2\left(t, C_1, C_2\right) = g_t\left(Z, w_1^0\left(C_1\right), w_2^0\left(C_1, C_2\right), w_3^0\left(C_2\right)\right).$$

Then with the same argument as the treatment of $H_2\left(X, C_2; \bar{W}(t)\right)$, one can show that

$$\mathbb{E}_{C_2}\left[\Delta_2^H\left(Z, C_2; \bar{W}(t)\right) \bar{w}_2\left(t, C_1, C_2\right)\right] = \int g_t\left(Z, w_1^0\left(C_1\right), u_2, u_3\right) \rho^2\left(du_2\right) \rho^3\left(du_3\right),$$

which is $\mathcal{S}_1^Z$-measurable.

Using these facts together with the existence of $\bar{w}_1^*$, $\bar{w}_2^*$ and $\bar{w}_3^*$, we have $\Delta_3\left(C_2; \bar{W}(t)\right)$ is $\mathcal{S}_3$-measurable, $\Delta_2\left(C_1, C_2; \bar{W}(t)\right)$ is $\mathcal{S}_{13}$-measurable and $\Delta_1\left(C_1; \bar{W}(t)\right)$ is $\mathcal{S}_1$-measurable.

**Step 4: Closeness between the MF dynamics and the reduced dynamics.** We shall use $\left\|W - \bar{W}\right\|_t$ with the same meaning as the distance between two sets of MF parameters. Recall by Lemma 11 that $\|\|W\|\|_T \leq K_T$ since $\|\|W\|\|_0 \leq K$. By the same argument, $\|\|\bar{W}\|\|_T \leq K_T$. Then by Lemma 13, we have for any $t \leq T$,

$$\text{ess-sup} \sup_{s \leq t}\left|\Delta_3\left(C_2; W(s)\right) - \Delta_3\left(C_2; \bar{W}(s)\right)\right| \leq K_T \left\|W - \bar{W}\right\|_t,$$

$$\text{ess-sup} \sup_{s \leq t}\left|\Delta_2\left(C_1, C_2; W(s)\right) - \Delta_2\left(C_1, C_2; \bar{W}(s)\right)\right| \leq K_T \left\|W - \bar{W}\right\|_t,$$

$$\text{ess-sup} \sup_{s \leq t}\left|\Delta_1\left(C_1; W(s)\right) - \Delta_1\left(C_1; \bar{W}(s)\right)\right| \leq K_T \left\|W - \bar{W}\right\|_t.$$

We have:

$$\text{ess-sup}\left|\bar{\Delta}_2\left(t, C_1, C_2\right) - \Delta_2\left(C_1, C_2; \bar{W}(t)\right)\right|$$
$$= \text{ess-sup}\left|\mathbb{E}\left[\Delta_2\left(C_1, C_2; W(t)\right) | \mathcal{S}_2\right] - \Delta_2\left(C_1, C_2; \bar{W}(t)\right)\right|$$
$$\leq \text{ess-sup}\left|\mathbb{E}\left[\Delta_2\left(C_1, C_2; \bar{W}(t)\right) | \mathcal{S}_2\right] - \Delta_2\left(C_1, C_2; \bar{W}(t)\right)\right|$$
$$+ \text{ess-sup}\left|\mathbb{E}\left[\Delta_2\left(C_1, C_2; \bar{W}(t)\right) - \Delta_2\left(C_1, C_2; W(t)\right) | \mathcal{S}_2\right]\right|$$
$$\overset{(a)}{=} \text{ess-sup}\left|\mathbb{E}\left[\Delta_2\left(C_1, C_2; \bar{W}(t)\right) - \Delta_2\left(C_1, C_2; W(t)\right) | \mathcal{S}_2\right]\right|$$
$$\leq \text{ess-sup}\left|\Delta_2\left(C_1, C_2; \bar{W}(t)\right) - \Delta_2\left(C_1, C_2; W(t)\right)\right|,$$

where step $(a)$ is because $\Delta_2\left(C_1, C_2; \bar{W}(t)\right)$ is $\mathcal{S}_{13}$-measurable from Step 3 and $\mathcal{S}_{13} \subseteq \mathcal{S}_2$. As such,

$$\text{ess-sup}\left|\bar{\Delta}_2\left(t, C_1, C_2\right) - \Delta_2\left(C_1, C_2; W(t)\right)\right| \leq 2K_T \left\|W - \bar{W}\right\|_t$$

almost surely for all rational $t \leq T$. By continuity in $t$ of both sides, the same holds for all $t \leq T$. Hence by Assumption 1,

$$\left|\frac{\partial}{\partial t} \bar{w}_2\left(t, C_1, C_2\right) - \frac{\partial}{\partial t} w_2\left(t, C_1, C_2\right)\right| \leq K \left|\bar{\Delta}_2\left(t, C_1, C_2\right) - \Delta_2\left(C_1, C_2; W(t)\right)\right|$$

$$\leq 2K_T \left\| W - \bar{W} \right\|_t,$$

for all $t \leq T$ almost surely, which leads to

$$\left| \bar{w}_2 \left( t, C_1, C_2 \right) - w_2 \left( t, C_1, C_2 \right) \right| \leq 2K_T \int_0^t \left\| W - \bar{W} \right\|_s ds$$

almost surely. One can obtain similar results for $\bar{w}_1$ versus $w_1$ and $\bar{w}_3$ versus $w_3$. Therefore,

$$\left\| W - \bar{W} \right\|_t \leq K_T \int_0^t \left\| W - \bar{W} \right\|_s ds.$$

Since $W(0) = \bar{W}(0)$, by Gronwall's inequality, $\left\| W - \bar{W} \right\|_t = 0$ for all $t \leq T$. In other words, since $T$ is arbitrary,

$$\bar{w}_1 \left( t, C_1 \right) = w_1 \left( t, C_1 \right), \quad \bar{w}_2 \left( t, C_1, C_2 \right) = w_2 \left( t, C_1, C_2 \right), \quad \bar{w}_3 \left( t, C_2 \right) = w_3 \left( t, C_2 \right),$$

for all $t \geq 0$ almost surely.

**Step 5: Concluding.** The first claim of the lemma is proven by the conclusion of Step 4 and by choosing $w_1^* = \bar{w}_1^*$, $w_2^* = \bar{w}_2^*$ and $w_3^* = \bar{w}_3^*$, as well as the measurability facts from Step 3. To prove the second claim, since $\Delta_2^H \left( Z, C_2; \bar{W}(t) \right)$ is $\mathcal{S}_3^Z$-measurable and $\left\| W - \bar{W} \right\|_t = 0$ for all $t \geq 0$, there exists a Borel function $\Delta_2^{H*}$ such that

$$\Delta_2^H \left( Z, C_2; \bar{W}(t) \right) = \Delta_2^H \left( Z, C_2; W(t) \right) = \Delta_2^{H*} \left( t, Z, w_3^0 \left( C_2 \right) \right)$$

for all $t \geq 0$ almost surely, by the same argument in Step 2. These facts, together with the dynamics of $w_1$ and $w_2$, imply that almost surely, for all $t \geq 0$,

$$\frac{\partial}{\partial t} w_2 \left( t, C_1, C_2 \right)$$
$$= -\xi_2 (t) \, \mathbb{E}_Z \left[ \Delta_2^{H*} \left( t, Z, w_3^0 \left( C_2 \right) \right) \varphi_1 \left( \left\langle w_1^* \left( t, w_1^0 \left( C_1 \right) \right), X \right\rangle \right) \right],$$
$$\frac{\partial}{\partial t} w_1^* \left( t, w_1^0 \left( C_1 \right) \right)$$
$$= -\xi_1 (t) \, \mathbb{E}_Z \left[ \mathbb{E}_{C_2} \left[ \Delta_2^{H*} \left( t, Z, w_3^0 \left( C_2 \right) \right) w_2 \left( t, C_1, C_2 \right) \right] \varphi_1' \left( \left\langle w_1^* \left( t, w_1^0 \left( C_1 \right) \right), X \right\rangle \right) X \right],$$

with initialization $w_1^* \left( 0, w_1^0 \left( C_1 \right) \right) = w_1^0 \left( C_1 \right)$. Substituting the first equation into the second one, we get:

$$\frac{\partial}{\partial t} w_1^* \left( t, w_1^0 \left( C_1 \right) \right) = -\xi_1 (t) \, \mathbb{E}_Z \left[ \mathbb{E}_{C_2} \left[ \Delta_2^{H*} \left( t, Z, w_3^0 \left( C_2 \right) \right) w_2^0 \left( C_1, C_2 \right) \right] \varphi_1' \left( \left\langle w_1^* \left( t, w_1^0 \left( C_1 \right) \right), X \right\rangle \right) X \right]$$
$$+ \xi_1 (t) \int_0^t \xi_2 (s) \, \mathbb{E}_{Z, Z'} \left[ \mathbb{E}_{C_2} \left[ \Delta_2^{H*} \left( t, Z, w_3^0 \left( C_2 \right) \right) \Delta_2^{H*} \left( s, Z', w_3^0 \left( C_2 \right) \right) \right] \right.$$
$$\left. \times \varphi_1 \left( \left\langle w_1^* \left( s, w_1^0 \left( C_1 \right) \right), X' \right\rangle \right) \varphi_1' \left( \left\langle w_1^* \left( t, w_1^0 \left( C_1 \right) \right), X \right\rangle \right) X \right] ds.$$

Note that by an argument similar to Step 3,

$$\mathbb{E}_{C_2} \left[ \Delta_2^{H*} \left( t, Z, w_3^0 \left( C_2 \right) \right) w_2^0 \left( C_1, C_2 \right) \right] = \int \Delta_2^{H*} \left( t, Z, u_3 \right) u_2 \rho^2 \left( du_2 \right) \rho^3 \left( du_3 \right),$$

which holds for all $t \geq 0$ almost surely by the same argument in Step 2. We thus obtain:

$$\frac{\partial}{\partial t} w_1^* \left( t, u_1 \right) = -\xi_1 (t) \int \mathbb{E}_Z \left[ \Delta_2^{H*} \left( t, Z, u_3 \right) u_2 \varphi_1' \left( \left\langle w_1^* \left( t, u_1 \right), X \right\rangle \right) X \right] \rho^2 \left( du_2 \right) \rho^3 \left( du_3 \right)$$
$$+ \xi_1 (t) \int_0^t \xi_2 (s) \, \mathbb{E}_{Z, Z'} \left[ \int \Delta_2^{H*} \left( t, Z, u_3 \right) \Delta_2^{H*} \left( s, Z', u_3 \right) \rho^3 \left( du_3 \right) \right.$$
$$\left. \times \varphi_1 \left( \left\langle w_1^* \left( s, u_1 \right), X' \right\rangle \right) \varphi_1' \left( \left\langle w_1^* \left( t, u_1 \right), X \right\rangle \right) X \right] ds,$$

with initialization $w_1^* \left( t, u_1 \right) = u_1$ for all $u_1 \in \text{supp} \left( \rho^1 \right)$ and $t \geq 0$. $\qquad \square$

An important ingredient of the proof is that the distribution of $w_1(t, C_1)$ has full support at all time $t \geq 0$, even though we only need to assume this property at initialization $t = 0$. This key property is proven by a topology argument, supported by the measurability result of Lemma 16. We remark that a similar property for two-layer networks is established in Chizat & Bach (2018) using a different topology argument.

**Lemma 17.** *Consider the same setting as Theorem 8. For all finite time $t \geq 0$, the support of* Law $(w_1(t, C_1))$ *is* $\mathbb{R}^d$.

*Proof.* By Lemma 16, one can choose a neural embedding such that there exists Borel functions $w_1^*$ and $\Delta_2^{H*}$ for which almost surely, for all $t \geq 0$,

$$w_1(t, C_1) = w_1^*\left(t, w_1^0(C_1)\right),$$
$$\Delta_2^H(z, C_2; W(t)) = \Delta_2^{H*}\left(t, z, w_3^0(C_2)\right),$$

where $W(t)$ is the MF dynamics formed under the coupling procedure with this neuronal embedding as described in Section 3.1. Furthermore,

$$\frac{\partial}{\partial t} w_1^*(t, u_1) = -\int \mathbb{E}_Z\left[\Delta_2^{H*}(t, Z, u_3) u_2 \varphi_1'\left(\langle w_1^*(t, u_1), X\rangle\right) X\right] \rho^2(du_2) \rho^3(du_3)$$

$$+ \int_0^t \mathbb{E}_{Z, Z'}\left[\int \Delta_2^{H*}(t, Z, u_3) \Delta_2^{H*}(s, Z', u_3) \rho^3(du_3)\right.$$

$$\left. \times \varphi_1\left(\langle w_1^*(s, u_1), X'\rangle\right) \varphi_1'\left(\langle w_1^*(t, u_1), X\rangle\right) X\right] ds,$$

with initialization $w_1^*(0, u_1) = u_1$ for all $u_1 \in \operatorname{supp}(\rho^1)$ and $t \geq 0$, where $Z'$ is an independent copy of $Z$. We recall from Lemma 12 that

$$\operatorname{ess-sup}\Delta_2^{H*}\left(t, Z, w_3^0(C_2)\right) = \operatorname{ess-sup}\Delta_2^H(Z, C_2; W(t)) \leq K_t,$$

where $K_t$ denotes a generic constant that depends on $t$ and is finite with finite $t$. Therefore, by Assumption 1, for $t \leq T$ and $u_1, u_1' \in \operatorname{supp}(\rho^1)$,

$$\left|\frac{\partial}{\partial t} w_1^*(t, u_1) - \frac{\partial}{\partial t} w_1^*(t, u_1')\right| \leq K_t |w_1^*(t, u_1) - w_1^*(t, u_1')| + K_t \int_0^t |w_1^*(s, u_1) - w_1^*(s, u_1')| \, ds$$

$$\leq K_T \sup_{s \leq t} |w_1^*(s, u_1) - w_1^*(s, u_1')|,$$

$$\left|\frac{\partial}{\partial t} w_1^*(t, u_1)\right| \leq K_T.$$

Applying Gronwall's lemma to the first bound:

$$\sup_{t \leq T} |w_1^*(t, u_1) - w_1^*(t, u_1')| \leq e^{K_T} |w_1^*(0, u_1) - w_1^*(0, u_1')|$$

$$= e^{K_T} |u_1 - u_1'|.$$

Furthermore the second bound implies

$$\sup_{t, t' \leq T} |w_1^*(t, u_1) - w_1^*(t', u_1)| \leq K_T |t - t'|.$$

Therefore $(t, u_1) \mapsto w_1^*(t, u_1)$ is a continuous mapping on $[0, T] \times \mathbb{R}^d$ for an arbitrary $T \geq 0$.

Given this continuity, we show the thesis by a topology argument. Consider the sphere $\mathbb{S}^d$ which is a compactification of $\mathbb{R}^d$. We can extend $w_1^*$ to a function $M : [0, T] \times \mathbb{S}^d \to \mathbb{S}^d$ fixing the point at infinity, which remains a continuous map since $|M(t, u_1) - u_1| = |M(t, u_1) - M(0, u_1)| \leq K_T t$. Let $M_t : \mathbb{R}^d \to \mathbb{R}^d$ be defined by $M_t(u_1) = M(t, u_1)$. We claim that $M_t$ is surjective for all finite $t$. Indeed, if $M_t$ fails to be surjective for some $t$, then for some $p \in \mathbb{S}^d$, $M_t : \mathbb{S}^d \to \mathbb{S}^d \backslash \{p\} \to \mathbb{S}^d$ is homotopic to the constant map, but $M$ then gives a homotopy from the identity map $M_0$ on the sphere to a constant map, which is a contradiction as the sphere $\mathbb{S}^d$ is not contractible. Hence $w_1^*(t, \cdot)$ is surjective for all finite $t$. Recall that $w_1(t, C_1) = w_1^*\left(t, w_1^0(C_1)\right)$ almost surely and $w_1^0(C_1)$ has full support. Now let us assume that $w_1(t, C_1)$ does not have full support at some

time $t$, which implies there is an open ball $B$ in $\mathbb{R}^d$ for which $\mathbb{P}\left(w_1\left(t, C_1\right) \in B\right) = 0$. Then $\mathbb{P}\left(w_1^*\left(t, w_1^0\left(C_1\right)\right) \in B\right) = 0$. Since $w_1^*\left(t, \cdot\right)$ has full support, there is an open set $U$ such that $w_1^*\left(t, u_1\right) \in B$ for all $u_1 \in U$. Then $\mathbb{P}\left(w_1^0\left(C_1\right) \in U\right) = 0$, contradicting the assumption that $w_1^0\left(C_1\right)$ has full support. Therefore $w_1\left(t, C_1\right)$ must have full support at all $t \geq 0$. $\qquad\square$

With this, we are ready to prove Theorem 8.

*Proof of Theorem 8.* Recall, by Theorem 1, the solution to the MF ODEs exists uniquely, and by Lemma 17, the support of Law $\left(w_1\left(t, C_1\right)\right)$ is $\mathbb{R}^d$ at all $t$. By the convergence assumption, we have that for any $\epsilon > 0$, there exists $T\left(\epsilon\right)$ such that for all $t \geq T\left(\epsilon\right)$ and $P$-almost every $c_1$:

$$\mathbb{E}_{C_2}\left[\left|\mathbb{E}_Z\left[\Delta_2^H\left(Z, C_2; W\left(t\right)\right)\varphi_1\left(\langle w_1\left(t, c_1\right), X\rangle\right)\right]\right|\right] \leq \epsilon.$$

Since Law $\left(w_1\left(t, C_1\right)\right)$ has full support, we obtain that for $u$ in a dense subset of $\mathbb{R}^d$,

$$\mathbb{E}_{C_2}\left[\left|\mathbb{E}_Z\left[\Delta_2^H\left(Z, C_2; W\left(t\right)\right)\varphi_1\left(\langle u, X\rangle\right)\right]\right|\right] \leq \epsilon.$$

By continuity of $u \mapsto \varphi_1(\langle u, x\rangle)$, we extend the above to all $u \in \mathbb{R}^d$. Since $\varphi_1$ is bounded,

$$\mathbb{E}_{C_2}\left[\left|\mathbb{E}_Z\left[\left(\Delta_2^H\left(Z, C_2; W\left(t\right)\right) - \Delta_2^H\left(Z, C_2; \bar{w}_1, \bar{w}_2, \bar{w}_3\right)\right)\varphi_1\left(\langle u, X\rangle\right)\right]\right|\right]$$
$$\leq K\mathbb{E}\left[\left|\Delta_2^H\left(Z, C_2; W\left(t\right)\right) - \Delta_2^H\left(Z, C_2; \bar{w}_1, \bar{w}_2, \bar{w}_3\right)\right|\right]$$
$$\leq K\mathbb{E}\Big[\left(1 + |\bar{w}_3(C_2)|\right)\Big(\left|w_3(t, C_2) - \bar{w}_3(C_2)\right| + |\bar{w}_3(C_2)|\left|w_2(t, C_1, C_2) - \bar{w}_2(C_1, C_2)\right|$$
$$+ |\bar{w}_3(C_2)|\left|\bar{w}_2(C_1, C_2)\right|\left|w_1(t, C_1) - \bar{w}_1(C_1)\right|\Big)\Big],$$

where the last step is by Assumption 1. Recall that the right-hand side converges to 0 as $t \to \infty$. We thus obtain that for all $u \in \mathbb{R}^d$,

$$\mathbb{E}_{C_2}\left[\left|\left\langle\mathbb{E}_Z\left[\Delta_2^H\left(Z, C_2; \bar{w}_1, \bar{w}_2, \bar{w}_3\right)|X = x\right], \varphi_1\left(\langle u, x\rangle\right)\right\rangle_{L^2(\mathcal{P}_X)}\right|\right]$$
$$= \mathbb{E}_{C_2}\left[\left|\mathbb{E}_Z\left[\Delta_2^H\left(Z, C_2; \bar{w}_1, \bar{w}_2, \bar{w}_3\right)\varphi_1\left(\langle u, X\rangle\right)\right]\right|\right]$$
$$= 0,$$

which yields that for all $u \in \mathbb{R}^d$ and $P$-almost every $c_2$,

$$\left|\left\langle\mathbb{E}_Z\left[\Delta_2^H\left(Z, c_2; \bar{w}_1, \bar{w}_2, \bar{w}_3\right)|X = x\right], \varphi_1\left(\langle u, x\rangle\right)\right\rangle_{L^2(\mathcal{P}_X)}\right| = 0.$$

Here we note that by Assumption 1,

$$\left|\mathbb{E}_Z\left[\Delta_2^H\left(Z, c_2; \bar{w}_1, \bar{w}_2, \bar{w}_3\right)|X = x\right]\right| \leq K\left|\bar{w}_3\left(c_2\right)\right|,$$

and so $\mathbb{E}_Z\left[\Delta_2^H\left(Z, c_2; \bar{w}_1, \bar{w}_2, \bar{w}_3\right)|X = x\right]$ is in $L^2\left(\mathcal{P}_X\right)$ for $P$-almost every $c_2$. Since $\left\{\varphi_1\left(\langle u, \cdot\rangle\right): u \in \mathbb{R}^d\right\}$ has dense span in $L^2\left(\mathcal{P}_X\right)$, we have $\mathbb{E}_Z\left[\Delta_2^H\left(Z, c_2; \bar{w}_1, \bar{w}_2, \bar{w}_3\right)|X = x\right] = 0$ for $\mathcal{P}_X$-almost every $x$ and $P$-almost every $c_2$, and hence

$$\mathbb{E}_Z\left[\partial_2\mathcal{L}\left(Y, \hat{y}\left(X; \bar{w}_1, \bar{w}_2, \bar{w}_3\right)\right)|X = x\right]\varphi_3'\left(H_3\left(x; \bar{w}_1, \bar{w}_2, \bar{w}_3\right)\right)\bar{w}_3\left(c_2\right)\varphi_2'\left(H_2\left(x, c_2; \bar{w}_1, \bar{w}_2\right)\right) = 0.$$

We note that our assumptions guarantee that $\mathbb{P}\left(\bar{w}_3\left(C_2\right) \neq 0\right)$ is positive. Indeed:

- In the case $w_3^0\left(C_2\right) \neq 0$ with positive probability and $\xi_3\left(\cdot\right) = 0$, the conclusion is obvious.

- In the case $\mathscr{L}\left(w_1^0, w_2^0, w_3^0\right) < \mathbb{E}_Z\left[\mathcal{L}\left(Y, \varphi_3\left(0\right)\right)\right]$, we recall the following standard property of gradient flows:

$$\mathscr{L}\left(w_1\left(t, \cdot\right), w_2\left(t, \cdot, \cdot\right), w_3\left(t, \cdot\right)\right) \leq \mathscr{L}\left(w_1\left(t', \cdot\right), w_2\left(t', \cdot, \cdot\right), w_3\left(t', \cdot\right)\right),$$

for $t \geq t'$. In particular, setting $t' = 0$ and taking $t \to \infty$, it is easy to see that

$$\mathscr{L}\left(\bar{w}_1, \bar{w}_2, \bar{w}_3\right) \leq \mathscr{L}\left(w_1^0, w_2^0, w_3^0\right) < \mathbb{E}_Z\left[\mathcal{L}\left(Y, \varphi_3\left(0\right)\right)\right].$$

If $\mathbb{P}\left(\bar{w}_3\left(C_2\right) = 0\right) = 1$ then $\mathscr{L}\left(\bar{w}_1, \bar{w}_2, \bar{w}_3\right) = \mathbb{E}_Z\left[\mathcal{L}\left(Y, \varphi_3\left(0\right)\right)\right]$, a contradiction.

Then since $\varphi_2'$ and $\varphi_3'$ are strictly non-zero, we have $\mathbb{E}_Z\left[\partial_2\mathcal{L}\left(Y,\hat{y}\left(X;\bar{w}_1,\bar{w}_2,\bar{w}_3\right)\right)|X=x\right]=0$ for $\mathcal{P}_X$-almost every $x$.

In Case 1, since $\mathcal{L}$ convex in the second variable, for any measurable function $\tilde{y}(x)$,

$$\mathcal{L}\left(y,\tilde{y}\left(x\right)\right)-\mathcal{L}\left(y,\hat{y}\left(x;\bar{w}_1,\bar{w}_2,\bar{w}_3\right)\right)\geq\partial_2\mathcal{L}\left(y,\hat{y}\left(x;\bar{w}_1,\bar{w}_2,\bar{w}_3\right)\right)\left(\tilde{y}\left(x\right)-\hat{y}\left(x;\bar{w}_1,\bar{w}_2,\bar{w}_3\right)\right).$$

Taking expectation, we get $\mathbb{E}_Z\left[\mathcal{L}\left(Y,\tilde{y}\left(X\right)\right)\right]\geq\mathscr{L}\left(\bar{w}_1,\bar{w}_2,\bar{w}_3\right)$, i.e. $\left(\bar{w}_1,\bar{w}_2,\bar{w}_3\right)$ is a global minimizer of $\mathscr{L}$.

In Case 2, since $y$ is a function of $x$, we obtain $\partial_2\mathcal{L}\left(y,\hat{y}\left(x;\bar{w}_1,\bar{w}_2,\bar{w}_3\right)\right)=0$ and hence $\mathcal{L}\left(y,\hat{y}\left(x;\bar{w}_1,\bar{w}_2,\bar{w}_3\right)\right)=0$ for $\mathcal{P}_X$-almost every $x$.

Finally we have from Assumptions 1, 3:

$$\begin{aligned}\left|\mathscr{L}\left(W\left(t\right)\right)-\mathscr{L}\left(\bar{w}_1,\bar{w}_2,\bar{w}_3\right)\right|&=\left|\mathbb{E}_Z\left[\mathcal{L}\left(Y,\hat{y}\left(X;W\left(t\right)\right)\right)-\mathcal{L}\left(Y,\hat{y}\left(X;\bar{w}_1,\bar{w}_2,\bar{w}_3\right)\right)\right]\right|\\&\leq K\mathbb{E}_Z\left[\left|\hat{y}\left(X;W\left(t\right)\right)-\hat{y}\left(X;\bar{w}_1,\bar{w}_2,\bar{w}_3\right)\right|\right]\\&\leq K\mathbb{E}\Big[\left|w_3\left(t,C_2\right)-\bar{w}_3\left(C_2\right)\right|+\left|\bar{w}_3\left(C_2\right)\right|\left|w_2\left(t,C_1,C_2\right)-\bar{w}_2\left(C_1,C_2\right)\right|\\&\quad+\left|\bar{w}_3\left(C_2\right)\right|\left|\bar{w}_2\left(C_1,C_2\right)\right|\left|w_1\left(t,C_1\right)-\bar{w}_1\left(C_1\right)\right|\Big]\end{aligned}$$

which tends to $0$ as $t\to\infty$. This completes the proof. $\qquad\square$

## E   CONVERSE FOR GLOBAL CONVERGENCE: REMARK 9

We prove a converse statement for global convergence in relation with the essential supremum condition (6).

**Proposition 18.** *Consider a neuronal embedding $\left(\Omega,\mathcal{F},P,\left\{w_i^0\right\}_{i=1,2,3}\right)$ of $\left(\rho^1,\rho^2,\rho^3\right)$-i.i.d. initialization. Consider the MF limit corresponding to the network (1), such that they are coupled together by the coupling procedure in Section 3.1, under Assumptions 1, 2, $\xi_1\left(\cdot\right)=\xi_2\left(\cdot\right)=1$. Assume that $\mathcal{L}(y,\hat{y})\to\infty$ as $|\hat{y}|\to\infty$ for each $y$. Further assume that there exists $\bar{w}_3$ such that as $t\to\infty$,*

$$\mathbb{E}_{C_2}\left[|w_3(t,C_2)-\bar{w}_3(C_2)|\right]\to0.$$

*Then the following hold:*

- *Case 1 (convex loss): If $\mathcal{L}$ is convex in the second variable and*

$$\lim_{t\to\infty}\mathscr{L}\left(W\left(t\right)\right)=\inf_V\mathscr{L}\left(V\right),$$

   *then it must be that*

$$\sup_{c_1\in\Omega_1}\mathbb{E}_{C_2}\left[\left|\frac{\partial}{\partial t}w_2\left(t,c_1,C_2\right)\right|\right]\to0\quad\text{as }t\to\infty.$$

- *Case 2 (generic non-negative loss): Suppose that $\partial_2\mathcal{L}\left(y,\hat{y}\right)=0$ implies $\mathcal{L}\left(y,\hat{y}\right)=0$, and $y=y(x)$ is a function of $x$. If $\mathscr{L}\left(W\left(t\right)\right)\to0$ as $t\to\infty$, then the same conclusion also holds.*

*Proof.* We recall

$$\begin{aligned}\frac{\partial}{\partial t}w_2\left(t,c_1,c_2\right)=-\mathbb{E}_Z\Big[&\partial_2\mathcal{L}\left(Y,\hat{y}\left(X;W\left(t\right)\right)\right)w_3\left(t,c_2\right)\\&\times\varphi_3'\left(H_3\left(X;W\left(t\right)\right)\right)\varphi_2'\left(H_2\left(X,c_2;W\left(t\right)\right)\right)\varphi_1\left(\langle w_1\left(t,c_1\right),X\rangle\right)\Big],\end{aligned}$$

for $c_1\in\Omega_1$, $c_2\in\Omega_2$. By Assumption 1,

$$\left|\frac{\partial}{\partial t}w_2\left(t,c_1,c_2\right)\right|\leq K\mathbb{E}_Z\left[\left|\partial_2\mathcal{L}\left(Y,\hat{y}\left(X;W\left(t\right)\right)\right)\right|\right]\left|w_3\left(t,c_2\right)\right|.$$

Note that the right-hand side is independent of $c_1$. Since $\mathbb{E}_{C_2}\left[\|w_3(t, C_2) - \bar{w}_3(C_2)\|\right] \to 0$ as $t \to \infty$, we have for some finite $t_0 \le K$,

$$\mathbb{E}_{C_2}\left[\|\bar{w}_3(C_2)\|\right] \le \mathbb{E}_{C_2}\left[\|w_3(t_0, C_2)\|\right] + K \le K,$$

where the last step is by Lemma 11 and Assumption 2. As such, for all $t$ sufficiently large, we have:

$$\sup_{c_1 \in \Omega_1} \mathbb{E}_{C_2}\left[\left\|\frac{\partial}{\partial t}w_2(t, c_1, C_2)\right\|\right] \le K\mathbb{E}_Z\left[\|\partial_2\mathcal{L}(Y, \hat{y}(X; W(t)))\|\right]\mathbb{E}_{C_2}\left[\|w_3(t, C_2)\|\right]$$
$$\le K\mathbb{E}_Z\left[\|\partial_2\mathcal{L}(Y, \hat{y}(X; W(t)))\|\right](K + \mathbb{E}_{C_2}\left[\|\bar{w}_3(C_2)\|\right])$$
$$\le K\mathbb{E}_Z\left[\|\partial_2\mathcal{L}(Y, \hat{y}(X; W(t)))\|\right].$$

The proof concludes once we show that $\mathbb{E}_Z\left[\|\partial_2\mathcal{L}(Y, \hat{y}(X; W(t)))\|\right] \to 0$ as $t \to \infty$.

For a fixed $z = (x, y)$, let us write $\mathcal{L}(t, z) = \mathcal{L}(y, \hat{y}(x; W(t)))$ and $\partial_2\mathcal{L}(t, z) = \partial_2\mathcal{L}(y, \hat{y}(x; W(t)))$ for brevity. Consider Case 1. We claim that if there is an increasing sequence of time $t_i$ so that $\lim_{i \to \infty}\left[\mathcal{L}(t_i, z) - \inf_{\hat{y}}\mathcal{L}(y, \hat{y})\right] = 0$, then $\lim_{i \to \infty}|\partial_2\mathcal{L}(t_i, z)| = 0$. Indeed, it suffices to show that for any subsequence $t_{i_j}$ of $t_i$, there exists a further subsequence $t_{i_{j_k}}$ such that $\lim_{k \to \infty}\left|\partial_2\mathcal{L}(t_{i_{j_k}}, z)\right| = 0$. In any subsequence $t_{i_j}$ of $t_i$, using that $\mathcal{L}(t_{i_j}, z)$ is convergent and the fact $\mathcal{L}(y, \hat{y}) \to \infty$ as $|\hat{y}| \to \infty$, we have $\hat{y}(x; W(t_{i_j}))$ is bounded. Hence, we obtain a subsequence $t_{i_{j_k}}$ for which $\hat{y}(x; W(t_{i_{j_k}}))$ converges to some limit $\hat{y}^*$. By continuity, we have $\mathcal{L}(y, \hat{y}^*) = \lim_{k \to \infty}\mathcal{L}(t_{i_{j_k}}, z) = \inf_{\hat{y}}\mathcal{L}(y, \hat{y})$. Thus, since $\mathcal{L}$ is convex in the second variable, we have $\partial_2\mathcal{L}(y, \hat{y}^*) = 0$. Thus, $\lim_{k \to \infty}\left|\partial_2\mathcal{L}(t_{i_{j_k}}, z)\right| = |\partial_2\mathcal{L}(y, \hat{y}^*)| = 0$, as claimed. Similarly, we obtain in Case 2 that if there is an increasing sequence of time $t_i$ so that $\lim_{i \to \infty}\left[\mathcal{L}(t_i, z)\right] = 0$, then $\lim_{i \to \infty}|\partial_2\mathcal{L}(t_i, z)| = 0$.

To show that $\mathbb{E}_Z\left[\|\partial_2\mathcal{L}(t, Z)\|\right] \to 0$ as $t \to \infty$, it suffices to show that for any increasing sequence of times $t_i$ tending to infinity, there exists a subsequence $t_{i_j}$ of $t_i$ such that $\mathbb{E}_Z\left[\left\|\partial_2\mathcal{L}\left(t_{i_j}, Z\right)\right\|\right] \to 0$. In Case 1, we have $\lim_{i \to \infty}\mathscr{L}(W(t_i)) = \inf_V\mathscr{L}(V)$, so $\lim_{i \to \infty}\mathbb{E}_Z\left[\mathcal{L}(t_i, Z) - \inf_{\hat{Y}}\mathcal{L}(Y, \hat{Y})\right] = 0$. Since $\mathcal{L}(t_i, Z) - \inf_{\hat{Y}}\mathcal{L}(Y, \hat{Y})$ is nonnegative, this implies that $\mathcal{L}(t_i, Z) - \inf_{\hat{Y}}\mathcal{L}(Y, \hat{Y})$ converges to $0$ in probability. Thus, there is a further subsequence $t_{i_j}$ for which $\mathcal{L}\left(t_{i_j}, Z\right) - \inf_{\hat{Y}}\mathcal{L}(Y, \hat{Y})$ converges to $0$ $\mathcal{P}$-almost surely. By the previous claim, $\left|\partial_2\mathcal{L}\left(t_{i_j}, Z\right)\right|$ converges to $0$ $\mathcal{P}$-almost surely. Since $\left|\partial_2\mathcal{L}\left(t_{i_j}, Z\right)\right|$ is bounded $\mathcal{P}$-almost surely, we obtain that $\mathbb{E}_Z\left[\left\|\partial_2\mathcal{L}\left(t_{i_j}, Z\right)\right\|\right] \to 0$ from the bounded convergence theorem. The result in Case 2 can be established similarly. $\square$

# F  USEFUL TOOLS

We first present a useful concentration result. In fact, the tail bound can be improved using the argument in Feldman & Vondrak (2018), but the following simpler version is sufficient for our purposes.

**Lemma 19.** *Consider an integer $n \ge 1$ and let $x$, $c_1$, ..., $c_n$ be mutually independent random variables. Let $\mathbb{E}_x$ and $\mathbb{E}_c$ denote the expectations w.r.t. $x$ only and $\{c_i\}_{i \in [n]}$ only, respectively. Consider a collection of mappings $\{f_i\}_{i \in [n]}$, which map to a separable Hilbert space $\mathbb{F}$. Let $f_i(x) = \mathbb{E}_c\left[f_i(c_i, x)\right]$. Assume that for some $R > 0$, $|f_i(c_i, x) - f_i(x)| \le R$ almost surely, then for any $\delta > 0$,*

$$\mathbb{P}\left(\left\|\mathbb{E}_x\left[\left\|\frac{1}{n}\sum_{i=1}^n f_i(c_i, x) - f_i(x)\right\|\right]\right\| \ge \delta\right) \le \frac{8R}{\sqrt{n}\delta}\exp\left(-\frac{n\delta^2}{8R^2}\right) \le \frac{8R}{\delta}\exp\left(-\frac{n\delta^2}{8R^2}\right).$$

*Proof.* For brevity, let us define

$$Z_n(x) = \sum_{i=1}^n\left(f_i(c_i, x) - f_i(x)\right).$$

By Theorem 21,

$$\mathbb{P}\left(|Z_n(x)| \geq n\delta|x\right) \leq 2\exp\left(-n\delta^2/\left(4R^2\right)\right),$$

and therefore,

$$\mathbb{P}\left(|Z_n(x)| \geq n\delta\right) \leq 2\exp\left(-n\delta^2/\left(4R^2\right)\right),$$

since the right-hand side is uniform in $x$. Next note that, w.r.t. the randomness of $x$ only,

$$\mathbb{E}_x\left[|Z_n(x)|\right] = \mathbb{E}_x\left[|Z_n(x)|\,\mathbb{I}\left(|Z_n(x)| \geq n\delta/2\right)\right] + \mathbb{E}_x\left[|Z_n(x)|\,\mathbb{I}\left(|Z_n(x)| < n\delta/2\right)\right]$$
$$\leq \mathbb{E}_x\left[|Z_n(x)|\,\mathbb{I}\left(|Z_n(x)| \geq n\delta/2\right)\right] + n\delta/2.$$

As such, by Markov's inequality and Cauchy-Schwarz's inequality,

$$\mathbb{P}\left(\mathbb{E}_x\left[|Z_n(x)|\right] \geq n\delta\right) \leq \mathbb{P}\left(\mathbb{E}_x\left[|Z_n(x)|\,\mathbb{I}\left(|Z_n(x)| \geq n\delta/2\right)\right] \geq n\delta/2\right)$$
$$\leq \frac{2}{n\delta}\mathbb{E}\left[|Z_n(x)|\,\mathbb{I}\left(|Z_n(x)| \geq n\delta/2\right)\right]$$
$$\leq \frac{2}{n\delta}\mathbb{E}\left[|Z_n(x)|^2\right]^{1/2}\mathbb{P}\left(|Z_n(x)| \geq n\delta/2\right)^{1/2}$$
$$\leq \frac{4}{n\delta}\mathbb{E}\left[|Z_n(x)|^2\right]^{1/2}\exp\left(-\frac{n\delta^2}{8R^2}\right).$$

Notice that since $c_1, ..., c_n$ are independent and $f_i(x) = \mathbb{E}_c\left[f_i(c_i, x)\right]$,

$$\mathbb{E}\left[|Z_n(x)|^2\right] = \sum_{i=1}^n \mathbb{E}\left[|f_i(c_i, x) - f_i(x)|^2\right] \leq 2nR^2.$$

We thus get:

$$\mathbb{P}\left(\mathbb{E}_x\left[|Z_n(x)|\right] \geq n\delta\right) \leq \frac{8R}{\sqrt{n}\delta}\exp\left(-\frac{n\delta^2}{8R^2}\right).$$

This proves the claim. $\qquad\square$

We state a martingale concentration result, which is a special case of (Pinelis, 1994, Theorem 3.5) which applies to a more general Banach space.

**Theorem 20** (Concentration of martingales in Hilbert spaces.)**.** *Consider a martingale $Z_n \in \mathbb{Z}$ a separable Hilbert space such that $|Z_n - Z_{n-1}| \leq R$ and $Z_0 = 0$. Then for any $t > 0$,*

$$\mathbb{P}\left(\max_{k \leq n}|Z_k| \geq t\right) \leq 2\inf_{\lambda > 0}\exp\left(-\lambda t + \text{ess-sup}\sum_{k=1}^n \mathbb{E}\left[e^{\lambda|Z_k - Z_{k-1}|} - 1 - \lambda|Z_k - Z_{k-1}| \mid \mathcal{F}_{k-1}\right]\right).$$

*In particular, for any $\delta > 0$,*

$$\mathbb{P}\left(\max_{k \leq n}|Z_k| \geq n\delta\right) \leq 2\exp\left(-\frac{n\delta^2}{2R^2}\right).$$

The following concentration result for i.i.d. random variables in Hilbert spaces is a corollary.

**Theorem 21** (Concentration of i.i.d. sum in Hilbert spaces.)**.** *Consider $n$ i.i.d. random variables $X_1, ..., X_n$ in a separable Hilbert space. Suppose that there exists a constant $R > 0$ such that $|X_i - \mathbb{E}[X_i]| \leq R$ almost surely. Then for any $\delta > 0$,*

$$\mathbb{P}\left(\frac{1}{n}\left|\sum_{i=1}^n X_i - \mathbb{E}[X_i]\right| \geq \delta\right) \leq 2\exp\left(-\frac{n\delta^2}{2R^2}\right).$$

