# OpenReview forum: "Global Convergence of Three-layer Neural Networks in the Mean Field Regime"
_ICLR.cc/2021/Conference — ICLR 2021 Oral_

### Official Review · AnonReviewer4 · 2020-10-24
**Overall, this paper is thought to provide a promising idea to analyze not only three-layer NNs but deep NNs.**

**Rating:** 7
**Confidence:** 3

**Review:**

This paper studies some theoretical properties of three-layer neural networks (NNs) under the mean-field (MF) regime. The authors proposed neuronal embedding in order to study large-width neural networks. Then, the quantitative relation between finite-width NN and the MF limit was clarified. The global convergence of the continuous limit of the stochastic gradient descent (SGD) was proved without assuming the convexity of the loss function. The global convergence of the MF limit is used to establish the optimization efficiency of the neural network with SGD.

The problem considered in this paper is important. The definitions and the statement of theorems are clearly described. In order to prove the global convergence, the authors assumed the uniform approximation property rather than the convexity of the loss function. This approach is interesting. Though I'm not very familiar with the MF regime, the high-level idea to prove the theorem is well-written. I read some proofs in the appendix, and I found that the description is accessible to a wide range of audiences. Overall, this paper is thought to provide a promising idea to analyze not only three-layer NNs but deep NNs.

Some comments are shown below:
- Neuronal ensemble is introduced to analyze the dynamics of the MF limit. Apart from the MF regime, is there any similar idea of the neuronal ensemble? Showing some references would be beneficial to readers.
- The constant K appears in the upper bound in Theorem 3. Showing a more concrete expression of K is informative. What is the typical K in this case?
- In Theorem 8 and Corollary 10, the global convergence was proved. If the convexity was also assumed in addition to the assumptions in the theorems, is it possible to derive the convergence rate?
- In the paper, the global convergence of three-layer NNs was analyzed. What is the main obstacle to investigating the global convergence of multi-layer NNs or deep NNs according to the idea proposed in this paper?

---

> ### Author Response · Authors · 2020-11-18
> **Response to Reviewer 4**
>
> We thank the reviewer for the thoughtful review.
>
> Regarding the idea of neuronal ensemble / neuronal embedding, we are not aware of similar ideas in the context of neural networks. Perhaps there are some distantly related ideas from other fields, such as limits of combinatorial objects (e.g. graphon, hypergraphon, permuton) in probabilistic combinatorics. All these ideas take a continuumization perspective, but ultimately serve very different purposes and assume different forms.
>
> Regarding the constant K in Theorem 3, it is dependent on other constants that appear elsewhere, e.g. the bound on the magnitude of the first layer’s activation or the absolute constants in the concentration inequalities. We are not able to come up with a precise answer as such, except to say that one should think K = O(1) morally. It is actually an interesting and challenging question to understand these deviation bounds precisely.
>
> Regarding convergence rate, we believe that convexity of the loss function is generally insufficient to establish a convergence rate. This is already evident from the 2-layer case; for example, Javanmard et al. (2019) derives a convergence rate for 2-layer nets via a notion that cannot be attained by merely assuming (strongly) convex losses. This notion is a grand idea in the studies of Wasserstein gradient flow of the past two decades, so it’s highly non-trivial. In general, the question of convergence rate is a difficult one, and we hope to find new insights in the future.
>
> Regarding global convergence of multilayer nets, it requires new ideas on top of the insights already described in the 3-layer case. The conceptual obstacle is the following. As mentioned in Section 5, i.i.d. initialization will cause the mean field limit to be degenerate when there are more than 3 layers. This degeneracy prevents one to generally have universal approximation hold at any finite training time.

---

### Official Review · AnonReviewer2 · 2020-10-28
**Nice theoretical result. Relation to previous work to be precised**

**Rating:** 7
**Confidence:** 2

**Review:**

This paper studies the behavior of a 3-layer fully connected network when the width of the network is large. The authors define a mean field regime and prove that the behavior of the network under stochastic gradient descent converges to this mean field regime (for any finite time horizon).  This result complements nicely previous works like (Nguyen 2019) that contain an informal derivation of the mean field regime. This transient regime is complemented by a long-term analysis under quite restrictive assumptions (which imply essentially than the mean field regime always converge to the minimizer of the loss function).

I did not check all details of the proof but the approach seems mathematically sound. Once the model is defined, the proof for the finite regime is relatively classical: it relies on Martingale concentration plus Gronwall's lemma. Yet, as always, the devil being in the details and defining the right model and using the right notations is a difficult task. The result of the stationnary regime seems also reasonable but I must admit that the proof of the infinite horizon is not really clear to me. I would have appreciated more pedagogical effort from the authors.

To summarize, the paper seems a nice theoretical contribution. Yet, to me one thing that this paper is missing is an explanation or illustration of how useful is their result to understand the behavior of deep neural networks.

That being said, one major concern that I have about the paper is the link with https://arxiv.org/pdf/2001.11443.pdf The arXiv paper considers a very similar model (but more general as it considers L layers instead of 3). It uses almost the exact same notations and the same structure (overall paper and proofs). I think that the authors should clarify the link between the two papers. Also, if I can admit that the present paper is a resubmission of the arXiv paper, I do not understand why does the current paper focus on 3 layers and not the more general model of the arXiv paper.

---

> ### Author Response · Authors · 2020-11-18
> **Response to Reviewer 2**
>
> We thank the reviewer for the detailed and constructive comments on our submission.
>
> Regarding the proof of the global convergence result, we would appreciate it if the reviewer can point to any specific unclear point in Section 4.3 on the high-level idea of the proof. That way we hope to make our writing more accessible.
>
> Regarding the understanding of deep neural networks, we think that this is a grand objective and our work wishes to make a small step. That is, firstly we build a rigorous framework that facilitates the understanding of the mean field limit, and secondly we demonstrate one such application by proving global convergence. Our global convergence result emphasizes on an insight that connects universal approximation and global convergence. While universal approximation is a natural property of infinite-width neural nets, there has been no clear connection with the optimization dynamics and optimization efficiency in the literature. Our insight says if the neural net is designed in such a way that universal approximation is guaranteed to propagate through time, global convergence can be obtained even at the absence of a convex loss.
>
> Regarding the focus on 3 layers instead of more general ones, we think that the 3-layer case serves as a key stepping stone when one transitions from the shallow case to a multilayer one. In particular, the idea of neuronal embedding for 3-layer fully-connected nets with SGD naturally extends to multilayer nets that are not necessarily of the fully-connected type and are trained with more general stochastic algorithms than SGD. Similarly the insight on the universal approximation for global convergence is also a key element for global convergence in the multilayer case. We think these ideas are already quite substantial for one to consume at the first reading, especially in a conference venue. On the other hand, the extension to general multilayer case requires novel ideas on top of the ideas in the 3-layer case, as well as significantly more technicality and more cumbersome notations that may hinder the conveyance of core ideas at the first reading.

---

### Official Review · AnonReviewer1 · 2020-10-28
**Good results**

**Rating:** 7
**Confidence:** 3

**Review:**

**Summary**:
Analysis of neural networks in the mean field regime gains more and more attention as it helps to study the dynamics in the wide regime. The paper extends the recent studies and provides global convergence guarantees for an unregularized feedforward three-layer NN. This is the first time global convergence is established for neural networks of more than two layers in the mean-field regime.

I find the writing a bit chaotic and overdosed with notations. But, overall, I think the results are significant and will help to extend the line of research in the mean-field regime applied to neural networks.

**Questions**:
- In the paper, it is said several times that the convergence result “does not rely critically on convexity”. What do you mean by “critically”? You still assume the convexity. Do you mean that it can be easily relaxed in future works? I think it should be better stated throughout the paper.

**Minor suggestions**:
- Mean field -> mean-field
- Section 2.1 “k” is introduced at the very end. Maybe it would be better to add in the beginning, e.g. “the following network at time k”?
- I would say “W(k) consist of the weights …” instead of “W(k) is the weight with ….”
- It perturbs me a bit that the difference between NN notations and MF is its boldness. Boldness usually means a vector while in authors’ notations $\mathbf{w}_2$ is an element. But I guess with superscripts or symbols like $\hat w$ it would be heavier in notations...
- Section 2.2. I would like to see the description of \Omega, F and P  in the beginning. Like “Given a NN \Omega_i would be a space  of…”. Just to connect it from the start and to easify the following read.
- Definition 2. “The following hold” -> holds
- Section 4.2 “where V a set…” -> is
- “Helps avoiding” -> helps to avoid

---

> ### Author Response · Authors · 2020-11-18
> **Response to Reviewer 1**
>
> We thank the reviewer for detailed reading and comments on our submission.
>
> Regarding the statement that our global convergence result does not rely critically on convexity, we mean two things. Firstly, in Theorem 8, when the target Y is a function of X, we do not need a convex loss to establish global convergence. Secondly, in Section 4.3, our high-level plan of the proof demonstrates that we make no use of convexity-inspired properties in the key steps of the proof. The message is rather than the key lies in a universal approximation property, which holds at any finite training time. We have added some more details in the text after Theorem 8 to clarify this.
>
> We will incorporate several of your suggestions on the writing in our revision, and thank you for pointing these out. We hope that will make the paper easier to follow. Regarding the description of \Omega, F and P in Section 2.2, we allow them to be arbitrary probability spaces, and define the neuronal embedding and MF limit as a general class of objects. These objects are a priori not directly connected with finite-width neural nets. We find that the presentation is clearer by intentionally avoiding making this connection explicit in Section 2.2 where we define the neuronal embedding and MF limit, and delaying the connection to Section 3.

---

### Official Review · AnonReviewer3 · 2020-11-01
**ICLR review for "Global Convergence of Three-layer Neural Networks in the Mean Field Regime"**

**Rating:** 9
**Confidence:** 2

**Review:**

This article is concerned with convergence guarantees of online stochastic gradient descent for a rather generic class of three layers neural networks (instead of similar analyses that treated two layers). The main results state that in a proper limit of infinite width + vanishing learning rate, the dynamics of online SGD is proven to be tracked thanks a mean-field description in the form of coupled ordinary differential equations. Once this mean-field description at disposal, the main result is obtained: in the infinite width + vanishing learning rate + infinite time (= number of training samples), the generalization error tends to it minimal value for a broad class of models and losses (not necessarily convex, which is a novelty of the work) as well as generic data distribution.

Overall this paper is very well written, enjoyable to read despite the technicality of the results, and understandable even for non-specialists of this line of works (like myself). I did not check the appendices and proofs. In the main part there are no typos, and I have no main concerns to bring about. Yet: I would find interesting to know more details about the differences with the refs Nguyen (2019); Araújo et al. (2019); Sirignano & Spiliopoulos (2019); that is not clear.  Also I would find useful to have some hints about the meaning of the (trained third layer) hypothesis in Theorem 8. Finally I find a bit surprising that there are nor restrictions whatsoever on the data distribution (or I missed that). The authors may comment on that in the final version.

I recommend publication. Even if I'm not a specialist, it is obvious that the authors made a big effort of redaction, that the results are very solid, the proof technique seems original and requires less assumptions than previous works (I liked very much the "idea of proof" part). I have very few doubts about the quality of the paper despite I did not read the proof details, and the fact that I'm not aware of the literature in this specific field.

---

> ### Author Response · Authors · 2020-11-18
> **Response to Reviewer 3**
>
> We thank the reviewer for an encouraging review.
>
> We note the difference with the references Nguyen (2019); Araujo et al. (2019) and Sirignano & Spiliopoulos (2019):
> - On the technical side, our neuronal embedding idea allows to describe the MF phenomenon in a clean manner. This helps avoid extra assumptions in Araujo et al. (2019), which assumes untrained first and last layers and requires non-trivial technical tools, and in Sirignano & Spiliopoulos (2019), which requires unnatural sequential limits of the widths and obtains only non-quantitative results.
> - The fundamental reason for the above advantage of the neuronal embedding framework is the following. In Araujo et al. (2019) and Sirignano & Spiliopoulos (2019), their mean field (MF) formulations are specific to and exploit properties of i.i.d. initializations. Remarkably with i.i.d. initializations, when there are more than 3 layers, the MF limit degenerates into a simplified form. Our neuronal embedding idea, on the contrary, avoids this simplification, describes the MF limit in the general non-degenerate form and allows non-i.i.d. initialization.
> - In this aspect, Nguyen (2019) is possibly closer to our spirit: the formulation there is also not specific to i.i.d. initialization. In fact, we take the inspiration of an “unsimplified” MF limit from that work as a key qualification when building our framework. On the other hand, Nguyen (2019)’s proposal is a heuristic formalism and does not prove global convergence. Comparing our work and Nguyen (2019), the MF formulations are constructed under two different perspectives; there is no immediate relationship between the two.
> - Likewise our neuronal embedding idea is also distant from ideas in Araujo et al. (2019), which describes the MF limit via paths over weights, and Sirignano & Spiliopoulos (2019), which describes the MF limit via time-dependent functions of the initialization and simplifies these functions upon the randomness of i.i.d. initialization.
>
> We will expand our discussion in Section 5 accordingly. In the extension to multilayer case, it turns out that the ability to support non-i.i.d. initialization and non-degenerate MF limit is crucial to obtain global convergence when there are more than 3 layers. In this sense, our ICLR submission on the 3-layer case presents the core of the neuronal embedding idea (as well as several other elements for global convergence). These form the foundation for the extension to more than 3 layers, to be reported in another paper.
>
> Regarding the condition in the trained third layer case in Theorem 8, it assumes a sensible choice of initialization, such that the loss at initialization is strictly less than the loss of a trivial estimator $\varphi_3(0)$. When $\varphi_3$ is a linear activation, it is the trivial estimator $0$. This is a mild assumption. The role of this assumption is to avoid the undesirable scenario where the third layer converges to zero.
>
> Regarding the assumptions on the data distribution, we indeed allow general data distribution with some mild conditions. We assume |X| is bounded by a constant, as stated in Assumption 1, mainly to simplify the technicality. There is no similar assumption on Y, since we assume a Lipschitz smooth loss function (like Huber loss). One can prove the results for squared loss by further assuming that |Y| is bounded by a constant, as mentioned in the text after Theorem 1. Perhaps an “implicit” assumption on the data is the universal approximation (Assumption 3.3): it assumes the activation to be universal w.r.t. the input distribution. This assumption is satisfied, for instance, when the input X has the form X = (X’, 1) (where the last entry 1 corresponds to the bias term of the weight vector) and the activation is nonpolynomial. Finally in the case of non-convex losses, we assume Y is a function of X.

---

### Decision · Program_Chairs · 2021-01-07
**Final Decision**

**Decision:**

Accept (Oral)

**Comment:**

This paper provides a global convergence guarantee for feedforward three-layer networks trained with SGD in the MF regime. By introducing the novel concept of neuronal embedding of a random initialization procedure, SGD trajectories of large-width networks  are shown to be well approximated by the MF limit, a continuous-time infinite-width limit (Theorem 3). Furthermore, under some additional assumptions the MF limit is shown to converge to the global optimum when the loss is convex (Theorem 8, case 1) and for a generic loss when $y=y(x)$ is a deterministic function of input $x$ (Theorem 8, case 2). The global convergence guarantee presented in this paper is based on less restrictive assumptions compared with existing studies. All the reviewers rated this paper quite positively, with less confidence however, seemingly because of mathematical thickness of the proofs. Although the reviewers did not manage to check every detail of the proofs, they agreed that the reasoning seems mathematically sound as far as they can tell. The authors response adequately addressed minor concerns raised by the reviewers. I am thus glad to recommend acceptance of this paper.

Pros:
- Introduces the idea of a neuronal embedding, which allows establishing relation between SGD on large-width three-layer networks and its MF limit in a quantitative way with a less restrictive setting.
- Provides a global convergence guarantee under the iid initialization, in the sense that if the MF limit converges it attains the global optimum.
- Shows that the global convergence guarantee does not require convexity of the loss when a deterministic function is to be learned.

In particular, the uniform approximation property, rather than the convexity of the loss, plays a crucial role in proving the  global convergence guarantee (it allows translation of the vanishing gradient in expectation at convergence into the almost-sure vanishing gradient), which is a quite original contribution of this paper.